# Context-Alignment: Activating and Enhancing LLM Capabilities in Time Series

Yuxiao Hu[1,2*], Qian Li[3,2*], Dongxiao Zhang[2], Jinyue Yan[1], Yuntian Chen[2,4†]

[1]The Hong Kong Polytechnic University, Hong Kong, China
[2]Ningbo Institute of Digital Twin, Eastern Institute of Technology, Ningbo, China
[3]Shanghai Jiao Tong University, Shanghai, China
[4]Zhejiang Key Laboratory of Industrial Intelligence and Digital Twin, Eastern Institute of Technology, Ningbo, China
`huyuxiao20@mails.ucas.ac.cn, qianl01205@sjtu.edu.cn`
`dzhang@eias.ac.cn, j-jerry.yan@polyu.edu.hk, ychen@eitech.edu.cn,`

## Abstract

Recently, leveraging pre-trained Large Language Models (LLMs) for time series (TS) tasks has gained increasing attention, which involves activating and enhancing LLMs' capabilities. Many methods aim to activate LLMs' capabilities based on token-level alignment, but overlook LLMs' inherent strength in natural language processing — *their deep understanding of linguistic logic and structure rather than superficial embedding processing.* We propose Context-Alignment (CA), a new paradigm that aligns TS with a linguistic component in the language environments familiar to LLMs to enable LLMs to contextualize and comprehend TS data, thereby activating their capabilities. Specifically, such context-level alignment comprises structural alignment and logical alignment, which is achieved by Dual-Scale Context-Alignment GNNs (DSCA-GNNs) applied to TS-language multimodal inputs. Structural alignment utilizes dual-scale nodes to describe hierarchical structure in TS-language, enabling LLMs to treat long TS data as a whole linguistic component while preserving intrinsic token features. Logical alignment uses directed edges to guide logical relationships, ensuring coherence in the contextual semantics. Following the DSCA-GNNs framework, we propose an instantiation method of CA, termed Few-Shot prompting Context-Alignment (FSCA), to enhance the capabilities of pre-trained LLMs in handling TS tasks. FSCA can be flexibly and repeatedly integrated into various layers of pre-trained LLMs to improve awareness of logic and structure, thereby enhancing performance. Extensive experiments show the effectiveness of FSCA and the importance of Context-Alignment across tasks, particularly in few-shot and zero-shot forecasting, confirming that Context-Alignment provides powerful prior knowledge on context. The code is open-sourced at https://github.com/tokaka22/ICLR25-FSCA.

## 1 Introduction

Time series (TS) tasks are essential in many real-world applications, from weather prediction to navigation optimization. The field has shifted from traditional statistical methods to advanced neural architectures like Recurrent Neural Networks (RNNs), Convolutional Neural Networks (CNNs), and Transformers, improving the handling of complex dependencies. However, challenges in generalizing across diverse datasets and adapting to various TS tasks remain.

Recently, large language models (LLMs) have excelled in various fields, primarily due to their extensive and diverse text corpora training dataset. With a rich foundation of multi-domain knowledge,

---

[*]Alphabetical ordering: The first two authors contributed equally.
[†]Corresponding author

LLMs achieve impressive generalization in downstream tasks. Thus, there is a growing interest in utilizing pre-trained LLMs to solve TS problems. However, distinct differences between the training data of LLMs and TS data have hindered LLMs' full potential in TS applications. To effectively utilize LLMs in TS tasks, two main issues must be addressed in turn:

1) How to make LLMs understand TS data and **activate** their capabilities in TS tasks?
2) How to **enhance** the performance of LLMs on TS tasks?

Regarding the first issue, existing works primarily focus on aligning TS token embeddings with language token embeddings (Jin et al., 2024; Sun et al., 2024; Pan et al., 2024). However, whether such token-level alignment can fully leverage the LLMs' potential remains questionable. Inspired by recent research on LLMs (Ethayarajh, 2019; Nie et al., 2024; Wang et al., 2023), we reconsider the inherent advantages of LLMs in natural language processing (NLP). We believe the strength of LLMs primarily stems from their deep comprehension of language logic and structure, rather than superficial token embedding processing. Clearly, the excessive accumulation of tokens without logical guidance often struggles to effectively convey meaning. Especially TS-language multimodal inputs are lengthy, and lack structure and coherent semantics, greatly challenging LLMs' comprehension. Regarding the second issue, current methods aim to directly enhance LLMs' capabilities in TS tasks through techniques such as TS decomposition (Cao et al., 2024) and optimizing prompts (Chuang et al., 2024). However, without adequately addressing the first issue, these methods need more interpretability and the improvements remain limited. A natural solution to these issues is to fully leverage the strengths of LLMs to transform TS tasks into NLP-like tasks, activating LLMs' capabilities first. Then, leveraging NLP techniques further enhances LLMs' performance on TS tasks.

In this paper, we propose Context-Alignment, a new paradigm that aligns TS data with a linguistic component in the language environment familiar with LLMs. Such context-level alignment leverages the LLMs' inherent strength in logic and structure to enable LLMs to contextualize and comprehend TS data, thereby **activating** their capabilities. Context-Alignment contains structural alignment and logical alignment to construct a consistent context for TS-language multimodal inputs. We develop a Dual-Scale Context-Alignment Graph Neural Networks (DSCA-GNNs) framework to achieve both structural and logical alignment. Specifically, structural alignment employs dual-scale nodes to describe hierarchical structure in TS-language, i.e. the structural independence of tokens and the overall structure of modalities. Structural alignment provides LLMs with structural segmentation information for lengthy TS language inputs, enabling LLMs to treat long TS data as an individual linguistic component while preserving intrinsic token features. Logical alignment uses directed edges in both scale GNNs to guide the local and global logical relationship between TS data and language prompts, integrating TS within the language environment and ensuring semantic coherence across two modalities. Utilizing the few-shot prompting technique (Brown, 2020), we propose Few-Shot prompting based Context-Alignment (FSCA) following the DSCA-GNNs framework, which further **enhances** the LLMs' performance on TS tasks. FSCA can be flexibly and repeatedly integrated into various layers of pre-trained LLMs to improve awareness of logic and structure. Extensive experiments across various TS tasks demonstrate the effectiveness of our method. Notably, in few-shot and zero-shot forecasting tasks, our approach significantly outperforms others, confirming that the logical and structural alignment provides powerful prior knowledge on context. Ablation studies further validate the importance of Context-Alignment.

In summary, our core contributions can be summarized below:

- We emphasize that effectively leveraging LLMs for TS tasks requires first activating their capabilities and then enhancing them. Besides, we pinpoint that token-level alignment fails to fully activate pre-trained LLMs due to their neglect of LLMs' inherent strengths, which primarily stem from a deep understanding of logic and structure, rather than superficial token processing.

- We are the first to propose Context-Alignment paradigm, which aims to construct a context-level alignment between TS and language, thereby activating LLMs' potential capabilities in TS tasks.

- We develop a Dual-Scale Context-Alignment GNNs framework, which achieves structural and logical alignment through dual-scale nodes and directed edges, thus realizing Context-Alignment. Furthermore, by integrating the few-shot prompting technique, we introduce (FSCA), which enhances LLMs' performance in TS tasks.

• Our experiments across multiple datasets and various TS tasks demonstrate that our method surpasses existing techniques, especially in few-shot and zero-shot forecasting tasks. Ablation studies further emphasize the importance of Context-Alignment.

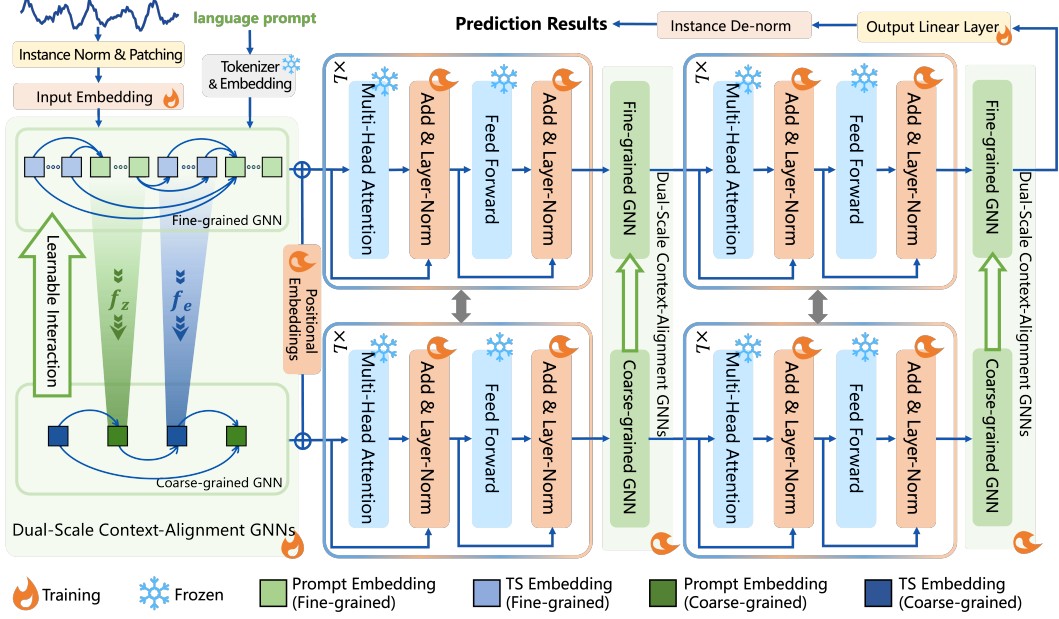

Figure 1: The architecture of our method, where the graph structure demonstrates the prediction task based on FSCA as detailed in Sec.3.3. Dual-Scale Context-Alignment GNNs can be flexibly and repeatedly integrated into pre-trained LLMs at various layers, enhancing LLMs' awareness of logic and structure and improving performance.

# 2 RELATED WORK

## 2.1 TIME SERIES TASKS

TS tasks are crucial for various applications, including financial forecasting, weather prediction, and activity recognition, and involve analyzing time-ordered data. Traditional statistical methods like ARIMA (Anderson, 1976) and Prophet (Taylor & Letham, 2018) effectively model trends and seasonality. The rise of deep learning introduced CNN-based methods, which use convolutional neural networks to extract features automatically (Bai et al., 2018; Wu et al., 2022). Additionally, RNNs, such as LSTM and GRU, excel in sequence prediction by capturing dynamic temporal behaviors and long-range dependencies (Lai et al., 2018b; Qin et al., 2017; Siami-Namini et al., 2018). More recently, Transformer-based models have advanced the field by processing sequences in parallel and applying attention mechanisms to focus on significant temporal aspects, thus improving performance in complex scenarios (Wen et al., 2023; Zhou et al., 2022; Wu et al., 2021). However, these methods often struggle with long, intricate sequences and lack the flexibility and generalizability required for diverse real-world TS data across diverse domains.

## 2.2 LARGE LANGUAGE MODELS FOR TIME SERIES

LLMs have demonstrated strong capabilities in various fields, becoming a focal point for advancing TS tasks. Recent research highlights their potential in TS tasks. Some methods aim to directly enhance the capabilities of LLMs for TS tasks. For example, Cao et al. (2024) captures complex interactions among trend, seasonal, and residual components to aid distribution adaptation. Chuang et al. (2024) propose a statistical prompting strategy to enhance the performance. However, these methods overlook an important step: enable LLMs to understand TS inputs first. Other methods aim to enable LLMs to understand TS data and activate their potential for TS tasks. For example, Jin et al. (2024) reprograms input TS into text prototypes and enhances them with prompts. Sun

et al. (2024) aligns TS embeddings space with LLMs embeddings space. Pan et al. (2024) proposes S$^2$IP-LLM, a semantic space-informed prompt learning to align TS embeddings with language token embeddings. However, these methods overlook that LLMs' inherent strength stems from their understanding of the logic and structure, rather than superficial token embedding processing. Merely aligning token embeddings without considering the coherence and consistency of the context fails to leverage this inherent advantage.

# 3 METHODOLOGY

To address both structural and logical alignment in Context-Alignment, we develop a Dual-Scale Context-Alignment GNNs (DSCA-GNNs) framework. In this framework, dual-scale nodes achieve structural alignment, while directed edges realize logical alignment. Additionally, the specific structure of the graph depends on the language prompts introduced, as different prompts lead to varying logic and structures in the TS-language. That is, each prompting method corresponds to a specific DSCA-GNNs framework. In this section, we present the corresponding DSCA-GNNs frameworks for both the vanilla prompt and the demonstration examples prompt.

## 3.1 PRELIMINARIES AND TOKEN EMBEDDING

**Time-Series Forecasting.** Given past data $\boldsymbol{X} \in \mathbb{R}^{D \times T}$, where each row represents a TS of $T$ steps across $D$ variables, the goal is to build a predictive model $\mathcal{F}$ with parameters $\boldsymbol{\Theta}_{\mathcal{F}}$ and prompt $\boldsymbol{P}_{\mathcal{F}}$ to forecast the next $T'$ time steps. The forecasting model can be formulated as $\hat{\boldsymbol{X}} = \mathcal{F}(\boldsymbol{X}; \boldsymbol{P}_{\mathcal{F}}; \boldsymbol{\Theta}_{\mathcal{F}})$.

**Time-Series Classification.** Given TS data $\boldsymbol{X} \in \mathbb{R}^{D \times T}$, the task is to build a classification model $\mathcal{C}$ with parameters $\boldsymbol{\Theta}_{\mathcal{C}}$ and prompt $\boldsymbol{P}_{\mathcal{C}}$ to assign a class label $\hat{y} \in \{1, 2, ..., C\}$, where $C$ is the number of classes. The classification is defined as $\hat{y} = \mathcal{C}(\boldsymbol{X}; \boldsymbol{\Theta}_{\mathcal{C}}, \boldsymbol{P}_{\mathcal{C}})$.

**Token Embedding.** Multivariate TS data $\boldsymbol{X}$ is segmented into patches $\boldsymbol{X}_i$ using a sliding window of size $p$ and stride $s$. Each patch is embedded into an $M$-dimensional space compatible with LLMs. The embedded patches are denoted as $\{\boldsymbol{e}_i\}_{i=1}^n$, where $n = \frac{T-p+s}{s}$ is the number of patches.

## 3.2 DEMO: VANILLA CONTEXT-ALIGNMENT (VCA)

The most straightforward approach to utilizing pre-trained LLMs for TS tasks is to input both TS data and vanilla language prompts into the model directly. For TS forecasting tasks, a vanilla prompt like "Predict future sequences using previous data:" can be employed to guide LLMs in completing the prediction. The embeddings of the prompt can be represented as $\{\boldsymbol{z}_1, \boldsymbol{z}_2, \ldots, \boldsymbol{z}_m\}$, and the overall input embeddings of the model can be represented as below:

$$[\boldsymbol{e}_1, \boldsymbol{e}_2, \ldots, \boldsymbol{e}_n, \boldsymbol{z}_1, \boldsymbol{z}_2, \ldots, \boldsymbol{z}_m], \quad \boldsymbol{e}_i, \boldsymbol{z}_j \in \mathbb{R}^M. \tag{1}$$

Due to the verbose and lack of clear structural divisions of TS embeddings $\{\boldsymbol{e}_i\}_{i=1}^n$, LLMs lack information that TS embedding $\{\boldsymbol{e}_i\}_{i=1}^n$ is an integral entity, making it difficult to analyze TS. Furthermore, form 1 lacks logical guidance. Directly concatenating the TS embeddings and language prompt embeddings loses essential contextual coherence between TS data and the prompt.

**Dual-Scale Context-Alignment GNNs of VCA.** Context-Alignment leverages the LLMs' inherent strength in logic and structure to enable LLMs to contextualize and comprehend TS data. We delineate clear structures (structural alignment) and establish correct logical guidance (logical alignment) by a Dual-Scale Context-Alignment GNNs framework to achieve Context-Alignment. Firstly, using dual-scale nodes, structural alignment aggregates tokens from the same modality into one linguistic component while preserving the feature of each token. Fine-grained GNN $G_F$ treats each token, i.e., each element in the form 1 as a node. Coarse-grained GNN $G_C$ treats consecutive tokens with the same modality as a node. Specifically, $G_C$ employs two learnable linear layers, $f_e$ and $f_z$, to embed the TS tokens and language tokens into an $M$-dimensional space, respectively, which can be formalized as

$$\tilde{\boldsymbol{e}} = f_e(\boldsymbol{e}_1, \boldsymbol{e}_2, \ldots, \boldsymbol{e}_n); \quad \tilde{\boldsymbol{z}} = f_z(\boldsymbol{z}_1, \boldsymbol{z}_2, \ldots, \boldsymbol{z}_m).$$

Then, form 1 is transformed into form 2

$$[\tilde{\boldsymbol{e}}, \tilde{\boldsymbol{z}}], \quad \tilde{\boldsymbol{e}}, \tilde{\boldsymbol{z}} \in \mathbb{R}^M. \tag{2}$$

Each element in the form 2 is regarded as a node in $G_C$. Secondly, using directed edges, logical alignment emphasizes the correct semantic association between different components. Clearly, in the prompt "Predict future sequences using previous data:", "previous data:" refers to the lengthy TS data, and requires information from TS data. Therefore, in $G_C$, we construct directed edge $E : \tilde{e} \to \tilde{z}$ to indicate that the entire TS data serves as the upstream information source for the prompt. In $G_F$, we construct directed edges $\{E_{ij} : e_i \to z_j | i \in \{1, \ldots, n\}, j \in \{1, \ldots, m\}\}$ to convey varying information from each TS token to the prompt token. We also constrain the sum of edge weights from TS tokens to one language token to be 1, i.e. $\sum_{i=1}^{n} w_{ij} = 1$, $w_{ij}$ is the edge weight of $E_{ij}$, implicitly emphasizing that TS should be treated as a whole. In $G_C$, the weights of all directed edges are set to 1. Besides, $\{w_{ij}\}_{i=1}^{n}$ are proportional to the cosine similarity between the embeddings of two nodes.

The updated node embedding matrices for $G_F$ and $G_C$ denoted as $\hat{N}_F$ and $\hat{N}_C$, respectively. Based on the GNN update strategy, $\hat{N}_F$ and $\hat{N}_C$ can be formalized as:

$$\hat{N}_k = \sigma(D_k^{-\frac{1}{2}} A_k' D_k^{-\frac{1}{2}} N_k^T W_k), \quad k \in \{F, C\}, \tag{3}$$

where $N_k^T$ denotes the transpose of the pre-update node embedding matrix. In $G_F$, $N_F$ is shown in form 1, while in $G_C$, $N_C$ follows form 2. Besides, $W_k \in \mathbb{R}^{M \times M}$ represents the learnable matrix of the GNN, $\sigma$ denotes the nonlinear activation function ReLU, and $A_k' = A_k + I$ is the weighted adjacency matrix with unit matrix added. $D_k$ is a diagonal matrix where $D_{k,ii} = \Sigma_j A_{k,ij}'$.

**Learnable interaction.** Macroscopic logical-structural information of $G_C$ helps LLMs understand TS, but loses key details necessary for TS tasks. In contrast, $G_F$ retains detailed information. Thus, we introduce a learnable interaction to transfer macroscopic information from $G_C$ to $G_F$. The learnable interactions can be represented by the following formula:

$$\Delta N = W_{c \to f} \hat{N}_C \Gamma_{c \to f}, \tag{4}$$

where $\Gamma_{c \to f} \in \mathbb{R}^{2 \times (n+m)}$ is a $0-1$ assignment matrix, $\Gamma_{ij} = 1$ means that the $j$-th node in $G_F$ be aggregated into the $i$-th node in $G_C$. $W_{c \to f} \in \mathbb{R}^{M \times M}$ is a learnable weight matrix. $\hat{N}_F$ can be updates as $\hat{N}_F \leftarrow \hat{N}_F + \Delta N$ after aggregating the information from $\hat{N}_C$.

**Overview.** After structural and logical alignment, $\hat{N}_F$ maintains clear structural and coherent semantics information, helping LLMs to comprehend TS tasks and activating potential capabilities. Both $\hat{N}_F$ and $\hat{N}_C$ are input into pre-trained LLMs. The DSCA-GNNs can be flexibly integrated into various layers of pre-trained LLMs, and only the first time apply it need $f_e$ and $f_z$ to obtain coarse-grained GNN. The output from the $G_F$ branch is used to compute the MSE loss against the ground truth. In this section, we use the vanilla prompt to construct a specific DSCA-GNNs framework, we call it Vanilla Context-Alignment (VCA).

### 3.3 FEW-SHOT PROMPTING BASED CONTEXT-ALIGNMENT (FSCA)

The demo VCA in Sec. 3.2 is the simplest and most direct attempt of Context-Alignment. Moving forward, we will naturally consider whether Context-Alignment can further enhance the performance of LLMs on TS tasks by leveraging more advanced prompt techniques from NLP.

In NLP, "few-shot prompting" refers to a small set of instances provided to the model to demonstrate a task, enabling the model to perform similar tasks effectively (Brown, 2020). Based on this, we divide the TS embeddings $\{e_i\}_{i=1}^{n}$ into $N$ parts while preserving their original order. Since the subsequent TS part can be used as ground truth for predictions based on the preceding TS parts, we can construct $N - 1$ prediction demonstration examples using $N$ parts of $\{e_i\}_{i=1}^{n}$ and language prompt embeddings $\{z_1, z_2, \ldots, z_m\}$ same as demo in Sec. 3.2. The $j$-th part of $\{e_i\}_{i=1}^{n}$ is denoted as $\{e_{j,1}, \ldots, e_{j,l_j}\}$. We arrange the TS-language embeddings in the format as:

$$[e_{1,1}, \ldots, e_{1,l_1}, z_1, \ldots, z_m, e_{2,1}, \ldots, e_{2,l_2}, z_1, \ldots, z_m, \ldots, e_{N,1}, \ldots, e_{N,l_N}, z_1, \ldots, z_m]. \tag{5}$$

**Dual-Scale Context-Alignment GNNs of FSCA.** Due to the verbose and lack of clear structural divisions of the format 5, LLMs struggle to understand the input. We first construct the coarse-grained GNN $G_C$. We introduce two learnable linear layers $f_e$ and $f_z$ to embed the TS tokens and language tokens into an $M$-dimensional space, respectively, which can be formalized as

$$\tilde{e}_j = f_e(e_{j,1}, e_{j,2}, \ldots, e_{j,l_j}); \quad \tilde{z} = f_z(z_1, z_2, \ldots, z_m).$$

Then form 5 is transformed into form 6

$$[\tilde{\boldsymbol{e}}_1, \tilde{\boldsymbol{z}}^{(1)}, \tilde{\boldsymbol{e}}_2, \tilde{\boldsymbol{z}}^{(2)}, \ldots, \tilde{\boldsymbol{e}}_N, \tilde{\boldsymbol{z}}^{(N)}], \quad \tilde{\boldsymbol{e}}_j, \tilde{\boldsymbol{z}} \in \mathbb{R}^M, \quad j = 1, \ldots, N. \tag{6}$$

For clarity in our discussion, we number the $\tilde{\boldsymbol{z}}$ in form 6, in fact, $\tilde{\boldsymbol{z}} = \tilde{\boldsymbol{z}}^{(i)} = \tilde{\boldsymbol{z}}^{(j)}, i, j = 1, \ldots, N$. The directed edge set of the $G_C$ can be represented as follows:

$$\{E_C : \tilde{\boldsymbol{e}}_j \to \tilde{\boldsymbol{z}}^{(i)} | i = 1, \ldots, N, j = 1, \ldots, i\} \cup \{E_C : \tilde{\boldsymbol{z}}^{(i)} \to \tilde{\boldsymbol{e}}_{i+1} | i = 1, \ldots, N-1\}. \tag{7}$$

In $G_C$, there are two types of edges, which guide two different logical relationships. The first type, as shown in the first item of 7, signifies that the prompt $\tilde{\boldsymbol{z}}^{(i)}$ receives TS information from all preceding TS parts. The second type, as indicated in the second item of 7, implies that $\tilde{\boldsymbol{e}}_{i+1}$ is the correct output result of prompt $\tilde{\boldsymbol{z}}^{(i)}$. In $G_C$, the weights of directed edges are set to 1.

Fine-grained GNN $G_F$ treats each token in the form 5 as a node. The directed edge set of the $G_F$ can be represented as follows:

$$\begin{aligned}&\{E_F : \boldsymbol{e}_{j,s} \to \boldsymbol{z}_t^{(i)} | i = 1, \ldots, N, j = 1, \ldots, i, s = 1, \ldots, l_j, t = 1, \ldots, m\}\\ &\cup\{E_F : \boldsymbol{z}_t^{(i)} \to \boldsymbol{e}_{i+1,s} | i = 1, \ldots, N-1, s = 1, \ldots, l_{i+1}, t = 1, \ldots, m\},\end{aligned} \tag{8}$$

The formula 8 means that the directed edges of $G_F$ are a decomposition of the directed edges in $G_C$. Additionally, since LLMs have strong comprehension abilities for language prompts, we can prune the directed edges in $G_F$, transforming form 8 into form 9, i.e., TS tokens are only connected to the first and last tokens of the prompt, thereby prevent overfitting. We constrain $\sum_{s=1}^{l_j} w_{j,s}^{(i)} = 1$ for the first type edges, $\sum_{s=1}^{l_{i+1}} w_{i+1,s}^{(i)} = 1$ for the second type edges, where $w_{j,s}^{(i)}$ is the edge weight of $\boldsymbol{e}_{j,s} \to \boldsymbol{z}_1^{(i)}$, $w_{i+1,s}^{(i)}$ is the edge weight of $\boldsymbol{z}_m^{(i)} \to \boldsymbol{e}_{i+1,s}$. $\{w_{i+1,s}^{(i)}\}_{s=1}^{l_j}$ and $\{w_{i+1,s}^{(i)}\}_{s=1}^{l_{i+1}}$ are proportional to the cosine similarity between node embeddings. Fig. 1 provides a schematic of the structure. The node embedding update formula is similar to formula 3.

$$\begin{aligned}&\{E_F : \boldsymbol{e}_{j,s} \to \boldsymbol{z}_1^{(i)} | i = 1, \ldots, N, j = 1, \ldots, i, s = 1, \ldots, l_j\}\\ &\cup\{E_F : \boldsymbol{z}_m^{(i)} \to \boldsymbol{e}_{i+1,s} | i = 1, \ldots, N-1, s = 1, \ldots, l_{i+1}\},\end{aligned} \tag{9}$$

**Learnable interactions & Overview.** Similar to the demo in Sec. 3.2, we introduce an assignment matrix and a learnable weight matrix to achieve learnable interactions between the two scales. The training of FSCA can refer to Sec. 3.2 and Fig. 1.

## 4 EXPERIMENTS

The proposed Context-Alignment demonstrates robust performance across a variety of tasks, detailed in Sec. 4.2 (long-term forecasting), Sec. 4.3 (short-term forecasting), Sec. 4.4 (few-shot forecasting), Sec. 4.5 (zero-shot forecasting), and Sec. 4.6 (classification). Most experiments leverage FSCA. Specifically, VCA validates its efficacy using logic guidance and structural division alone in section 4.1. For classification tasks with multiple classes, where GPT-2's length constraints are limiting, FSCA is reserved for binary classes, and VCA is applied to multi-class datasets. Context-Alignment significantly boosts training efficiency and cost-effectiveness, especially in few-shot and zero-shot forecasting, by establishing robust a priori structural understanding and logical relationships. Full results and dataset description are in Appendix C and A.2, respectively.

**Baselines.** Building on the groundwork of Zhou et al. (2023) and Jin et al. (2024), and mindful of page constraints, we selected a representative array of high-performing baselines for extensive evaluation. These encompass Transformer-based models such as iTransformer (Liu et al., 2023b), FEDformer (Zhou et al., 2022), Non-stationary Transformer (Liu et al., 2022), ETSformer (Woo et al., 2022), PatchTST (Nie et al., 2022), alongside notable non-Transformer methods like Times-Net (Wu et al., 2022), and DLinear (Zeng et al., 2023). We also included advanced LLM-based models—GPT4TS (Zhou et al., 2023), Time-LLM (Jin et al., 2024), and S$^2$IP-LLM (Pan et al., 2024)—all utilizing GPT-2 as the standard LLM backbone to ensure model consistency. To ensure a fair comparison, we adhere to the experimental framework outlined in Zhou et al. (2023) and Wu et al. (2022). Detailed evaluations are expanded upon in subsequent sections.

## 4.1 Demo: Vanilla Context-Alignment

VCA achieves logical and structural alignment through DSCA-GNNs. As demonstrated on the ETT dataset (Table 1), VCA substantially outperforms both variants without DSCA-GNNs and other baselines. A performance decline is observed compared to FSCA, which further augments LLMs' TS processing capabilities using demonstration examples prompt. VCA without DSCA-GNNs, despite incorporating task description prompts, lacks context-alignment GNNs, resulting in more verbose and semantically confused inputs for LLM, thus yielding the worst outcomes.

Table 1: Results for VCA and variants. **Bold**: best, Underline: second best.

| Method/Variant | Long-term Forecasting | | | |
|---|---|---|---|---|
| | ETTh1 | ETTh2 | ETTm1 | ETTm2 |
| GPT4TS | 0.427 | 0.354 | 0.352 | 0.266 |
| FSCA | **0.394** | **0.316** | **0.342** | **0.250** |
| VCA w/o DSCA-GNNs | 0.435 | 0.362 | 0.374 | 0.271 |
| VCA | 0.417 | 0.335 | 0.349 | 0.259 |

## 4.2 Long-term Forecasting

**Setups.** For long-term forecasting tasks, we validate the efficacy of FSCA across eight prevalent datasets (Wu et al., 2022): ETTh1, ETTh2, ETTm1, ETTm2, Weather, Electricity, Traffic, and ILI. Consistent with GPT4TS (Zhou et al., 2023), Time-LLM (Jin et al., 2024), and $S^2$IP-LLM (Pan et al., 2024), we utilize an input TS length of 512 except for the ILI dataset. Performance is evaluated over four prediction horizons: $\{24, 36, 48, 60\}$ for ILI and $\{96, 192, 336, 720\}$ for the other datasets, using Mean Squared Error (MSE) and Mean Absolute Error (MAE) as metrics.

**Results.** As shown in Table 2, FSCA surpasses all baseline methods in most scenarios. Specifically, it reduces average MSE of 3.1% over the suboptimal method PatchTST, and outperforms other LLM-based methods—$S^2$IP-LLM, Time-LLM, and GPT4TS by 7.3%, 12.2%, and 16.6%. This consistent superiority across diverse datasets highlights the critical role of logical and structural alignment. Furthermore, demonstration examples prompt boost LLMs' contextual understanding of TS data.

Table 2: Long-term forecasting tasks, all results are based on different horizons: {24, 36, 48, 60} for ILI and {96, 192, 336, 720} for others. **Bold**: best, Underline: second best. Full results are provided in Appendix C.1

| Methods | FSCA | | $S^2$IP-LLM | | Time-LLM | | GPT4TS | | iTransformer | | DLinear | | PatchTST | | TimesNet | | FEDformer | | Stationary | | ETSformer | |
|---|---|---|---|---|---|---|---|---|---|---|---|---|---|---|---|---|---|---|---|---|---|---|
| Metric | MSE | MAE | MSE | MAE | MSE | MAE | MSE | MAE | MSE | MAE | MSE | MAE | MSE | MAE | MSE | MAE | MSE | MAE | MSE | MAE | MSE | MAE |
| ILI | **1.380** | **0.783** | 1.552 | 0.826 | 1.713 | 0.858 | 1.925 | 0.903 | 2.073 | 0.941 | 2.169 | 1.041 | 1.443 | 0.797 | 2.139 | 0.931 | 2.847 | 1.144 | 2.077 | 0.914 | 2.497 | 1.004 |
| Weather | **0.224** | **0.262** | 0.228 | 0.265 | 0.237 | 0.269 | 0.237 | 0.270 | 0.304 | 0.335 | 0.248 | 0.300 | 0.225 | 0.264 | 0.259 | 0.287 | 0.309 | 0.360 | 0.288 | 0.314 | 0.271 | 0.334 |
| ECL | **0.159** | **0.252** | 0.166 | 0.262 | 0.167 | 0.264 | 0.167 | 0.263 | 0.203 | 0.298 | 0.166 | 0.263 | 0.161 | **0.252** | 0.192 | 0.295 | 0.214 | 0.327 | 0.193 | 0.296 | 0.208 | 0.323 |
| Traffic | **0.386** | **0.263** | 0.405 | 0.286 | 0.407 | 0.289 | 0.414 | 0.294 | 0.389 | 0.295 | 0.433 | 0.295 | 0.390 | **0.263** | 0.620 | 0.336 | 0.610 | 0.376 | 0.624 | 0.340 | 0.621 | 0.396 |
| ETTh1 | **0.394** | **0.424** | 0.418 | 0.436 | 0.426 | 0.435 | 0.427 | 0.426 | 0.451 | 0.462 | 0.422 | 0.437 | 0.413 | 0.430 | 0.458 | 0.450 | 0.440 | 0.460 | 0.570 | 0.537 | 0.542 | 0.510 |
| ETTh2 | **0.316** | **0.375** | 0.355 | 0.399 | 0.361 | 0.398 | 0.354 | 0.394 | 0.382 | 0.414 | 0.431 | 0.446 | 0.330 | 0.379 | 0.414 | 0.427 | 0.437 | 0.449 | 0.526 | 0.516 | 0.439 | 0.452 |
| ETTm1 | **0.342** | **0.378** | 0.346 | 0.382 | 0.354 | 0.384 | 0.352 | 0.383 | 0.370 | 0.399 | 0.357 | **0.378** | 0.351 | 0.380 | 0.400 | 0.406 | 0.448 | 0.452 | 0.481 | 0.456 | 0.429 | 0.425 |
| ETTm2 | **0.250** | **0.314** | 0.262 | 0.326 | 0.275 | 0.334 | 0.266 | 0.326 | 0.272 | 0.331 | 0.267 | 0.333 | 0.255 | 0.315 | 0.291 | 0.333 | 0.305 | 0.349 | 0.306 | 0.347 | 0.293 | 0.342 |

## 4.3 Short-term Forecasting

Table 3: Short-term forecasting on M4, with prediction horizons ranging from [6, 48]. The results are weighted averages across all datasets under different sampling intervals. **Bold**: best, Underline: second best. Full results are provided in Appendix C.2

| | Methods | FSCA | $S^2$IP-LLM | Time-LLM | GPT4TS | iTransformer | DLinear | PatchTST | N-HiTS | N-BEATS | TimesNet | FEDformer | Stationary |
|---|---|---|---|---|---|---|---|---|---|---|---|---|---|
| Average | SMAPE | **11.828** | 12.021 | 12.494 | 12.690 | 12.142 | 13.639 | 12.059 | 12.035 | 12.250 | 12.880 | 13.160 | 12.780 |
| | MASE | **1.580** | 1.612 | 1.731 | 1.808 | 1.631 | 2.095 | 1.623 | 1.625 | 1.698 | 1.836 | 1.775 | 1.756 |
| | OWA | **0.850** | 0.857 | 0.913 | 0.940 | 0.869 | 1.051 | 0.869 | 0.869 | 0.896 | 0.955 | 0.949 | 0.930 |

**Setups.** We conduct short-term forecasting experiments on the M4 dataset (Makridakis et al., 2018), which includes market data across various frequencies, with prediction horizons ranging from 6 to 48. We incorporate N-HiTS (Challu et al., 2023) and N-BEATS (Oreshkin et al., 2019) as additional baselines. Performance is quantified using symmetric mean absolute percentage error (SMAPE), mean absolute scaled error (MASE), and the overall weighted average (OWA) as evaluation metrics.

**Results.** Results in Table 3 indicate that FSCA exhibits competitive performance compared to SOTA methods. FSCA maintains robustness in both long-term and short-term forecasting, attributable to the effectiveness of structural and logical alignment across varying sequence lengths.

## 4.4 FEW-SHOT FORECASTING

Table 4: Few-shot forecasting task on 5% training data. Results are averaged across different prediction lengths {96, 192, 336, 720}. **Bold**: best, Underline: second best. Full results are provided in Appendix C.3

| Methods | FSCA | | S²IP-LLM | | Time-LLM | | GPT4TS | | iTransformer | | DLinear | | PatchTST | | TimesNet | | FEDformer | | Stationary | | ETSformer | |
|---|---|---|---|---|---|---|---|---|---|---|---|---|---|---|---|---|---|---|---|---|---|---|
| Metric | MSE | MAE | MSE | MAE | MSE | MAE | MSE | MAE | MSE | MAE | MSE | MAE | MSE | MAE | MSE | MAE | MSE | MAE | MSE | MAE | MSE | MAE |
| ETTh1 | **0.575** | **0.508** | 0.650 | 0.550 | 0.648 | 0.549 | 0.681 | 0.560 | 1.070 | 0.710 | 0.750 | 0.611 | 0.695 | 0.569 | 0.925 | 0.647 | 0.658 | 0.562 | 0.943 | 0.646 | 1.189 | 0.839 |
| ETTh2 | **0.366** | **0.397** | 0.380 | 0.413 | 0.398 | 0.426 | 0.400 | 0.433 | 0.488 | 0.475 | 0.827 | 0.615 | 0.439 | 0.448 | 0.463 | 0.454 | 0.463 | 0.454 | 0.470 | 0.489 | 0.809 | 0.681 |
| ETTm1 | 0.435 | 0.429 | 0.455 | 0.446 | 0.477 | 0.451 | 0.472 | 0.450 | 0.784 | 0.596 | **0.400** | **0.417** | 0.526 | 0.476 | 0.717 | 0.561 | 0.730 | 0.592 | 0.857 | 0.598 | 1.125 | 0.782 |
| ETTm2 | **0.284** | **0.332** | 0.296 | 0.342 | 0.307 | 0.348 | 0.308 | 0.346 | 0.356 | 0.388 | 0.399 | 0.426 | 0.314 | 0.352 | 0.344 | 0.372 | 0.381 | 0.404 | 0.341 | 0.372 | 0.534 | 0.547 |
| Average | **0.415** | **0.416** | 0.445 | 0.438 | 0.458 | 0.443 | 0.465 | 0.447 | 0.675 | 0.542 | 0.594 | 0.517 | 0.493 | 0.461 | 0.612 | 0.509 | 0.558 | 0.503 | 0.653 | 0.526 | 0.914 | 0.712 |

**Setups.** The expressive potential of LLMs often results in a robust performance in few-shot scenarios (Brown, 2020; Liu et al., 2023a). While current LLM-based methods outperform earlier models like DLinear, PatchTST, and TimesNet, they do not fully exploit LLMs' deep TS comprehension, thus not maximizing their few-shot capabilities. To explore whether Context-Alignment can boost the few-shot efficacy of LLMs, we perform experiments on the ETT datasets, following the protocol established by Jin et al. (2024). We evaluate the performance of FSCA using only 5% of training data, with results from 10% data detailed in Appendix C.3.

**Results.** As shown in Table 4, our method consistently outperforms all baselines. It achieves a 6.7% reduction in average MSE compared to the leading LLM-based model, S²IP-LLM. We observe further improvements of 9.4% and 10.8% against Time-LLM and GPT4TS. Moreover, FSCA shows a 15.8% performance gain over the SOTA transformer model PatchTST. We attribute these to the integration of prior knowledge in structural division and logic guidance by Context-Alignment. The strong generalizability of these priors, even with limited training data, effectively activates LLMs' latent few-shot capabilities in TS, further boosted by a few demonstration examples.

## 4.5 ZERO-SHOT FORECASTING

**Setups.** Despite the inherent zero-shot capabilities of LLMs (Kojima et al., 2022), LLM-based methods struggle to fundamentally understand the structure of TS data and its logical associations with prompts, leading to underperformance compared to Transformer-based methods like PatchTST. Using the setup from Time-LLM (Jin et al., 2024), we also evaluate the cross-domain efficacy of FSCA on ETT dataset. Specifically, the model is trained on Dataset A and then tested on Dataset B without utilizing any training data from Dataset B.

Table 5: Zero-shot learning results: the first column A → B indicates training on dataset A and testing on dataset B. **Bold**: best, Underline: second best. Full results are provided in Appendix C.4

| Methods | FSCA | | S²IP-LLM | | Time-LLM | | GPT4TS | | iTransformer | | DLinear | | PatchTST | | TimesNet | |
|---|---|---|---|---|---|---|---|---|---|---|---|---|---|---|---|---|
| Metric | MSE | MAE | MSE | MAE | MSE | MAE | MSE | MAE | MSE | MAE | MSE | MAE | MSE | MAE | MSE | MAE |
| ETTh1 → ETTh2 | **0.313** | **0.369** | 0.403 | 0.417 | 0.384 | 0.409 | 0.406 | 0.422 | 0.457 | 0.455 | 0.493 | 0.488 | 0.380 | 0.405 | 0.421 | 0.431 |
| ETTh1 → ETTm2 | **0.290** | **0.348** | 0.325 | 0.360 | 0.317 | 0.370 | 0.325 | 0.363 | 0.360 | 0.390 | 0.415 | 0.452 | 0.314 | 0.360 | 0.327 | 0.361 |
| ETTh2 → ETTh1 | **0.527** | **0.507** | 0.669 | 0.560 | 0.663 | 0.540 | 0.757 | 0.578 | 0.868 | 0.625 | 0.703 | 0.574 | 0.565 | 0.513 | 0.865 | 0.621 |
| ETTh2 → ETTm2 | **0.288** | **0.347** | 0.327 | 0.363 | 0.339 | 0.371 | 0.335 | 0.370 | 0.335 | 0.382 | 0.328 | 0.386 | 0.325 | 0.365 | 0.342 | 0.376 |
| ETTm1 → ETTh2 | **0.353** | **0.398** | 0.442 | 0.439 | 0.440 | 0.449 | 0.433 | 0.439 | 0.455 | 0.458 | 0.464 | 0.475 | 0.439 | 0.438 | 0.457 | 0.454 |
| ETTm1 → ETTm2 | **0.264** | **0.319** | 0.304 | 0.347 | 0.311 | 0.343 | 0.313 | 0.348 | 0.319 | 0.363 | 0.335 | 0.389 | 0.296 | 0.334 | 0.322 | 0.354 |
| ETTm2 → ETTh2 | **0.343** | **0.393** | 0.406 | 0.429 | 0.429 | 0.448 | 0.435 | 0.443 | 0.432 | 0.447 | 0.455 | 0.471 | 0.409 | 0.425 | 0.435 | 0.443 |
| ETTm2 → ETTm1 | **0.480** | **0.463** | 0.622 | 0.532 | 0.588 | 0.503 | 0.769 | 0.567 | 0.706 | 0.572 | 0.649 | 0.537 | 0.568 | 0.492 | 0.769 | 0.567 |
| Average | **0.357** | **0.393** | 0.437 | 0.431 | 0.434 | 0.429 | 0.472 | 0.441 | 0.491 | 0.461 | 0.480 | 0.472 | 0.412 | 0.417 | 0.492 | 0.451 |

**Results.** Table 5 shows that FSCA substantially outperforms the most competitive baselines. Compared to the second-best model, PatchTST, FSCA exhibits a 13.3% improvement in performance. Against LLM-based models like S²IP-LLM, Time-LLM, and GPT4TS, FSCA achieves performance gains of 18.3%, 17.7%, 24.3%. Experiments in both few-shot and zero-shot settings highlight FSCA's exceptional performance under data-scarce conditions. In these scenarios, Context-Alignment paradigm provides a robust contextual prior, enabling accurate logical and structural understanding that enhances the potential of LLMs for cross-domain TS processing.

## 4.6 TIME SERIES CLASSIFICATION

**Setups.** To assess the model's capability in learning advanced representations, we conduct comparative experiments on series classification using the setup outlined by Zhou et al. (2023). We select 10 multivariate UEA classification datasets (Bagnall et al., 2018) from domains including ECG, audio, gesture, spectral recognition, etc. For binary class datasets like FaceDetection, we apply FSCA framework, adding a demonstration example prompt for each category at the beginning of the prediction sequence. For datasets with multiple classes, such as Handwriting with 26 classes, we employ VCA due to GPT-2's input length constraints.

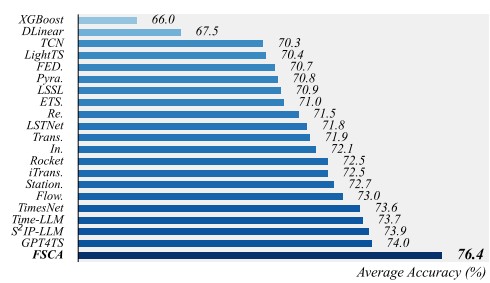

Figure 2: For the classification task, the results show average accuracy across 10 subsets from the UEA dataset. Detailed results are in Appendix F.1.

**Results.** Fig. 2 presents the average accuracy of various methods on all UEA classification datasets. FSCA achieves an accuracy of 76.4%, surpassing all baseline methods and recording a 2.4% increase over the next best model, Zhou et al. (2023). This performance suggests the effectiveness of FSCA or VCA extends beyond predictive tasks. Additionally, Context-Alignment shows generality, indicating its potential applicability across various contexts in the field.

## 4.7 ABLATION STUDY

As shown in Table 6, we conduct an ablation study on the framework design and analyze the effectiveness of FSCA. FSCA* represents the optimal results, highlighted in bold in Table 6. D.4 is optimal for long-term forecasting, whereas D.3 is best for classification tasks.

**Validity of Dual-Scale Context-Alignment GNNs.** To evaluate DSCA-GNNs, we conduct comparative experiments from two perspectives. In **A.1**, without Dual-Scale GNNs, we rely solely on demonstration examples as prompts, resulting in higher average MSE compared to FSCA*. In **A.2**, random initialization of adjacency matrix leads to further performance decline compared to A.1, underscoring that incorrect logical information can impair model performance. A comprehensive comparison across A.1, A.2, and FSCA* confirms that GNNs guided by appropriate logical frameworks effectively leverage the inherent capabilities of LLMs in TS tasks.

**Validity of Coarse-Grained Branch.** In **B.1**, omitting this module impairs the model's ability to understand the overall structure and the macro-level logical relationships, resulting in reduced performance compared to FSCA*. This decline is due to reliance solely on the fine-grained branch for token-scale context alignment, which still leaves the inputs verbose and lacking structure.

Table 6: Ablations results (average MSE for four prediction lengths in long-term forecasting and accuracy for the classification). **Bold**: best. In C.*, default insertion positions are the first and last layers; therefore, C.2 and D.3 are identical.

| Variant | Long-term Forecasting | | Classification | |
|---|---|---|---|---|
| | ETTh1 | ETTm1 | FaceDet. | Heartbeat |
| Metric | MSE | | Accuracy | |
| **A.1** w/o Dual-Scale GNNs | 0.441 | 0.379 | 67.7 | 76.1 |
| **A.2** with Random Init | 0.463 | 0.392 | 64.5 | 73.1 |
| **B.1** w/o Coarse-grained Branch | 0.401 | 0.353 | 69.6 | 78.5 |
| **C.1** FSCA(2) | 0.402 | 0.357 | 69.1 | 78.0 |
| **C.2** FSCA(**4**) | 0.396 | 0.345 | **70.4** | **79.5** |
| **C.3** FSCA(6) | 0.399 | 0.347 | 70.1 | 79.0 |
| **C.4** FSCA(8) | 0.418 | 0.362 | 67.7 | 77.0 |
| **C.5** FSCA(10) | 0.439 | 0.383 | 66.4 | 75.6 |
| **C.6** FSCA(12) | 0.455 | 0.396 | 63.2 | 72.6 |
| **D.1** Insertion Position [0] | 0.405 | 0.352 | 69.4 | 77.5 |
| **D.2** Insertion Position [0, 2] | 0.403 | 0.350 | 69.5 | 78.0 |
| **D.3** Insertion Position [0, 4] | 0.396 | 0.345 | **70.4** | **79.5** |
| **D.4** Insertion Position [0, 2, 4] | **0.394** | **0.342** | 69.7 | 78.5 |
| **D.5** Insertion Position [2, 4] | 0.417 | 0.353 | 68.7 | 76.5 |

**The Number of LLM Layers.** We conduct ablation studies to assess the impact of varying numbers of GPT-2 layers. **C.*** indicates that models with 4 and 6 layers perform optimally while adding more layers causes overfitting (Zhou et al., 2023). For computational efficiency, we select 4 layers (C.2).

**Insertion Position of Dual-Scale Context-Alignment GNNs.** DSCA-GNNs can seamlessly integrate into various layers of LLMs. Firstly, when priors for structural and logical alignment are present at the input, deeper layers of LLMs can more fully leverage this prior knowledge, hence, we default to employing this module at the model's input, as shown in configuration **D.1**. Secondly, **D.2**, **D.3**, and **D.4** involve repeating integration of this module at the mid and output stages of the

LLMs encoder, enhancing prior guidance and yielding performance improvements. Ablation studies for long-term forecasting and classification tasks reveal that optimal performance results from different insertion points, attributed to domain differences in token features among various LLMs layers (Ethayarajh, 2019) and varying demands of different tasks; Thirdly, configuration **D.5**, which omits GNNs module at model's input, leads to decreased performance, confirming that incorporating context-alignment at initial stage enhances the overall utilization of pre-trained LLMs.

## 5 CONCLUSION

In this paper, we point out that effectively utilizing LLMs in TS tasks requires first activating their capabilities, and then enhancing their performance. By rethinking the inherent strength of LLMs, we are the first to propose Context-Alignment paradigm, which aims to construct a context-level alignment between TS and language. Unlike previous methods based on token-level alignment, Context-Alignment constructs a consistent context for TS-language multimodal inputs, better harnessing LLMs' deep understanding of context to activate their potential on TS tasks. We develop DSCA-GNNs to achieve Context-Alignment. Besides, by integrating the Demonstration Examples Prompt technique, we introduce FSCA, which enhances LLMs' performance in TS tasks. Experiments demonstrate that our method significantly outperforms others, particularly in few-shot and zero-shot forecasting tasks, and ablation studies confirm the importance of Context-Alignment.

## 6 ACKNOWLEDGEMENT

This work was supported by the National Natural Science Foundation of China (Grant No. 62106116), the China Meteorological Administration (Grant No. QBZ202316), the National Key Research and Development Program of China (Grant No. 2024YFF1500600), and the Major Science and Technology Projects of Ningbo (Grant No. 2022Z236), as well as by the High Performance Computing Centers at Eastern Institute of Technology, Ningbo, and Ningbo Institute of Digital Twin.

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

# A EXPERIMENTAL DETAILS

## A.1 IMPLEMENTATION

All deep learning networks are implemented in PyTorch and trained on NVIDIA H800 80GB GPUs and GeForce RTX 4090 GPUs. We conduct our experiments using the pre-trained models from Wolf et al. (2020). To ensure fair comparisons, we adhere to the experimental configurations outlined in Wu et al. (2022) across all baselines and maintain a uniform evaluation procedure. As discussed in the ablation study in Sec. 4.7 (The Number of LLM Layers), we adopt the first 4 layers of GPT-2. For other LLM-based methods (Zhou et al., 2023; Jin et al., 2024; Pan et al., 2024), we strictly follow their provided experimental settings or cite their performance if applicable. Most of our proposed method's training settings are based on Zhou et al. (2023): For predictive tasks, we utilize FSCA with $N = 2$ in forms 5 and 6, providing one demonstration example. The Adam optimizer is used with decay rates $\beta = (0.9, 0.999)$ and initial learning rates from $\{10^{-4}, 5 \times 10^{-4}\}$. We implement a cosine annealing schedule with $T_{\max} = 20$ and $\eta_{\min} = 10^{-8}$, and set the batch size to 256. Early stopping is configured throughout the training process. MSE loss is employed for long-term forecasting, while SMAPE loss is used for short-term predictions. In classification tasks, as described in Sec. 4.6, the FSCA framework is applied to binary class datasets, including a demonstration example for each category. For multi-class datasets, such as handwriting recognition with 26 classes, we employ the VCA approach due to the input constraints of LLMs. The RAdam optimizer with initial learning rates from $\{10^{-2}, 10^{-3}\}$ and a batch size of 64 is used. Training also incorporates early stopping and employs cross-entropy loss.

## A.2 DATASET DETAILS

For the long-term forecasting task, we utilized eight widely used multivariate datasets (Wu et al., 2022), as detailed in Table 7. These include the Electricity Transformer Temperature (ETT) datasets (Zhou et al., 2021), as well as Illness, Weather, Electricity, and Traffic datasets. The ETT dataset comprises ETTh1 and ETTh2, while the ETTm dataset includes ETTm1 and ETTm2. Specifically, the ETT datasets contain power load data from two power stations at varying resolutions; the Weather dataset features 21 meteorological indicators from Germany; the ILI dataset captures weekly patient counts and influenza-like illness rates; the Electricity dataset consists of hourly consumption data from 321 customers; and the Traffic dataset records road occupancy from various sensors on San Francisco highways. Table 7 consolidates the feature details of these datasets, with the ETT dataset being applied in both few-shot and zero-shot learning tasks.

Table 7: Dataset details of long-term forecasting, wherein both ETTh and ETTm are concurrently utilized for few-shot and zero-shot learning.

| Dataset | Length | Dimension | Frequency |
|---|---|---|---|
| ETTh | 17420 | 7 | 1 hour |
| ETTm | 69680 | 7 | 15 min |
| Weather | 52696 | 22 | 10 min |
| ILI | 966 | 7 | 7 days |
| Electricity | 26304 | 321 | 1 hour |
| Traffic | 17544 | 862 | 1 hour |

For the short-term forecasting task, we employed the M4 benchmark dataset (Makridakis et al., 2018), which includes 10,000 time series across various domains, from business to economic forecasting, as shown in Table 8. The time series data is categorized into six groups, with sampling rates ranging from annually to hourly.

Table 8: Dataset details of short-term forecasting.

| Dataset | Length | Horizon | Frequency | Information |
|---|---|---|---|---|
| M4 Yearly | 23000 | 6 | Yearly | Demographic |
| M4 Quarterly | 24000 | 8 | Quarterly | Finance |
| M4 Monthly | 48000 | 18 | Monthly | Industry |
| M4 Weekly | 359 | 13 | Weekly | Macro |
| M4 Daily | 4227 | 14 | Daily | Macro |
| M4 Hourly | 414 | 48 | Hourly | Other |

For the time series classification task, we utilized 10 multivariate UEA datasets from Bagnall et al. (2018). Table 9 summarizes the number of classes, series lengths, feature dimensions, and sample sizes for training and testing.

Table 9: Dataset details of time series classification.

| Dataset | Train Cases | Test Cases | Dimensions | Length | Classes |
|---|---|---|---|---|---|
| EthanolConcentration | 261 | 263 | 3 | 1751 | 4 |
| FaceDetection | 5890 | 3524 | 144 | 62 | 2 |
| Handwriting | 150 | 850 | 3 | 152 | 26 |
| Heartbeat | 204 | 205 | 61 | 405 | 2 |
| JapaneseVowels | 270 | 370 | 12 | 29 | 9 |
| PEMS-SF | 267 | 173 | 963 | 144 | 7 |
| SelfRegulationSCP1 | 268 | 293 | 6 | 896 | 2 |
| SelfRegulationSCP2 | 200 | 180 | 7 | 1152 | 2 |
| SpokenArabicDigits | 6599 | 2199 | 13 | 93 | 10 |
| UWaveGestureLibrary | 120 | 320 | 3 | 315 | 8 |

## A.3 BASELINE DETAILS

**iTransformer** (Liu et al., 2023b): iTransformer repurposes the Transformer architecture by applying attention and feed-forward networks on inverted dimensions to improve multivariate time series forecasting, achieving state-of-the-art performance on real-world datasets.

**FEDformer** (Zhou et al., 2022): FEDformer combines Transformer models with seasonal-trend decomposition to capture both global trends and detailed structures in time series data.

**Stationary** (Liu et al., 2022): Stationary introduces Non-stationary Transformers, which include Series Stationarization for improved predictability and De-stationary Attention to restore intrinsic non-stationary information, resulting in significant performance improvements and a substantial reduction in mean squared error compared to mainstream Transformer models.

**ETSFormer** (Woo et al., 2022): ETSFormer is a novel Transformer architecture designed specifically for time-series forecasting, addressing the limitations of traditional models by incorporating exponential smoothing principles.

**PatchTST** (Nie et al., 2022): PatchTST introduces an efficient Transformer-based architecture for multivariate time series forecasting and self-supervised representation learning, utilizing a segmentation approach that divides time series into subseries-level patches. This design enhances local semantic retention, significantly reduces computation and memory usage of attention maps, and allows the model to consider longer historical data.

**TimesNet** (Wu et al., 2022): TimesNet is introduced as a novel approach for time series analysis that focuses on modeling temporal variations by transforming 1D time series into 2D tensors, thereby capturing complex intraperiod and interperiod variations.

**Dlinear** (Zeng et al., 2023): Dlinear challenges the efficacy of Transformer-based models for long-term time series forecasting (LTSF), highlighting their limitations in capturing temporal relationships due to the permutation-invariant nature of self-attention mechanisms.

**N-HiTs** (Challu et al., 2023): N-HiTs introduces a novel model for long-horizon forecasting that utilizes hierarchical interpolation and multi-rate data sampling techniques to effectively address the challenges of prediction volatility and computational complexity.

**N-BEATS** (Oreshkin et al., 2019): N-BEATS addresses the univariate time series point forecasting problem using a novel deep neural architecture featuring backward and forward residual links with a deep stack of fully connected layers. Importantly, the model's configuration without time-series-specific components suggests that deep learning primitives alone can effectively tackle a variety of forecasting challenges while also providing interpretable outputs with minimal accuracy loss.

**GPT4TS** (Zhou et al., 2023): GPT4TS presents a novel approach that leverages pre-trained LLMs for general time series analysis, addressing the challenge of limited training data by utilizing the Frozen Pretrained Transformer (FPT) architecture without altering the self-attention and feedforward layers.

**Time-LLM** (Jin et al., 2024): Time-LLM is a reprogramming framework designed to adapt LLMs for general time series forecasting by aligning time series data with natural language through input transformation and context enrichment techniques. Here, we use GPT-2 (Radford et al., 2019) as the base LLM.

**S$^2$IP-LLM** (Pan et al., 2024): S$^2$IP-LLM aligns pre-trained LLMs with time series embeddings to enhance forecasting performance. By designing a tokenization module for cross-modality alignment

and utilizing semantic anchors from pre-trained word embeddings, $S^2$IP-LLM effectively encodes temporal dynamics and retrieves relevant context for time series.

## A.4 EVALUATION METRICS

We use Mean Squared Error (MSE) and Mean Absolute Error (MAE) to evaluate the performance of long-term, few-shot, and zero-shot forecasting. For short-term forecasting on the M4 benchmark, following the approach in N-BEATS (Oreshkin et al., 2019), we adopt Symmetric Mean Absolute Percentage Error (SMAPE), Mean Absolute Scaled Error (MASE), and Overall Weighted Average (OWA). For time series classification tasks, as referenced in GPT4TS (Zhou et al., 2023), accuracy is used as the evaluation metric.

$$\text{MSE} = \frac{1}{H} \sum_{h=1}^{H} (Y_h - \hat{Y}_h)^2, \qquad\qquad \text{MAE} = \frac{1}{H} \sum_{h=1}^{H} |Y_h - \hat{Y}_h|,$$

$$\text{SMAPE} = \frac{200}{H} \sum_{h=1}^{H} \frac{|Y_h - \hat{Y}_h|}{|Y_h| + |\hat{Y}_h|}, \qquad\qquad \text{MAPE} = \frac{100}{H} \sum_{h=1}^{H} \frac{|Y_h - \hat{Y}_h|}{|Y_h|},$$

$$\text{MASE} = \frac{1}{H} \sum_{h=1}^{H} \frac{|Y_h - \hat{Y}_h|}{\frac{1}{H-s} \sum_{j=s+1}^{H} |Y_j - Y_{j-s}|}, \quad \text{OWA} = \frac{1}{2} \left[ \frac{\text{SMAPE}}{\text{SMAPE}_{\text{Naïve2}}} + \frac{\text{MASE}}{\text{MASE}_{\text{Naïve2}}} \right],$$

where $H$ denotes the number of data samples, corresponding to the forecasting horizon in the experiment. $s$ represents the periodicity of the time series. $Y_h$ and $\hat{Y}_h$ refer to the $h$-th ground truth and its corresponding prediction, respectively.

## B DUAL-SCALE CONTEXT-ALIGNMENT GNNs IN CLASSIFICATION

The construction of Dual-Scale Context-Alignment GNNs relies on the given prompt. In this section, we introduce the Dual-Scale Context-Alignment GNNs based on the vanilla prompt and few-shot prompting technique in classification tasks. Due to input length constraints, it is challenging to provide examples for many categories. Therefore, for multi-category classification tasks, we employ VCA approach. For binary classification tasks, we use FSCA and select one example from each category to serve as fixed input examples for both training and testing.

### B.1 VANILLA CONTEXT-ALIGNMENT (VCA)

We only replace the prompt content in Sec. 3.2 with "Predict category ($x$ in total) using previous data:", and the method for constructing the graph remains similar to Sec. 3.2, where $x$ denotes the number of categories.

### B.2 FEW-SHOT PROMPTING CONTEXT-ALIGNMENT (FSCA)

We replace the prompt content in Sec. 3.2 with "Predict category ($x$ in total) using previous data:", and arrange the input embeddings as the format 10, where every element belongs to an $M$-dimensional space.

$$[e_1^{(1)},\dots,e_n^{(1)}, z_1^{(1)},\dots,z_m^{(1)}, y^{(1)},\dots,e_1^{(l)},\dots,e_n^{(l)}, z_1^{(l)},\dots,z_m^{(l)}, y^{(l)}, e_1^{(l+1)},\dots,e_n^{(l+1)}, z_1^{(l+1)},\dots,z_m^{(l+1)}]. \tag{10}$$

$\{z_1^{(k)},\dots,z_m^{(k)}\} = \{z_1^{(l+1)},\dots,z_m^{(l+1)}\} = \{z_1,\dots,z_m\}$ are the token embeddings of the prompt. We utilise $\{e_1^{(k)},\dots,e_n^{(k)}\}$ as fixed examples during training and testing phases, where $\{e_1^{(k)},\dots,e_n^{(k)}\}$ is from the $k$-th categories and $k = 1,\dots,l$. $y^{(k)}$ is the embedding of the correct label for $\{e_1^{(k)},\dots,e_n^{(k)}\}$. $\{e_1^{(l+1)},\dots,e_n^{(l+1)}\}$ is the TS that needs to be classified. In the fine-grained GNN $G_F$, each element in the form 10 is treated as a node. Similar to Sec. 3.2 and Sec. 3.3, two learnable linear layers $f_e$, $f_z$ map form 10 to form 11. Every element in the form 11 belongs to

an $M$-dimensional space.

$$[\tilde{e}^{(1)}, \tilde{z}^{(1)}, y^{(1)}, \ldots, \tilde{e}^{(l)}, \tilde{z}^{(l)}, y^{(l)}, \tilde{e}^{(l+1)}, \tilde{z}^{(l+1)}]. \tag{11}$$

For the coarse-grained GNN $G_C$, we construct directed edges based on logical relationships as described in formula 12. The first term indicates that the TS data provides information for the prompt, while the second term implies that $y^{(k)}$ is the output result of the prompt.

$$\{E_C : \tilde{e}^{(k)} \to \tilde{z}^{(k)} | k = 1, \ldots, l+1\} \cup \{E_C : \tilde{z}^{(k)} \to y^{(k)} | k = 1, \ldots, l\}. \tag{12}$$

The directed edges of $G_F$ are decompositions of the directed edges of $G_C$, which can be represented by the formula 13.

$$\{E_F : e_i^{(k)} \to z_j^{(k)} | i = 1, \ldots, n, j = 1, \ldots, m, k = 1, \ldots, l+1\}$$
$$\cup \{E_F : z_j^{(k)} \to y^{(k)} | j = 1, \ldots, m, k = 1, \ldots, l\}. \tag{13}$$

Since LLMs have strong comprehension abilities for language modality prompts, we can prune the directed edges in $G_F$, transforming form 13 into form 14, i.e., TS tokens and label embedding $y$ are only connected to the first and last tokens of the prompt, thereby prevent overfitting. We constrain $\sum_{i=1}^{n} w_i^{(k)} = 1$ for the first type edges, $w^{(k)} = 1$ for the second type edges. $\{w_i^{(k)}\}_{i=1}^{n}$ are proportional to the cosine similarity between node embeddings. The updated node embedding matrices for $G_F$ and $G_C$ denoted as $\hat{N}_F$ and $\hat{N}_C$, respectively, which is similar to formula 3.

$$\{E_F : e_i^{(k)} \to z_1^{(k)} | i = 1, \ldots, n, k = 1, \ldots, l+1\} \cup \{E_F : z_m^{(k)} \to y^{(k)} | k = 1, \ldots, l\}. \tag{14}$$

Similar to the demo in Sec. 3.2, we introduce an assignment matrix and a learnable weight matrix to achieve learnable interactions between the two scales. Both $\hat{N}_F$ and $\hat{N}_C$ are input into pre-trained LLMs. The Dual-Scale Context-Alignment GNNs can be flexibly integrated into various layers of pre-trained LLMs as depicted in Fig. 1, and only the first time apply it needs $f_e$ and $f_z$ to obtain coarse-grained GNN. The output from the $G_F$ branch is used to compute the cross entropy loss against the ground truth.

# C  FULL RESULTS

## C.1  LONG-TERM FORECASTING FULL RESULTS

Table 10 presents the detailed long-term forecasting results across four prediction horizons. Compared to other models, including LLM-based methods, Transformer-based models, and other high-performing approaches, FSCA demonstrates strong and relatively stable performance across various datasets. It achieves an average 3.1% MSE reduction compared to the second-best method, PatchTST, and outperforms LLM-based methods (S$^2$IP-LLM, Time-LLM, and GPT4TS) by 7.3%, 12.2%, and 16.6%, respectively. We attribute this consistent advantage to the successful Context-Alignment, which effectively guides LLMs' deep understanding of time series data, with demonstration examples prompt serving as a key factor in enhancement.

## C.2  SHORT-TERM FORECASTING FULL RESULTS

We present the comprehensive short-term forecasting results in Table 11. Under different frequency settings, FSCA consistently outperforms most baseline models.

## C.3  FEW-SHOT FORECASTING FULL RESULTS

Table 12 presents the detailed results of the few-shot forecasting trained on 5% data across different prediction lengths. Except for DLinear's strong performance on the ETTm1 dataset, FSCA demonstrates significant enhancements in most scenarios across various datasets, outperforming the second-best model, S$^2$IP-LLM, by 6.7%.

Table 13 presents the detailed results of the few-shot forecasting task using 10% of the training data. FSCA demonstrates superior performance, outperforming the baseline across nearly all settings, with an average MSE reduction of 7.8% over the next best model, GPT4TS.

Table 10: Full results of long-term forecasting tasks, all results are based on different prediction horizons: 24, 36, 48, 60 for ILI and 96, 192, 336, 720 for others. **Bold**: best, Underline: second best.

| Methods | | FSCA | | $S^2$IP-LLM | | Time-LLM | | GPT4TS | | iTransformer | | DLinear | | PatchTST | | TimesNet | | FEDformer | | Stationary | | ETSformer | |
|---|---|---|---|---|---|---|---|---|---|---|---|---|---|---|---|---|---|---|---|---|---|---|---|
| Metric | | MSE | MAE | MSE | MAE | MSE | MAE | MSE | MAE | MSE | MAE | MSE | MAE | MSE | MAE | MSE | MAE | MSE | MAE | MSE | MAE | MSE | MAE |
| ILI | 24 | **1.206** | **0.728** | 1.467 | 0.778 | 1.622 | 0.806 | 2.063 | 0.881 | 1.694 | 0.874 | 2.215 | 1.081 | 1.319 | 0.754 | 2.317 | 0.934 | 3.228 | 1.260 | 2.294 | 0.945 | 2.527 | 1.020 |
| | 36 | **1.251** | **0.750** | 1.534 | 0.841 | 1.695 | 0.857 | 1.868 | 0.892 | 2.229 | 0.983 | 1.963 | 0.963 | 1.430 | 0.834 | 1.972 | 0.920 | 2.679 | 1.080 | 1.825 | 0.848 | 2.615 | 1.007 |
| | 48 | 1.566 | 0.818 | 1.608 | 0.836 | 1.654 | 0.863 | 1.790 | 0.884 | 2.382 | 0.995 | 2.130 | 1.024 | **1.553** | **0.815** | 2.238 | 0.940 | 2.622 | 1.078 | 2.010 | 0.900 | 2.359 | 0.972 |
| | 60 | 1.495 | 0.835 | 1.597 | 0.849 | 1.880 | 0.905 | 1.979 | 0.957 | 1.988 | 0.913 | 2.368 | 1.096 | **1.470** | **0.788** | 2.027 | 0.928 | 2.857 | 1.157 | 2.178 | 0.963 | 2.487 | 1.016 |
| | Avg | **1.380** | **0.783** | 1.552 | 0.826 | 1.713 | 0.858 | 1.925 | 0.903 | 2.073 | 0.941 | 2.169 | 1.041 | 1.443 | 0.797 | 2.139 | 0.931 | 2.847 | 1.144 | 2.077 | 0.914 | 2.497 | 1.004 |
| Weather | 96 | **0.146** | **0.196** | 0.149 | 0.200 | 0.163 | 0.210 | 0.162 | 0.212 | 0.253 | 0.304 | 0.176 | 0.237 | 0.149 | 0.198 | 0.172 | 0.220 | 0.217 | 0.296 | 0.173 | 0.223 | 0.197 | 0.281 |
| | 192 | **0.193** | **0.241** | 0.195 | 0.244 | 0.205 | 0.245 | 0.204 | 0.248 | 0.280 | 0.319 | 0.220 | 0.282 | 0.194 | **0.241** | 0.219 | 0.261 | 0.276 | 0.336 | 0.245 | 0.285 | 0.237 | 0.312 |
| | 336 | **0.244** | **0.279** | 0.246 | 0.280 | 0.257 | 0.287 | 0.254 | 0.286 | 0.321 | 0.344 | 0.265 | 0.319 | 0.245 | 0.282 | 0.280 | 0.306 | 0.339 | 0.380 | 0.321 | 0.338 | 0.298 | 0.353 |
| | 720 | **0.314** | 0.333 | 0.320 | 0.336 | 0.323 | 0.332 | 0.326 | 0.337 | 0.364 | 0.374 | 0.333 | 0.362 | **0.314** | 0.334 | 0.365 | 0.359 | 0.403 | 0.428 | 0.414 | 0.410 | 0.352 | **0.288** |
| | Avg | **0.224** | **0.262** | 0.228 | 0.265 | 0.237 | 0.269 | 0.237 | 0.270 | 0.304 | 0.335 | 0.248 | 0.300 | 0.225 | 0.264 | 0.259 | 0.287 | 0.309 | 0.360 | 0.288 | 0.314 | 0.271 | 0.334 |
| ECL | 96 | **0.128** | **0.222** | 0.138 | 0.234 | 0.140 | 0.236 | 0.139 | 0.238 | 0.147 | 0.248 | 0.140 | 0.237 | 0.129 | **0.222** | 0.168 | 0.272 | 0.193 | 0.308 | 0.169 | 0.273 | 0.187 | 0.304 |
| | 192 | **0.146** | **0.239** | 0.153 | 0.252 | 0.150 | 0.249 | 0.153 | 0.251 | 0.165 | 0.267 | 0.153 | 0.249 | 0.157 | 0.240 | 0.184 | 0.289 | 0.201 | 0.315 | 0.182 | 0.286 | 0.199 | 0.315 |
| | 336 | **0.163** | **0.258** | 0.169 | 0.270 | 0.168 | 0.267 | 0.169 | 0.266 | 0.178 | 0.279 | 0.169 | 0.267 | **0.163** | 0.259 | 0.198 | 0.300 | 0.214 | 0.329 | 0.200 | 0.304 | 0.212 | 0.329 |
| | 720 | 0.199 | **0.287** | 0.204 | 0.293 | 0.209 | 0.302 | 0.206 | 0.297 | 0.322 | 0.398 | 0.203 | 0.301 | **0.197** | 0.290 | 0.220 | 0.320 | 0.246 | 0.355 | 0.222 | 0.321 | 0.233 | 0.345 |
| | Avg | **0.159** | **0.252** | 0.166 | 0.262 | 0.167 | 0.264 | 0.167 | 0.263 | 0.203 | 0.298 | 0.166 | 0.263 | 0.161 | **0.252** | 0.192 | 0.295 | 0.214 | 0.327 | 0.193 | 0.296 | 0.208 | 0.323 |
| Traffic | 96 | **0.355** | **0.246** | 0.379 | 0.274 | 0.384 | 0.278 | 0.388 | 0.282 | 0.367 | 0.288 | 0.410 | 0.282 | 0.360 | 0.249 | 0.593 | 0.321 | 0.587 | 0.366 | 0.612 | 0.338 | 0.607 | 0.392 |
| | 192 | **0.377** | **0.255** | 0.397 | 0.282 | 0.398 | 0.286 | 0.407 | 0.290 | 0.378 | 0.293 | 0.423 | 0.287 | 0.379 | 0.256 | 0.617 | 0.336 | 0.604 | 0.373 | 0.613 | 0.340 | 0.621 | 0.399 |
| | 336 | **0.387** | 0.265 | 0.407 | 0.289 | 0.408 | 0.289 | 0.412 | 0.294 | 0.389 | 0.294 | 0.436 | 0.296 | 0.392 | **0.264** | 0.629 | 0.336 | 0.621 | 0.383 | 0.618 | 0.328 | 0.622 | 0.396 |
| | 720 | 0.425 | 0.287 | 0.440 | 0.301 | 0.436 | 0.303 | 0.450 | 0.312 | **0.401** | 0.304 | 0.466 | 0.315 | 0.432 | **0.286** | 0.640 | 0.350 | 0.626 | 0.382 | 0.653 | 0.355 | 0.632 | 0.396 |
| | Avg | **0.386** | **0.263** | 0.405 | 0.286 | 0.407 | 0.289 | 0.414 | 0.294 | 0.389 | 0.295 | 0.433 | 0.295 | 0.390 | **0.263** | 0.620 | 0.336 | 0.610 | 0.376 | 0.624 | 0.340 | 0.621 | 0.396 |
| ETTh1 | 96 | **0.349** | **0.389** | 0.367 | 0.398 | 0.383 | 0.404 | 0.376 | 0.397 | 0.395 | 0.420 | 0.375 | 0.399 | 0.370 | 0.399 | 0.384 | 0.402 | 0.376 | 0.419 | 0.513 | 0.491 | 0.494 | 0.479 |
| | 192 | **0.390** | **0.415** | 0.402 | 0.422 | 0.427 | 0.431 | 0.416 | 0.418 | 0.427 | 0.441 | 0.405 | 0.416 | 0.413 | 0.421 | 0.436 | 0.429 | 0.423 | 0.448 | 0.534 | 0.504 | 0.538 | 0.504 |
| | 336 | **0.402** | **0.432** | 0.432 | 0.451 | 0.430 | 0.436 | 0.442 | 0.433 | 0.445 | 0.457 | 0.439 | 0.443 | 0.422 | 0.436 | 0.491 | 0.469 | 0.459 | 0.465 | 0.588 | 0.535 | 0.574 | 0.521 |
| | 720 | **0.433** | 0.460 | 0.472 | 0.474 | 0.465 | 0.469 | 0.477 | **0.456** | 0.537 | 0.530 | 0.472 | 0.490 | 0.447 | 0.466 | 0.521 | 0.500 | 0.506 | 0.507 | 0.643 | 0.616 | 0.562 | 0.535 |
| | Avg | **0.394** | **0.424** | 0.418 | 0.436 | 0.426 | 0.435 | 0.427 | 0.426 | 0.451 | 0.462 | 0.422 | 0.437 | 0.413 | 0.430 | 0.458 | 0.450 | 0.440 | 0.460 | 0.570 | 0.537 | 0.542 | 0.510 |
| ETTh2 | 96 | **0.256** | **0.328** | 0.284 | 0.345 | 0.293 | 0.348 | 0.285 | 0.342 | 0.304 | 0.360 | 0.289 | 0.353 | 0.274 | 0.336 | 0.340 | 0.374 | 0.358 | 0.397 | 0.476 | 0.458 | 0.340 | 0.391 |
| | 192 | **0.311** | **0.372** | 0.349 | 0.387 | 0.356 | 0.391 | 0.354 | 0.389 | 0.377 | 0.403 | 0.383 | 0.418 | 0.339 | 0.379 | 0.402 | 0.414 | 0.429 | 0.439 | 0.512 | 0.493 | 0.430 | 0.439 |
| | 336 | **0.308** | **0.372** | 0.368 | 0.417 | 0.372 | 0.408 | 0.373 | 0.407 | 0.405 | 0.429 | 0.448 | 0.465 | 0.329 | 0.380 | 0.452 | 0.452 | 0.496 | 0.487 | 0.552 | 0.551 | 0.485 | 0.479 |
| | 720 | 0.390 | 0.428 | 0.419 | 0.445 | 0.421 | 0.446 | 0.406 | 0.441 | 0.443 | 0.464 | 0.605 | 0.551 | **0.379** | **0.422** | 0.462 | 0.468 | 0.463 | 0.474 | 0.562 | 0.560 | 0.500 | 0.497 |
| | Avg | **0.316** | **0.375** | 0.355 | 0.399 | 0.361 | 0.398 | 0.354 | 0.394 | 0.382 | 0.414 | 0.431 | 0.446 | 0.330 | 0.379 | 0.414 | 0.427 | 0.437 | 0.449 | 0.526 | 0.516 | 0.439 | 0.452 |
| ETTm1 | 96 | **0.282** | 0.343 | 0.291 | 0.348 | 0.294 | 0.345 | 0.292 | 0.346 | 0.312 | 0.366 | 0.299 | 0.343 | 0.290 | **0.342** | 0.338 | 0.375 | 0.379 | 0.419 | 0.386 | 0.398 | 0.375 | 0.398 |
| | 192 | 0.324 | 0.369 | **0.323** | **0.368** | 0.330 | 0.368 | 0.332 | 0.372 | 0.347 | 0.385 | 0.335 | 0.365 | 0.332 | 0.369 | 0.374 | 0.387 | 0.426 | 0.441 | 0.459 | 0.444 | 0.408 | 0.410 |
| | 336 | **0.356** | **0.386** | 0.361 | 0.392 | 0.365 | 0.392 | 0.366 | 0.394 | 0.379 | 0.404 | 0.369 | **0.386** | 0.366 | 0.392 | 0.410 | 0.411 | 0.445 | 0.459 | 0.495 | 0.464 | 0.435 | 0.428 |
| | 720 | **0.405** | **0.417** | 0.410 | 0.420 | 0.427 | 0.431 | 0.417 | 0.421 | 0.441 | 0.442 | 0.425 | 0.421 | 0.416 | 0.420 | 0.478 | 0.450 | 0.543 | 0.490 | 0.585 | 0.516 | 0.499 | 0.462 |
| | Avg | 0.342 | 0.378 | 0.346 | 0.382 | 0.354 | 0.384 | 0.352 | 0.383 | 0.370 | 0.399 | 0.357 | **0.378** | 0.351 | 0.380 | 0.400 | 0.406 | 0.448 | 0.452 | 0.481 | 0.456 | 0.429 | 0.425 |
| ETTm2 | 96 | **0.164** | **0.254** | 0.167 | 0.257 | 0.175 | 0.265 | 0.173 | 0.262 | 0.179 | 0.271 | 0.167 | 0.269 | 0.165 | 0.255 | 0.187 | 0.267 | 0.203 | 0.287 | 0.192 | 0.274 | 0.189 | 0.280 |
| | 192 | 0.222 | **0.296** | 0.227 | 0.303 | 0.243 | 0.316 | 0.229 | 0.301 | 0.242 | 0.313 | 0.224 | 0.303 | **0.220** | 0.292 | 0.249 | 0.309 | 0.269 | 0.328 | 0.280 | 0.339 | 0.253 | 0.319 |
| | 336 | **0.269** | **0.326** | 0.285 | 0.346 | 0.294 | 0.343 | 0.286 | 0.341 | 0.288 | 0.344 | 0.281 | 0.342 | 0.274 | 0.329 | 0.321 | 0.351 | 0.325 | 0.366 | 0.334 | 0.361 | 0.314 | 0.357 |
| | 720 | **0.346** | **0.381** | 0.368 | 0.398 | 0.389 | 0.410 | 0.378 | 0.401 | 0.378 | 0.397 | 0.397 | 0.421 | 0.362 | 0.385 | 0.408 | 0.403 | 0.421 | 0.415 | 0.417 | 0.413 | 0.414 | 0.413 |
| | Avg | **0.250** | **0.314** | 0.262 | 0.326 | 0.275 | 0.334 | 0.266 | 0.326 | 0.272 | 0.331 | 0.267 | 0.333 | 0.255 | 0.315 | 0.291 | 0.333 | 0.305 | 0.349 | 0.306 | 0.347 | 0.293 | 0.342 |

Table 11: Full results of short-term time series forecasting on M4, with prediction horizons ranging from [6, 48]. The last three rows are weighted averages across all datasets under different sampling intervals. **Bold**: best, Underline: second best.

| Methods | | FSCA | $S^2$IP-LLM | Time-LLM | GPT4TS | iTransformer | DLinear | PatchTST | N-HiTS | N-BEATS | TimesNet | FEDformer | Stationary |
|---|---|---|---|---|---|---|---|---|---|---|---|---|---|
| Year. | SMAPE | **13.288** | 13.413 | 13.750 | 15.110 | 13.652 | 16.965 | 13.477 | 13.422 | 13.487 | 15.378 | 14.021 | 14.727 |
| | MASE | **2.974** | 3.024 | 3.055 | 3.565 | 3.095 | 4.283 | 3.019 | 3.056 | 3.036 | 3.554 | 3.036 | 3.078 |
| | OWA | **0.781** | 0.792 | 0.805 | 0.911 | 0.807 | 1.058 | 0.792 | 0.795 | 0.795 | 0.918 | 0.811 | 0.807 |
| Quart. | SMAPE | **10.037** | 10.352 | 10.671 | 10.597 | 10.353 | 12.145 | 10.380 | 10.185 | 10.564 | 10.465 | 11.100 | 10.958 |
| | MASE | **1.174** | 1.228 | 1.276 | 1.253 | 1.209 | 1.520 | 1.233 | 1.180 | 1.252 | 1.227 | 1.350 | 1.325 |
| | OWA | **0.884** | 0.922 | 0.950 | 0.938 | 0.911 | 1.106 | 0.921 | 0.893 | 0.936 | 0.923 | 0.996 | 0.981 |
| Month. | SMAPE | **12.762** | 12.995 | 13.416 | 13.258 | 13.079 | 13.514 | 12.959 | 13.059 | 13.089 | 13.513 | 14.403 | 13.917 |
| | MASE | **0.947** | 0.970 | 1.045 | 1.003 | 0.974 | 1.037 | 0.970 | 1.013 | 0.996 | 1.039 | 1.147 | 1.097 |
| | OWA | **0.897** | 0.910 | 0.957 | 0.931 | 0.911 | 0.956 | 0.905 | 0.929 | 0.922 | 0.957 | 1.038 | 0.998 |
| Others. | SMAPE | 4.761 | 4.805 | 4.973 | 6.124 | 4.78 | 6.709 | 4.952 | **4.711** | 6.599 | 6.913 | 7.148 | 6.302 |
| | MASE | 3.207 | 3.247 | 3.412 | 4.116 | 3.231 | 4.953 | 3.347 | **3.054** | 4.430 | 4.507 | 4.064 | 4.064 |
| | OWA | 1.007 | 1.017 | 1.053 | 1.259 | 1.012 | 1.487 | 1.049 | **0.977** | 1.393 | 1.438 | 1.304 | 1.304 |
| Avg. | SMAPE | **11.828** | 12.021 | 12.494 | 12.690 | 12.142 | 13.639 | 12.059 | 12.035 | 12.250 | 12.880 | 13.160 | 12.780 |
| | MASE | **1.580** | 1.612 | 1.731 | 1.808 | 1.631 | 2.095 | 1.623 | 1.625 | 1.698 | 1.836 | 1.775 | 1.756 |
| | OWA | **0.850** | 0.857 | 0.913 | 0.940 | 0.874 | 1.051 | 0.869 | 0.869 | 0.896 | 0.955 | 0.949 | 0.930 |

## C.4 ZERO-SHOT FORECASTING FULL RESULTS

Table 14 provides detailed results of the zero-shot forecasting task across different prediction lengths. FSCA consistently achieves optimal performance across all settings, with a significant margin. Compared to the second-best method, PatchTST, it shows an average improvement of 13.3%. Additionally, FSCA achieves MSE reductions of 18.3%, 17.7%, and 24.3% over other LLM-

Table 12: Full results of few-shot learning on 5% training data. All results are from four different prediction horizons {96, 192, 336, 720}. A lower MSE indicates better performance. **Bold**: best, Underline: second best. '-' means that 5% time series is not sufficient to constitute a training set.

| Methods | | FSCA | | S²IP-LLM | | Time-LLM | | GPT4TS | | iTransformer | | DLinear | | PatchTST | | TimesNet | | FEDformer | | Stationary | | ETSformer | |
|---|---|---|---|---|---|---|---|---|---|---|---|---|---|---|---|---|---|---|---|---|---|---|---|
| Metric | | MSE | MAE | MSE | MAE | MSE | MAE | MSE | MAE | MSE | MAE | MSE | MAE | MSE | MAE | MSE | MAE | MSE | MAE | MSE | MAE | MSE | MAE |
| ETTh1 | 96 | **0.482** | **0.457** | 0.500 | 0.493 | 0.518 | 0.498 | 0.543 | 0.506 | 0.808 | 0.610 | 0.547 | 0.503 | 0.557 | 0.519 | 0.892 | 0.625 | 0.593 | 0.529 | 0.952 | 0.650 | 1.169 | 0.832 |
| | 192 | **0.537** | **0.492** | 0.690 | 0.539 | 0.702 | 0.547 | 0.748 | 0.580 | 0.928 | 0.658 | 0.720 | 0.604 | 0.711 | 0.570 | 0.940 | 0.665 | 0.652 | 0.563 | 0.943 | 0.645 | 1.221 | 0.853 |
| | 336 | **0.707** | **0.574** | 0.761 | 0.620 | 0.725 | 0.603 | 0.754 | 0.595 | 1.475 | 0.861 | 0.984 | 0.727 | 0.816 | 0.619 | 0.945 | 0.653 | 0.731 | 0.594 | 0.935 | 0.644 | 1.179 | 0.832 |
| | 720 | - | - | - | - | - | - | - | - | - | - | - | - | - | - | - | - | - | - | - | - | - | - |
| | Avg | **0.575** | **0.508** | 0.650 | 0.550 | 0.648 | 0.549 | 0.681 | 0.560 | 1.070 | 0.710 | 0.750 | 0.611 | 0.695 | 0.569 | 0.925 | 0.647 | 0.658 | 0.562 | 0.943 | 0.646 | 1.189 | 0.839 |
| ETTh2 | 96 | **0.312** | **0.352** | 0.363 | 0.409 | 0.384 | 0.420 | 0.376 | 0.421 | 0.397 | 0.427 | 0.442 | 0.456 | 0.401 | 0.421 | 0.409 | 0.420 | 0.390 | 0.424 | 0.408 | 0.423 | 0.678 | 0.619 |
| | 192 | 0.389 | **0.409** | 0.375 | 0.411 | 0.394 | 0.424 | 0.418 | 0.441 | 0.438 | 0.445 | 0.617 | 0.542 | 0.452 | 0.455 | 0.483 | 0.464 | 0.457 | 0.465 | 0.497 | 0.468 | 0.845 | 0.697 |
| | 336 | **0.397** | 0.430 | 0.403 | **0.421** | 0.416 | 0.433 | 0.408 | 0.439 | 0.631 | 0.553 | 1.424 | 0.849 | 0.464 | 0.469 | 0.499 | 0.479 | 0.477 | 0.483 | 0.507 | 0.481 | 0.905 | 0.727 |
| | 720 | - | - | - | - | - | - | - | - | - | - | - | - | - | - | - | - | - | - | - | - | - | - |
| | Avg | **0.366** | **0.397** | 0.380 | 0.413 | 0.398 | 0.426 | 0.400 | 0.433 | 0.488 | 0.475 | 0.827 | 0.615 | 0.439 | 0.448 | 0.463 | 0.454 | 0.463 | 0.454 | 0.470 | 0.489 | 0.809 | 0.681 |
| ETTm1 | 96 | 0.355 | 0.383 | 0.357 | 0.390 | 0.422 | 0.424 | 0.386 | 0.405 | 0.589 | 0.510 | **0.332** | **0.374** | 0.399 | 0.414 | 0.606 | 0.518 | 0.628 | 0.544 | 0.823 | 0.587 | 1.031 | 0.747 |
| | 192 | 0.397 | 0.405 | 0.432 | 0.434 | 0.448 | 0.440 | 0.440 | 0.438 | 0.703 | 0.565 | **0.358** | **0.390** | 0.441 | 0.436 | 0.681 | 0.539 | 0.666 | 0.566 | 0.844 | 0.591 | 1.087 | 0.766 |
| | 336 | 0.450 | 0.440 | 0.440 | 0.442 | 0.452 | 0.447 | 0.485 | 0.459 | 0.898 | 0.641 | **0.402** | **0.416** | 0.499 | 0.467 | 0.786 | 0.597 | 0.807 | 0.628 | 0.870 | 0.603 | 1.138 | 0.787 |
| | 720 | 0.538 | 0.486 | 0.593 | 0.521 | 0.585 | 0.491 | 0.577 | 0.499 | 0.948 | 0.671 | **0.511** | **0.489** | 0.767 | 0.587 | 0.796 | 0.593 | 0.822 | 0.633 | 0.893 | 0.611 | 1.245 | 0.831 |
| | Avg | 0.435 | 0.429 | 0.455 | 0.446 | 0.477 | 0.451 | 0.472 | 0.450 | 0.784 | 0.596 | **0.400** | **0.417** | 0.526 | 0.476 | 0.717 | 0.561 | 0.730 | 0.592 | 0.857 | 0.598 | 1.125 | 0.782 |
| ETTm2 | 96 | **0.189** | **0.274** | 0.197 | 0.278 | 0.205 | 0.277 | 0.199 | 0.280 | 0.265 | 0.339 | 0.236 | 0.326 | 0.206 | 0.288 | 0.220 | 0.299 | 0.229 | 0.320 | 0.238 | 0.316 | 0.404 | 0.485 |
| | 192 | **0.250** | **0.311** | 0.254 | 0.322 | 0.267 | 0.336 | 0.256 | 0.316 | 0.310 | 0.362 | 0.306 | 0.373 | 0.264 | 0.324 | 0.311 | 0.361 | 0.394 | 0.361 | 0.298 | 0.349 | 0.479 | 0.521 |
| | 336 | **0.298** | **0.341** | 0.315 | 0.350 | 0.309 | 0.347 | 0.318 | 0.353 | 0.373 | 0.399 | 0.380 | 0.423 | 0.334 | 0.367 | 0.338 | 0.366 | 0.378 | 0.427 | 0.353 | 0.380 | 0.552 | 0.555 |
| | 720 | **0.399** | **0.403** | 0.421 | 0.421 | 0.448 | 0.432 | 0.460 | 0.436 | 0.478 | 0.454 | 0.674 | 0.583 | 0.454 | 0.432 | 0.509 | 0.465 | 0.523 | 0.510 | 0.475 | 0.445 | 0.701 | 0.627 |
| | Avg | **0.284** | **0.332** | 0.296 | 0.342 | 0.307 | 0.348 | 0.308 | 0.346 | 0.356 | 0.388 | 0.399 | 0.426 | 0.314 | 0.352 | 0.344 | 0.372 | 0.381 | 0.404 | 0.341 | 0.372 | 0.534 | 0.547 |

Table 13: Full results of few-shot learning on 10% training data. All results are from four different prediction horizons {96, 192, 336, 720}. A lower MSE indicates better performance. **Bold**: best, Underline: second best.

| Methods | | FSCA | | S²IP-LLM | | Time-LLM | | GPT4TS | | iTransformer | | DLinear | | PatchTST | | TimesNet | | FEDformer | | Stationary | | ETSformer | |
|---|---|---|---|---|---|---|---|---|---|---|---|---|---|---|---|---|---|---|---|---|---|---|---|
| Metric | | MSE | MAE | MSE | MAE | MSE | MAE | MSE | MAE | MSE | MAE | MSE | MAE | MSE | MAE | MSE | MAE | MSE | MAE | MSE | MAE | MSE | MAE |
| ETTh1 | 96 | **0.449** | **0.448** | 0.481 | 0.474 | 0.720 | 0.533 | 0.458 | 0.456 | 0.790 | 0.586 | 0.492 | 0.495 | 0.516 | 0.485 | 0.861 | 0.628 | 0.512 | 0.499 | 0.918 | 0.639 | 1.112 | 0.806 |
| | 192 | **0.491** | **0.469** | 0.518 | 0.491 | 0.747 | 0.545 | 0.570 | 0.516 | 0.837 | 0.609 | 0.565 | 0.538 | 0.598 | 0.524 | 0.797 | 0.593 | 0.624 | 0.555 | 0.915 | 0.629 | 1.155 | 0.823 |
| | 336 | **0.549** | **0.499** | 0.664 | 0.570 | 0.793 | 0.551 | 0.608 | 0.535 | 0.780 | 0.615 | 0.721 | 0.622 | 0.657 | 0.550 | 0.941 | 0.648 | 0.691 | 0.574 | 0.939 | 0.644 | 1.179 | 0.832 |
| | 720 | **0.661** | **0.559** | 0.711 | 0.584 | 0.880 | 0.584 | 0.725 | 0.591 | 1.234 | 0.811 | 0.986 | 0.743 | 0.762 | 0.610 | 0.877 | 0.641 | 0.728 | 0.614 | 0.887 | 0.645 | 1.273 | 0.874 |
| | Avg | **0.538** | **0.494** | 0.593 | 0.529 | 0.785 | 0.553 | 0.590 | 0.525 | 0.910 | 0.860 | 0.691 | 0.600 | 0.633 | 0.542 | 0.869 | 0.628 | 0.639 | 0.561 | 0.915 | 0.639 | 1.180 | 0.834 |
| ETTh2 | 96 | **0.287** | **0.351** | 0.354 | 0.400 | 0.334 | 0.381 | 0.331 | 0.374 | 0.404 | 0.435 | 0.357 | 0.411 | 0.353 | 0.389 | 0.378 | 0.409 | 0.382 | 0.416 | 0.389 | 0.411 | 0.678 | 0.619 |
| | 192 | **0.351** | **0.392** | 0.401 | 0.423 | 0.430 | 0.438 | 0.402 | 0.411 | 0.470 | 0.474 | 0.569 | 0.519 | 0.403 | 0.414 | 0.490 | 0.467 | 0.478 | 0.474 | 0.473 | 0.455 | 0.785 | 0.666 |
| | 336 | **0.386** | **0.420** | 0.442 | 0.450 | 0.449 | 0.458 | 0.406 | 0.433 | 0.489 | 0.485 | 0.671 | 0.572 | 0.426 | 0.441 | 0.537 | 0.494 | 0.504 | 0.501 | 0.507 | 0.480 | 0.839 | 0.694 |
| | 720 | **0.426** | **0.447** | 0.480 | 0.486 | 0.485 | 0.490 | 0.449 | 0.464 | 0.593 | 0.538 | 0.824 | 0.648 | 0.477 | 0.480 | 0.510 | 0.491 | 0.499 | 0.509 | 0.477 | 0.472 | 1.273 | 0.874 |
| | Avg | **0.363** | **0.403** | 0.419 | 0.439 | 0.424 | 0.441 | 0.397 | 0.421 | 0.489 | 0.483 | 0.605 | 0.538 | 0.415 | 0.431 | 0.479 | 0.465 | 0.466 | 0.475 | 0.462 | 0.455 | 0.894 | 0.713 |
| ETTm1 | 96 | 0.371 | 0.393 | 0.388 | 0.401 | 0.412 | 0.422 | 0.390 | 0.404 | 0.709 | 0.556 | **0.352** | **0.392** | 0.410 | 0.419 | 0.583 | 0.501 | 0.578 | 0.518 | 0.761 | 0.568 | 0.911 | 0.688 |
| | 192 | 0.405 | **0.407** | 0.422 | 0.421 | 0.447 | 0.438 | 0.429 | 0.423 | 0.717 | 0.548 | **0.382** | 0.412 | 0.437 | 0.434 | 0.630 | 0.528 | 0.617 | 0.546 | 0.781 | 0.574 | 0.955 | 0.703 |
| | 336 | 0.444 | 0.424 | 0.456 | 0.430 | 0.497 | 0.465 | 0.469 | 0.439 | 0.735 | 0.575 | **0.419** | 0.434 | 0.476 | 0.454 | 0.725 | 0.568 | 0.998 | 0.775 | 0.803 | 0.587 | 0.991 | 0.719 |
| | 720 | 0.520 | 0.468 | 0.554 | 0.490 | 0.594 | 0.521 | 0.569 | 0.498 | 0.752 | 0.584 | **0.490** | **0.477** | 0.681 | 0.556 | 0.769 | 0.549 | 0.693 | 0.579 | 0.844 | 0.581 | 1.062 | 0.747 |
| | Avg | 0.435 | 0.423 | 0.455 | 0.435 | 0.487 | 0.461 | 0.464 | 0.441 | 0.728 | 0.565 | **0.411** | **0.429** | 0.501 | 0.466 | 0.677 | 0.537 | 0.722 | 0.605 | 0.797 | 0.578 | 0.980 | 0.714 |
| ETTm2 | 96 | 0.191 | 0.270 | 0.192 | 0.274 | 0.224 | 0.296 | **0.188** | **0.269** | 0.245 | 0.322 | 0.213 | 0.303 | 0.191 | 0.274 | 0.212 | 0.285 | 0.291 | 0.399 | 0.229 | 0.308 | 0.331 | 0.430 |
| | 192 | **0.242** | **0.306** | 0.246 | 0.313 | 0.260 | 0.317 | 0.251 | 0.309 | 0.274 | 0.338 | 0.278 | 0.345 | 0.252 | 0.317 | 0.270 | 0.323 | 0.307 | 0.379 | 0.291 | 0.343 | 0.400 | 0.464 |
| | 336 | **0.286** | **0.332** | 0.301 | 0.340 | 0.312 | 0.349 | 0.307 | 0.346 | 0.361 | 0.394 | 0.338 | 0.385 | 0.306 | 0.353 | 0.323 | 0.353 | 0.543 | 0.559 | 0.348 | 0.376 | 0.469 | 0.498 |
| | 720 | **0.376** | **0.386** | 0.400 | 0.403 | 0.424 | 0.416 | 0.426 | 0.417 | 0.467 | 0.442 | 0.436 | 0.440 | 0.433 | 0.427 | 0.474 | 0.449 | 0.712 | 0.614 | 0.461 | 0.438 | 0.589 | 0.557 |
| | Avg | **0.274** | **0.324** | 0.284 | 0.332 | 0.305 | 0.344 | 0.293 | 0.335 | 0.336 | 0.373 | 0.316 | 0.368 | 0.296 | 0.343 | 0.320 | 0.353 | 0.463 | 0.488 | 0.332 | 0.366 | 0.447 | 0.487 |

based methods S²IP-LLM, Time-LLM, and GPT4TS, respectively. This improvement is attributed to FSCA's successful activation of LLMs' structured understanding of TS data, while the design of the Dual-Scale Context-Alignment GNNs ensures that LLMs grasp the logical relationships in demonstration examples prompt.

## C.5 CLASSIFICATION FULL RESULTS

Table 15 presents the results on 10 multivariate UEA classification datasets. In the time series classification task, we have augmented the baselines with the following methods: XGBoost (Chen & Guestrin, 2016), Rocket (Dempster et al., 2020), LSTNet (Lai et al., 2018a), LSSL (Gu et al., 2021), LightTS (Zhang et al., 2022), Pyraformer (Liu et al., 2021), TCN (Franceschi et al., 2019), and Flowformer (Huang et al., 2022). Compared to classical methods, RNN-based, Transformer-based, MLP-based, and other LLM-based approaches, FSCA demonstrates consistently superior performance. It achieves an average accuracy improvement of 2.4%, 2.7%, and 2.8% over S²IP-LLM, Time-LLM, and GPT4TS, respectively. This demonstrates the versatility of FSCA, showing that

Table 14: Full results of Zero-shot learning: the first column A → B indicates training on dataset A and testing on dataset B. **Bold**: best, Underline: second best.

| Methods | | FSCA MSE | FSCA MAE | S²IP-LLM MSE | S²IP-LLM MAE | Time-LLM MSE | Time-LLM MAE | GPT4TS MSE | GPT4TS MAE | iTransformer MSE | iTransformer MAE | DLinear MSE | DLinear MAE | PatchTST MSE | PatchTST MAE | TimesNet MSE | TimesNet MAE |
|---|---|---|---|---|---|---|---|---|---|---|---|---|---|---|---|---|---|
| ETTh1 | 96 | **0.256** | **0.323** | 0.315 | 0.377 | 0.324 | 0.368 | 0.335 | 0.374 | 0.353 | 0.394 | 0.347 | 0.400 | 0.304 | 0.350 | 0.358 | 0.387 |
| | 192 | **0.310** | **0.361** | 0.402 | 0.407 | 0.398 | 0.396 | 0.412 | 0.417 | 0.437 | 0.445 | 0.447 | 0.460 | 0.386 | 0.400 | 0.427 | 0.429 |
| ↓ | 336 | **0.313** | **0.373** | 0.453 | 0.432 | 0.410 | 0.423 | 0.441 | 0.444 | 0.482 | 0.476 | 0.515 | 0.505 | 0.414 | 0.428 | 0.449 | 0.451 |
| ETTh2 | 720 | **0.374** | **0.419** | 0.442 | 0.451 | 0.403 | 0.449 | 0.438 | 0.452 | 0.556 | 0.506 | 0.665 | 0.589 | 0.419 | 0.443 | 0.448 | 0.458 |
| | Avg | **0.313** | **0.369** | 0.403 | 0.417 | 0.384 | 0.409 | 0.406 | 0.422 | 0.457 | 0.455 | 0.493 | 0.488 | 0.380 | 0.405 | 0.421 | 0.431 |
| ETTh1 | 96 | **0.202** | **0.295** | 0.242 | 0.319 | 0.236 | 0.320 | 0.236 | 0.315 | 0.247 | 0.319 | 0.255 | 0.357 | 0.215 | 0.304 | 0.239 | 0.313 |
| | 192 | **0.258** | **0.329** | 0.286 | 0.337 | 0.265 | 0.353 | 0.287 | 0.342 | 0.293 | 0.350 | 0.338 | 0.413 | 0.275 | 0.339 | 0.291 | 0.342 |
| ↓ | 336 | **0.311** | **0.360** | 0.351 | 0.367 | 0.337 | 0.376 | 0.341 | 0.374 | 0.364 | 0.419 | 0.425 | 0.465 | 0.334 | 0.373 | 0.342 | 0.371 |
| ETTm2 | 720 | **0.390** | **0.407** | 0.422 | 0.416 | 0.429 | 0.430 | 0.435 | 0.422 | 0.534 | 0.470 | 0.640 | 0.573 | 0.431 | 0.424 | 0.434 | 0.419 |
| | Avg | **0.290** | **0.348** | 0.325 | 0.360 | 0.317 | 0.370 | 0.325 | 0.363 | 0.360 | 0.390 | 0.415 | 0.452 | 0.314 | 0.360 | 0.327 | 0.361 |
| ETTh2 | 96 | **0.475** | 0.473 | 0.668 | 0.567 | 0.618 | 0.515 | 0.732 | 0.577 | 0.854 | 0.606 | 0.689 | 0.555 | 0.485 | **0.465** | 0.848 | 0.601 |
| | 192 | **0.520** | **0.500** | 0.575 | 0.526 | 0.715 | 0.570 | 0.758 | 0.559 | 0.863 | 0.615 | 0.707 | 0.568 | 0.565 | 0.509 | 0.860 | 0.610 |
| ↓ | 336 | **0.528** | **0.512** | 0.655 | 0.577 | 0.636 | 0.523 | 0.759 | 0.578 | 0.867 | 0.626 | 0.710 | 0.577 | 0.581 | 0.515 | 0.867 | 0.626 |
| ETTh1 | 720 | **0.586** | **0.545** | 0.778 | 0.568 | 0.683 | 0.553 | 0.781 | 0.597 | 0.887 | 0.654 | 0.704 | 0.596 | 0.628 | 0.561 | 0.887 | 0.648 |
| | Avg | **0.527** | **0.507** | 0.669 | 0.560 | 0.663 | 0.540 | 0.757 | 0.578 | 0.868 | 0.625 | 0.703 | 0.574 | 0.565 | 0.513 | 0.865 | 0.621 |
| ETTh2 | 96 | **0.200** | **0.293** | 0.221 | 0.303 | 0.258 | 0.326 | 0.253 | 0.329 | 0.244 | 0.330 | 0.240 | 0.336 | 0.226 | 0.309 | 0.248 | 0.324 |
| | 192 | **0.256** | **0.326** | 0.295 | 0.344 | 0.303 | 0.342 | 0.293 | 0.346 | 0.291 | 0.356 | 0.295 | 0.369 | 0.289 | 0.345 | 0.296 | 0.352 |
| ↓ | 336 | **0.310** | **0.358** | 0.340 | 0.376 | 0.356 | 0.383 | 0.347 | 0.376 | 0.351 | 0.391 | 0.345 | 0.397 | 0.348 | 0.379 | 0.353 | 0.383 |
| ETTm2 | 720 | **0.384** | **0.409** | 0.453 | 0.428 | 0.440 | 0.434 | 0.446 | 0.429 | 0.452 | 0.451 | 0.432 | 0.442 | 0.439 | 0.427 | 0.471 | 0.446 |
| | Avg | **0.288** | **0.347** | 0.327 | 0.363 | 0.339 | 0.371 | 0.335 | 0.370 | 0.335 | 0.382 | 0.328 | 0.386 | 0.325 | 0.365 | 0.342 | 0.376 |
| ETTm1 | 96 | **0.302** | **0.361** | 0.358 | 0.382 | 0.355 | 0.403 | 0.353 | 0.392 | 0.371 | 0.407 | 0.365 | 0.415 | 0.354 | 0.385 | 0.377 | 0.407 |
| | 192 | **0.354** | **0.395** | 0.454 | 0.444 | 0.449 | 0.450 | 0.443 | 0.437 | 0.463 | 0.458 | 0.454 | 0.462 | 0.447 | 0.434 | 0.471 | 0.453 |
| ↓ | 336 | **0.349** | **0.398** | 0.488 | 0.452 | 0.479 | 0.467 | 0.469 | 0.461 | 0.481 | 0.485 | 0.496 | 0.494 | 0.481 | 0.463 | 0.472 | 0.484 |
| ETTh2 | 720 | **0.407** | **0.438** | 0.469 | 0.478 | 0.477 | 0.476 | 0.466 | 0.468 | 0.503 | 0.482 | 0.541 | 0.529 | 0.474 | 0.471 | 0.495 | 0.482 |
| | Avg | **0.353** | **0.398** | 0.442 | 0.439 | 0.440 | 0.449 | 0.433 | 0.439 | 0.455 | 0.458 | 0.464 | 0.475 | 0.439 | 0.438 | 0.457 | 0.454 |
| ETTm1 | 96 | **0.175** | **0.261** | 0.203 | 0.299 | 0.218 | 0.271 | 0.217 | 0.294 | 0.219 | 0.305 | 0.221 | 0.314 | 0.195 | 0.271 | 0.222 | 0.295 |
| | 192 | **0.231** | **0.298** | 0.272 | 0.325 | 0.288 | 0.335 | 0.277 | 0.327 | 0.277 | 0.347 | 0.286 | 0.359 | 0.258 | 0.311 | 0.288 | 0.337 |
| ↓ | 336 | **0.285** | **0.334** | 0.303 | 0.347 | 0.322 | 0.355 | 0.331 | 0.360 | 0.354 | 0.378 | 0.357 | 0.406 | 0.317 | 0.348 | 0.341 | 0.367 |
| ETTm2 | 720 | **0.363** | **0.383** | 0.436 | 0.418 | 0.414 | 0.409 | 0.429 | 0.413 | 0.426 | 0.420 | 0.476 | 0.476 | 0.416 | 0.404 | 0.436 | 0.418 |
| | Avg | **0.264** | **0.319** | 0.304 | 0.347 | 0.311 | 0.343 | 0.313 | 0.348 | 0.319 | 0.363 | 0.335 | 0.389 | 0.296 | 0.334 | 0.322 | 0.354 |
| ETTm2 | 96 | **0.277** | **0.345** | 0.324 | 0.383 | 0.334 | 0.416 | 0.360 | 0.401 | 0.347 | 0.401 | 0.333 | 0.391 | 0.327 | 0.367 | 0.360 | 0.401 |
| | 192 | **0.349** | **0.393** | 0.403 | 0.422 | 0.439 | 0.441 | 0.434 | 0.437 | 0.438 | 0.444 | 0.441 | 0.456 | 0.411 | 0.418 | 0.434 | 0.437 |
| ↓ | 336 | **0.343** | **0.398** | 0.434 | 0.442 | 0.455 | 0.457 | 0.460 | 0.459 | 0.459 | 0.464 | 0.505 | 0.503 | 0.439 | 0.447 | 0.460 | 0.459 |
| ETTh2 | 720 | **0.401** | **0.437** | 0.462 | 0.467 | 0.488 | 0.479 | 0.485 | 0.477 | 0.485 | 0.477 | 0.543 | 0.534 | 0.459 | 0.470 | 0.485 | 0.477 |
| | Avg | **0.343** | **0.393** | 0.406 | 0.429 | 0.429 | 0.448 | 0.435 | 0.443 | 0.432 | 0.447 | 0.455 | 0.471 | 0.409 | 0.425 | 0.435 | 0.443 |
| ETTm2 | 96 | **0.401** | **0.415** | 0.583 | 0.524 | 0.488 | 0.445 | 0.747 | 0.558 | 0.619 | 0.564 | 0.570 | 0.490 | 0.491 | 0.437 | 0.747 | 0.558 |
| | 192 | **0.457** | **0.453** | 0.609 | 0.501 | 0.555 | 0.464 | 0.781 | 0.560 | 0.685 | 0.565 | 0.590 | 0.506 | 0.530 | 0.470 | 0.781 | 0.560 |
| ↓ | 336 | **0.497** | **0.480** | 0.585 | 0.522 | 0.608 | 0.538 | 0.778 | 0.578 | 0.792 | 0.578 | 0.706 | 0.567 | 0.565 | 0.497 | 0.778 | 0.578 |
| ETTm1 | 720 | **0.563** | **0.502** | 0.712 | 0.579 | 0.699 | 0.566 | 0.769 | 0.573 | 0.727 | 0.579 | 0.731 | 0.584 | 0.686 | 0.565 | 0.769 | 0.573 |
| | Avg | **0.480** | **0.463** | 0.622 | 0.532 | 0.588 | 0.503 | 0.769 | 0.567 | 0.706 | 0.572 | 0.649 | 0.537 | 0.568 | 0.492 | 0.769 | 0.567 |

its framework can be effectively applied beyond time-series forecasting tasks to achieve excellent performance in other tasks as well.

Table 15: Full results for the classification task. '.' indicates the name of *former. **Bold**: best, Underline: second best.

| Methods | Classical methods XGBoost | Classical methods Rocket | RNN LSTNet | RNN LSSL | TCN | Transformers Trans. | Transformers Re. | Transformers In. | Transformers Pyra. | Transformers iTrans. | Transformers Station. | Transformers FED. | Transformers ETS. | Transformers Flow. | MLP DLinear | MLP LightTS. | TimesNet | LLM-based GPT4TS | LLM-based Time-LLM | LLM-based S²IP-LLM | LLM-based FSCA |
|---|---|---|---|---|---|---|---|---|---|---|---|---|---|---|---|---|---|---|---|---|---|
| EthanolConcentration | 43.7 | **45.2** | 39.9 | 31.1 | 28.9 | 32.7 | 31.9 | 31.6 | 30.8 | 32.3 | 32.7 | 31.2 | 28.1 | 33.8 | 32.6 | 29.7 | 35.7 | 34.2 | 34.6 | 35.3 | 39.2 |
| FaceDetection | 63.3 | 64.7 | 65.7 | 66.7 | 52.8 | 67.3 | 68.6 | 67.0 | 65.7 | 68.5 | 68.0 | 66.0 | 66.3 | 67.6 | 68.0 | 67.5 | 68.6 | 69.2 | 67.9 | 68.5 | **70.4** |
| Handwriting | 15.8 | **58.8** | 25.8 | 24.6 | 53.3 | 32.0 | 27.4 | 32.8 | 29.4 | 31.7 | 31.6 | 28.0 | 32.5 | 33.8 | 27.0 | 26.1 | 32.1 | 32.7 | 32.0 | 33.1 | 38.4 |
| Heartbeat | 73.2 | 75.6 | 77.1 | 72.7 | 75.6 | 76.1 | 77.1 | **80.5** | 75.6 | 75.6 | 73.7 | 73.7 | 71.2 | 77.6 | 75.1 | 75.1 | 78.0 | 77.2 | 78.0 | 77.5 | 79.5 |
| JapaneseVowels | 86.5 | 96.2 | 98.1 | 98.4 | 98.9 | 98.7 | 97.8 | 98.9 | 98.4 | 98.3 | **99.2** | 98.4 | 95.9 | 98.9 | 96.2 | 96.2 | 98.4 | 98.6 | 98.1 | 98.6 | 98.9 |
| PEMS-SF | **98.3** | 75.1 | 86.7 | 86.1 | 68.8 | 82.1 | 82.7 | 81.5 | 83.2 | 88.4 | 87.3 | 80.9 | 86.0 | 83.8 | 75.1 | 88.4 | 89.6 | 87.9 | 87.2 | 88.4 | 91.3 |
| SelfRegulationSCP1 | 84.6 | 90.8 | 84.0 | 90.8 | 84.6 | 92.2 | 90.4 | 90.1 | 88.1 | 90.7 | 89.4 | 88.7 | 89.6 | 92.5 | 87.3 | 89.8 | 91.8 | 93.2 | 92.8 | 91.4 | **94.2** |
| SelfRegulationSCP2 | 48.9 | 53.3 | 52.8 | 52.2 | 55.6 | 53.9 | 56.7 | 53.3 | 53.3 | 56.6 | 57.2 | 54.4 | 55.0 | 56.1 | 50.5 | 51.1 | 57.2 | 59.4 | 57.2 | 58.3 | **61.1** |
| SpokenArabicDigits | 69.6 | 71.2 | **100.0** | 100.0 | 95.6 | 98.4 | 97.0 | **100.0** | 99.6 | **100.0** | **100.0** | **100.0** | **100.0** | 98.8 | 81.4 | 100.0 | 99.0 | 99.2 | 99.5 | 99.0 | 99.8 |
| UWaveGestureLibrary | 75.9 | **94.4** | 87.8 | 85.9 | 88.4 | 85.6 | 85.6 | 85.6 | 83.4 | 82.5 | 87.5 | 85.3 | 85.0 | 86.6 | 82.1 | 80.3 | 85.3 | 88.1 | 89.3 | 88.7 | 91.3 |
| Average | 66.0 | 72.5 | 71.8 | 70.9 | 70.3 | 71.9 | 71.5 | 72.1 | 70.8 | 72.5 | 72.7 | 70.7 | 71.0 | 73.0 | 67.5 | 70.4 | 73.6 | 74.0 | 73.7 | 73.9 | **76.4** |

# D VISUALIZATION

Figure 3 presents visualization examples of FSCA prediction results on the ETTh1, ETTm1, Electric, and Traffic datasets with an input length of 512 and a prediction length of 96. It can be observed that FSCA achieves good predictive performance across various datasets.

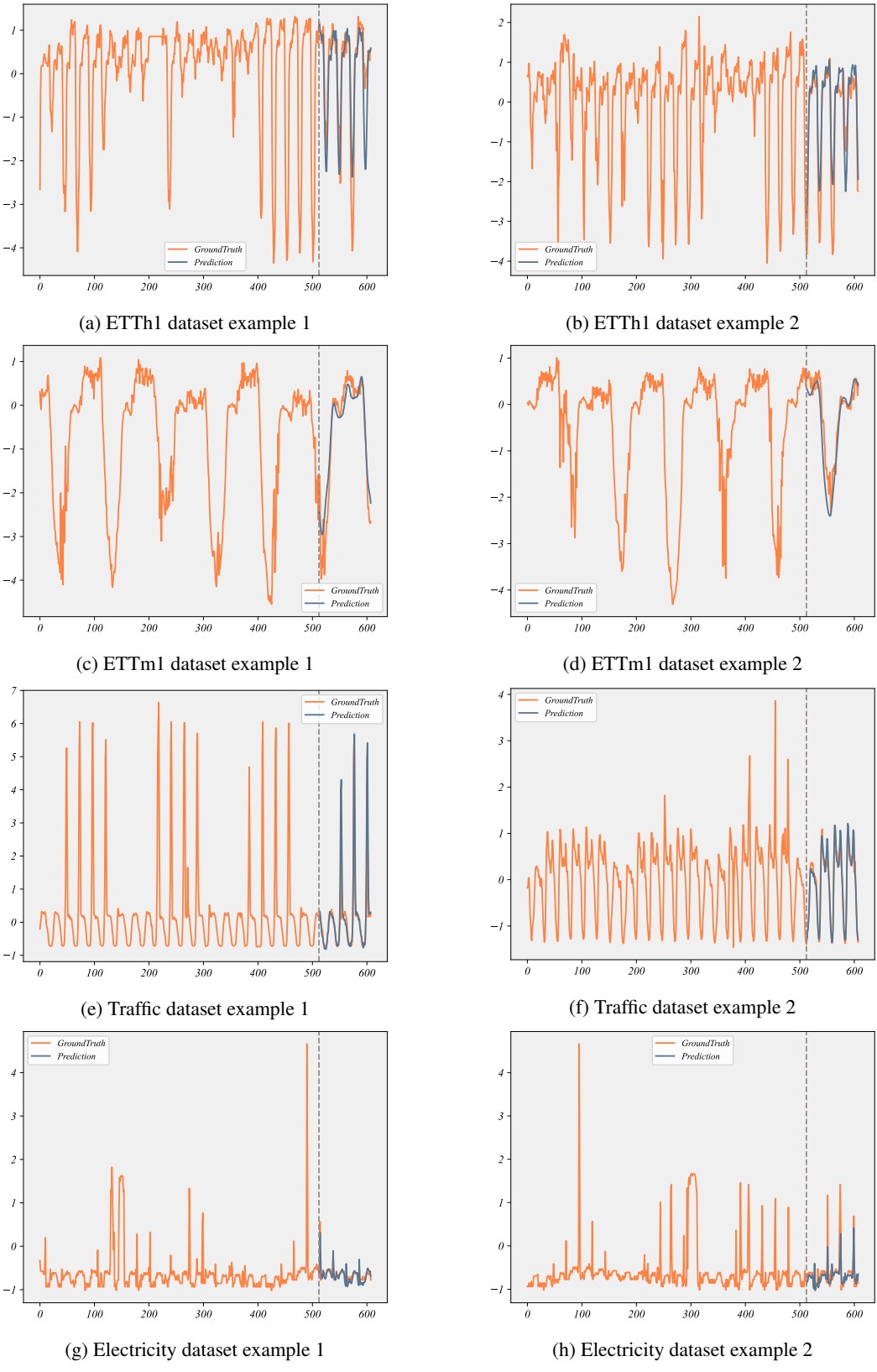

(a) ETTh1 dataset example 1

(b) ETTh1 dataset example 2

(c) ETTm1 dataset example 1

(d) ETTm1 dataset example 2

(e) Traffic dataset example 1

(f) Traffic dataset example 2

(g) Electricity dataset example 1

(h) Electricity dataset example 2

Figure 3: Sequential visualization examples of prediction results on the ETTh1, ETTm1, Electricity, and Traffic datasets are presented, with two examples per dataset. The blue lines represent predictions, while the orange lines indicate ground truth. The visualizations start at x-axis position 512.

# E  SCHEMATIC DIAGRAM OF THE FRAMEWORK

Figure 4: Schematic diagram of the process of Vanilla Context-Alignment (VCA) and Few-Shot Prompting based Context-Alignment (FSCA). The light-colored square sequence represents the fine-grained branch; the dark-colored square sequence represents the coarse-grained branch.

In this section, we describe our method in greater detail and specificity through a diagram and quantitative examples.

The bottom of Fig. 4 features the TS data input. The left subfigure presents the Vanilla Context-Alignment (VCA) process for any TS tasks, where the TS input is tokenized together with the task description prompt to construct the simple graph structure; edges in both coarse-grained and fine-grained graphs are built from the TS input directed toward the task prompt. The middle subfigure illustrates Few-Shot Prompting Context-Alignment (FSCA) for forecasting tasks, which is detailed in Sec. 3.3, showing the division of the TS input into subsequences, with the subsequences positioned later serving as the ground truth for earlier ones, thereby forming the edge connections. The right subfigure depicts the classification task using FSCA (detailed in Sec. B.2), differing from forecasting tasks in that, unable to segment the TS input to create examples, we must extract one sample from the training set for each category as the fixed example.

Below, we would facilitate the understanding of FSCA for tackling forecasting tasks by detailing a simple example. This could also provide clearer explanations of Eq.5 and Eq.6. The processes VCA and FSCA for classification tasks are similar and would not be reiterated.

Assuming we have inputs processed through patching and token embedding, these include TS embeddings of length 8 and task description prompt embeddings of length 2 (the prompt is "Predict future sequences using previous data:" in FSCA for forecasting, here the length is an example):

Firstly, for the fine-grained branch, consider the scenario in which the input TS embeddings are segmented into 2 subsequences, each comprising four embeddings. Thus, $TS_{sub}^1$ is $[e_{1,1}, e_{1,2}, e_{1,3}, e_{1,4}]$, and $TS_{sub}^2$ is $[e_{2,1}, e_{2,2}, e_{2,3}, e_{2,4}]$, where $e_{i,j}$ indicates $j$-th embedding in subsequence $i$. Similarly, $z_{i,j}$ refers to $j$-th embedding in the prompt of subsequence $i$. Here, $[z_{1,1}, z_{1,2}] = [z_{2,1}, z_{2,2}]$. Ultimately, Eq.5 is instantiated as:

$$[e_{1,1}, e_{1,2}, e_{1,3}, e_{1,4}, z_{1,1}, z_{1,2}, e_{2,1}, e_{2,2}, e_{2,3}, e_{2,4}, z_{2,1}, z_{2,2}].$$

Secondly, we need to construct a graph structure for this input before it enters LLM. The basic logic for constructing the graph is that $TS_{sub}^2$ serves as the ground truth for $TS_{sub}^1$ (The latter subsequence serves as the correct label for the former subsequence). Specifically, starting with all elements in $TS_{sub}^1$, construct directed edges to the first item of the corresponding task description, $z_{1,1}$. Subsequently, from the last item of the task description, $z_{1,2}$, construct directed edges to all elements in $TS_{sub}^2$. Since all TS subsequences are used to predict future sequences, the first token of the last prompt, $z_{2,1}$, needs to establish edge connections with both $TS_{sub}^1$ and $TS_{sub}^2$.

Thirdly, for the coarse-grained branch, it is essential to inform the LLM that a time series should be treated as a whole. Thus, $TS_{sub}^i$ must be mapped to individual node embedding by a linear

layer. To align the scales, the prompt embeddings are also mapped to a node embedding. Thus, the coarse-grained sequences can be denoted as $[\tilde{e}_1, \tilde{\mathbf{z}}^{(1)}, \tilde{e}_2, \tilde{\mathbf{z}}^{(2)}]$ (instantiation of Eq.6). Additionally, the graph construction logic is consistent with that of the fine-grained branch.

# F    EXPERIMENTAL ANALYSIS

## F.1    RANDOMLY REPLACE THE PRE-TRAINED WEIGHTS OF LLM

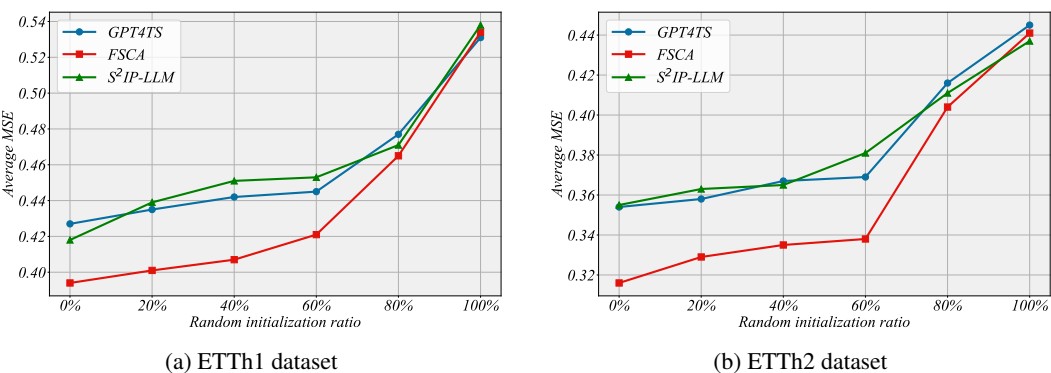

(a) ETTh1 dataset                                (b) ETTh2 dataset

Figure 5: Results of random initializing GPT-2 pre-trained weights. The x-axis represents the ratio of GPT-2 pre-trained weights replaced by random initialization, and the y-axis shows the Mean Squared Error (MSE) metric values. We conducted this experiment on GPT4TS, S$^2$IP-LLM and our FSCA.

As shown in Fig. 5, we randomly initialize GPT-2 pre-trained weights at varying ratios to demonstrate scenarios involving under-trained and untrained conditions. As the random initialization ratio increases, LLM's contextual understanding ability decreases, leading to a decline in our model's performance. When the model's capability is weak, our results are as poor as those of GPT4TS (the most direct method to utilize LLM for TS tasks) and S$^2$IP-LLM (a token-alignment based method). However, as the LLM's capability improves, our method significantly outperforms GPT4TS and S$^2$IP-LLM, achieving a lower MSE. These findings demonstrate that our approach more effectively activates the potential of pre-trained LLMs for TS tasks.

## F.2    SCALE ANALYSIS

We analyze the scaling effect of FSCA in three aspects:

**Model Size**: Ablation experiments on the number of GPT-2 layers (Table 6) show performance declines as the layer count increases, consistent with findings in GPT4TS.

**Training Data Size**: As shown in Table 17, using 5%, 10%, 25%, 50%, 75% and full of the training data reveals continuous performance improvement, particularly significant at the 50% data point.

**Few-Shot Prompting Examples' amount**: Table 16 shows that more examples yield modest gains for short prediction lengths (96) but reduce effectiveness for longer ones (336, 720). This likely results from shorter input lengths per example due to divisions of the TS input, creating a mismatch with longer required prediction lengths.

## F.3    FULLY TUNING ANALYSIS

GPT4TS has demonstrated that fully tuning LLMs for TS tasks, while straightforward, incurs high computational costs and yields suboptimal results by compromising the inherent generic knowledge of LLMs. Instead, our approach, like other pre-trained LLM-based TS methods (e.g., Time-LLM, TEST, S$^2$IP-LLM), focuses on freezing most LLM components to efficiently harness their potential. As shown in Table 18, we still supplement our results with full tuning, which similarly performed suboptimally.

Table 16: Results of different examples amount for the long-term forecasting task. All results are averaged on different prediction horizons: {96, 192, 336, 720}. **Bold**: best, Underline: second best.

| Examples amount | | 1 | | 2 | | 3 | | 4 | |
|---|---|---|---|---|---|---|---|---|---|
| Metric | | MSE | MAE | MSE | MAE | MSE | MAE | MSE | MAE |
| ETTh1 | 96 | 0.349 | 0.389 | 0.343 | 0.385 | 0.341 | 0.382 | 0.356 | 0.394 |
| | 192 | 0.390 | 0.415 | 0.387 | 0.416 | 0.393 | 0.419 | 0.402 | 0.428 |
| | 336 | 0.402 | 0.432 | 0.407 | 0.439 | 0.414 | 0.445 | 0.440 | 0.456 |
| | 720 | 0.433 | 0.460 | 0.446 | 0.471 | 0.462 | 0.488 | 0.485 | 0.495 |
| | Avg | **0.394** | **0.424** | 0.396 | 0.428 | 0.403 | 0.434 | 0.421 | 0.443 |
| ETTm1 | 96 | 0.282 | 0.343 | 0.277 | 0.340 | 0.275 | 0.341 | 0.296 | 0.352 |
| | 192 | 0.324 | 0.369 | 0.326 | 0.374 | 0.331 | 0.385 | 0.341 | 0.377 |
| | 336 | 0.356 | 0.386 | 0.366 | 0.391 | 0.370 | 0.395 | 0.391 | 0.408 |
| | 720 | 0.405 | 0.417 | 0.412 | 0.425 | 0.428 | 0.432 | 0.451 | 0.438 |
| | Avg | **0.342** | **0.378** | 0.345 | 0.383 | 0.351 | 0.388 | 0.370 | 0.394 |

Table 17: Results of different training data ratios.

| Training data ratios | ETTh1 | ETTm1 |
|---|---|---|
| 5% | 0.575 | 0.435 |
| 10% | 0.538 | 0.435 |
| 25% | 0.486 | 0.411 |
| 50% | 0.409 | 0.366 |
| 75% | 0.398 | 0.350 |
| 100% | 0.394 | 0.342 |

## F.4 COMPARISON BETWEEN VCA AND OTHER LLM-BASED METHODS

In this section, we focus on comparing VCA with other LLM-based methods. The results show that VCA generally performs second only to FSCA, demonstrating the effectiveness of Context-Alignment, which outperforms Token-Alignment and other LLM-based approaches. Compared to FSCA, the results also validate the enhancing effectiveness of the few-shot prompting technique employed in FSCA. Table 19, Table 20, Table 21, Table 22, Table 23, respectively show the results of long-term forecasting, short-term forecasting, few-shot forecasting, zero-shot forecasting and classification tasks.

## F.5 EXPERIMENTAL EFFICIENCY ANALYSIS

We incorporate comparisons of experimental efficiency with other LLM-based methods, focusing on the number of parameters and execution speed. As shown in Table 24. Our method ranks just behind GPT4TS, which only incorporates linear layers at the input and output stages of LLMs. In contrast, other popular methods require token-alignment to adapt LLMs for time series data, aligning TS data with word embeddings in the vocabulary. Additionally, these methods often include extra operations. For example, Time-LLM repeatedly generates prompts and retrieves corresponding embeddings each iteration, while $S^2$IP-LLM separates TS inputs and performs prompt retrieval.

The computational costs of our FSCA method mainly stem from two aspects. Firstly, the use of dual-scale GNNs introduces two learnable matrices (Eq. 3), which, as shown in the table comparing FSCA with the version without dual-scale GNNs, add a slight increase in computational load. Secondly, the process of constructing coarse-grained inputs necessitates two learnable linear layers to transform fine-grained node embeddings into coarse-grained ones. The input dimension of these layers is dictated by the number of input TS patches, making it the primary contributor to increased overhead.

## F.6 COMPARISON OF DIFFERENT PROMPT TYPES

We compare our approach with two other types of prompts detailed in Table 25: data domain and input statistics, referencing Time-LLM (Jin et al., 2024). Here are the prompt examples:

Table 18: Fully tuning results of FSCA and GPT4TS.

| Method | ETTh1 | ETTm1 |
|---|---|---|
| GPT4TS | 0.427 | 0.352 |
| GPT4TS(Fully tuning) | 0.469 | 0.406 |
| FSCA | 0.394 | 0.342 |
| FSCA(Fully tuning) | 0.457 | 0.383 |

Table 19: For long-term forecasting tasks, comparison between VCA and other LLM-based methods. All results are averaged on different prediction horizons: {24, 36, 48, 60} for ILI and {96, 192, 336, 720} for others. **Bold**: best, Underline: second best.

| Methods | FSCA | | VCA | | S$^2$IP-LLM | | Time-LLM | | GPT4TS | |
|---|---|---|---|---|---|---|---|---|---|---|
| Metric | MSE | MAE | MSE | MAE | MSE | MAE | MSE | MAE | MSE | MAE |
| ILI | 1.380 | 0.783 | 1.428 | 0.799 | 1.552 | 0.826 | 1.713 | 0.858 | 1.925 | 0.903 |
| Weather | 0.224 | 0.262 | 0.230 | 0.268 | 0.228 | 0.265 | 0.237 | 0.269 | 0.237 | 0.270 |
| ECL | 0.159 | 0.252 | 0.163 | 0.257 | 0.166 | 0.262 | 0.167 | 0.264 | 0.167 | 0.263 |
| Traffic | 0.386 | 0.263 | 0.389 | 0.271 | 0.405 | 0.286 | 0.407 | 0.289 | 0.414 | 0.294 |
| ETTh1 | 0.394 | 0.424 | 0.417 | 0.432 | 0.418 | 0.436 | 0.426 | 0.435 | 0.427 | 0.426 |
| ETTh2 | 0.316 | 0.375 | 0.335 | 0.382 | 0.355 | 0.399 | 0.361 | 0.398 | 0.354 | 0.394 |
| ETTm1 | 0.342 | 0.378 | 0.349 | 0.380 | 0.346 | 0.382 | 0.354 | 0.384 | 0.352 | 0.383 |
| ETTm2 | 0.250 | 0.314 | 0.259 | 0.318 | 0.262 | 0.326 | 0.275 | 0.334 | 0.266 | 0.326 |
| Avg | **0.431** | **0.381** | 0.446 | 0.388 | 0.466 | 0.398 | 0.492 | 0.404 | 0.518 | 0.407 |

Table 20: For the short-term time series forecasting, comparison between VCA and other LLM-based methods. **Bold**: best, Underline: second best.

| | Methods | FSCA | VCA | S$^2$IP-LLM | Time-LLM | GPT4TS |
|---|---|---|---|---|---|---|
| Average | SMAPE | **11.828** | 11.889 | 12.021 | 12.494 | 12.690 |
| | MASE | **1.580** | 1.596 | 1.612 | 1.731 | 1.808 |
| | OWA | **0.850** | 0.855 | 0.857 | 0.913 | 0.940 |

Table 21: For the few-shot learning task on 5% training data, comparison between VCA and other LLM-based methods. All results are averaged across four different prediction horizons {96, 192, 336, 720}. **Bold**: best, Underline: second best.

| Methods | FSCA | | VCA | | S$^2$IP-LLM | | Time-LLM | | GPT4TS | |
|---|---|---|---|---|---|---|---|---|---|---|
| Metric | MSE | MAE | MSE | MAE | MSE | MAE | MSE | MAE | MSE | MAE |
| ETTh1 | **0.575** | **0.508** | 0.598 | 0.524 | 0.650 | 0.550 | 0.648 | 0.549 | 0.681 | 0.560 |
| ETTh2 | **0.366** | **0.397** | 0.382 | 0.405 | 0.380 | 0.413 | 0.398 | 0.426 | 0.400 | 0.433 |
| ETTm1 | **0.435** | **0.429** | 0.457 | 0.442 | 0.455 | 0.446 | 0.477 | 0.451 | 0.472 | 0.450 |
| ETTm2 | **0.284** | **0.332** | 0.289 | 0.340 | 0.296 | 0.342 | 0.307 | 0.348 | 0.308 | 0.346 |
| Avg | **0.415** | **0.416** | 0.432 | 0.428 | 0.445 | 0.438 | 0.458 | 0.443 | 0.465 | 0.447 |

Table 22: For the zero-shot learning results, comparison between VCA and other LLM-based methods. The first column A → B indicates training on dataset A and testing on dataset B. **Bold**: best, Underline: second best.

| Methods | FSCA | | VCA | | S$^2$IP-LLM | | Time-LLM | | GPT4TS | |
|---|---|---|---|---|---|---|---|---|---|---|
| Metric | MSE | MAE | MSE | MAE | MSE | MAE | MSE | MAE | MSE | MAE |
| ETTh1 → ETTh2 | **0.313** | **0.369** | 0.336 | 0.390 | 0.403 | 0.417 | 0.384 | 0.409 | 0.406 | 0.422 |
| ETTh1 → ETTm2 | **0.290** | **0.348** | 0.303 | 0.362 | 0.325 | 0.360 | 0.317 | 0.370 | 0.325 | 0.363 |
| ETTh2 → ETTh1 | **0.527** | **0.507** | 0.561 | 0.534 | 0.669 | 0.560 | 0.663 | 0.540 | 0.757 | 0.578 |
| ETTh2 → ETTm2 | **0.288** | **0.347** | 0.297 | 0.361 | 0.327 | 0.363 | 0.339 | 0.371 | 0.335 | 0.370 |
| ETTm1 → ETTh2 | **0.353** | **0.398** | 0.372 | 0.416 | 0.442 | 0.439 | 0.440 | 0.449 | 0.433 | 0.439 |
| ETTm1 → ETTm2 | **0.264** | **0.319** | 0.285 | 0.333 | 0.304 | 0.347 | 0.311 | 0.343 | 0.313 | 0.348 |
| ETTm2 → ETTh2 | **0.343** | **0.393** | 0.352 | 0.401 | 0.406 | 0.429 | 0.429 | 0.448 | 0.435 | 0.443 |
| ETTm2 → ETTm1 | **0.480** | **0.463** | 0.514 | 0.487 | 0.622 | 0.532 | 0.588 | 0.503 | 0.769 | 0.567 |
| Average | **0.357** | **0.393** | 0.378 | 0.411 | 0.437 | 0.431 | 0.434 | 0.429 | 0.472 | 0.441 |

**The original prompt** is straightforward: "Predict future sequences using previous data."

**The data domain prompt**, illustrated using the ETTh dataset, is: "[Data domain:] The Electricity Transformer Temperature (ETT) is crucial for long-term electric power systems management. ETTh1 and ETTh2 represent 1-hour level data. Each data point includes the 'oil temperature' and six power load features. [Task:] Predict future sequences using previous data."

**The input statistics prompt** describes:"[Input statistics:] The input features a minimum value of <min_val>, a maximum of <max_val>, and a median of <median_val>. The overall trend is <upward or downward>. [Task:] Predict future sequences using previous data."

Our results show that the data domain prompt performs nearly the same as the original prompt. The input statistics prompt slightly enhances performance. However, it requires recalculating statistical features and regenerating corresponding embeddings by the LLM tokenizer with each iteration,

Table 23: For classification tasks, comparison between VCA and other LLM-based methods. **Bold**: best, Underline: second best.

| Methods | LLM-based | | | | |
|---|---|---|---|---|---|
| | GPT4TS | Time-LLM | $S^2$IP-LLM | VCA | FSCA |
| EthanolConcentration | 34.2 | 34.6 | 35.3 | **39.2** | - |
| FaceDetection | 69.2 | 67.9 | 68.5 | 69.0 | **70.4** |
| Handwriting | 32.7 | 32.0 | 33.1 | **38.4** | - |
| Heartbeat | 77.2 | 78.0 | 77.5 | 78.5 | **79.5** |
| JapaneseVowels | 98.6 | 98.1 | 98.6 | **98.9** | - |
| PEMS-SF | 87.9 | 87.2 | 88.4 | **91.3** | - |
| SelfRegulationSCP1 | 93.2 | 92.8 | 91.4 | 93.1 | **94.2** |
| SelfRegulationSCP2 | 59.4 | 57.2 | 58.3 | 60.5 | **61.1** |
| SpokenArabicDigits | 99.2 | 99.5 | 99.0 | **99.8** | - |
| UWaveGestureLibrary | 88.1 | 89.3 | 88.7 | **91.3** | - |
| Average | 74.0 | 73.7 | 73.9 | **76.0** | - |

Table 24: Comparisons of experimental efficiency with other LLM-based methods.

| | Training Params | Training Params Percentages | Training Time for 1 iteration(s) | Inference Time for 1 iteration(s) |
|---|---|---|---|---|
| GPT4TS | 17.33M | 17.6 | 0.457 | 0.215 |
| Time-LLM | 70.85M | 46.37 | 2.894 | 1.723 |
| $S^2$IP-LLM | 56.95M | 41.25 | 2.415 | 1.316 |
| FSCA w/o Coarse Branch | 12.43M | 13.29 | 0.348 | 0.155 |
| FSCA w/o Dual-Scale GNNs | 35.83M | 30.6 | 0.556 | 0.322 |
| FSCA | 37.02M | 31.3 | 0.587 | 0.331 |

significantly slowing down the process: training time for one iteration increased from 0.587 seconds to 1.431 seconds.

Table 25: Comparison of different prompt types.

| Variant | ETTh1 | ETTm1 | ETTm1 | ETTm2 |
|---|---|---|---|---|
| FSCA(Original) | 0.394 | 0.316 | 0.342 | 0.250 |
| FSCA(Domain) | 0.396 | 0.316 | 0.343 | 0.252 |
| FSCA(Statistics) | 0.392 | 0.313 | 0.346 | 0.246 |

## F.7 COMPARISON OF DIFFERENT NETWORK TYPES IMPLEMENTING CONTEXT-ALIGNMENT

We replace the GCN module in our dual-scale GNN framework with alternative networks, including MLP, CNNs, and self-attention mechanisms, to demonstrate that graph structures are the optimal choice within the Context-Alignment paradigm. Table 26 confirms that GCN-based methods surpass other network models. The superiority of GNNs stems from their unique ability to model node-edge structures, which allows for a more nuanced representation of structural and logical relationships. In our framework, dual-scale nodes articulate hierarchical structure, while edges represent logical connections. Consequently, this dual-scale GNN framework enhances the alignment of time series data within contexts that LLMs can understand, thereby leveraging pre-trained LLMs' capabilities for time series tasks. Additionally, we integrate GraphSAGE, another prominent GNN variant, to further confirm the robustness of our framework across different graph networks.

Table 26: Results of implementing Context-Alignment paradigm using different network types.

| Variant | ETTh1 | ETTm1 | ETTm1 | ETTm2 |
|---|---|---|---|---|
| FSCA(GCN) | 0.394 | 0.316 | 0.342 | 0.250 |
| FSCA(GraphSAGE) | 0.397 | 0.321 | 0.337 | 0.247 |
| FSCA(Attention) | 0.435 | 0.347 | 0.362 | 0.271 |
| FSCA(MLP) | 0.407 | 0.334 | 0.349 | 0.269 |
| FSCA(CNN) | 0.411 | 0.340 | 0.354 | 0.262 |

## F.8 COMPARISON WITH TS FOUNDATION MODELS

We include comparative results with pre-trained time series foundation models such as UniTS-ST (Gao et al., 2024), MOMENT (Goswami et al., 2024), and TSMixer (Chen et al., 2023). Table 28, 27 reveal that while UniTS-ST outperforms some LLM-based methods, our approach still shows superior performance. We attribute this advantage to our method's effective exploitation of the deep logical and structural understanding inherent in LLMs, which better harnesses their capabilities for TS tasks. This underscores the significant potential of LLMs in TS applications.

Table 27: For classification tasks, comparison results with time series foundation models.

| Methods | LLM-based Models | | | | TS Foundation Models | |
|---|---|---|---|---|---|---|
| | GPT4TS | Time-LLM | S$^2$IP-LLM | FSCA | UniTS-ST | MOMENT |
| EthanolConcentration | 34.2 | 34.6 | 35.3 | **39.2** | 37.6 | 35.7 |
| FaceDetection | 69.2 | 67.9 | 68.5 | 70.4 | **70.5** | 63.3 |
| Handwriting | 32.7 | 32 | 33.1 | **38.4** | 29.7 | 30.8 |
| Heartbeat | 77.2 | 78 | 77.5 | 79.5 | **80.0** | 72.2 |
| JapaneseVowels | 98.6 | 98.1 | 98.6 | **98.9** | 97.8 | 71.6 |
| PEMS-SF | 87.9 | 87.2 | 88.4 | 91.3 | **93.1** | 89.6 |
| SelfRegulationSCP1 | 93.2 | 92.8 | 91.4 | **94.2** | 93.9 | 84.0 |
| SelfRegulationSCP2 | 59.4 | 57.2 | 58.3 | **61.1** | **61.1** | 47.8 |
| SpokenArabicDigits | 99.2 | 99.5 | 99.0 | **99.8** | 98.9 | 98.1 |
| UWaveGestureLibrary | 88.1 | 89.3 | 88.7 | **91.3** | 87.7 | 90.9 |
| Average | 74 | 73.7 | 73.9 | **76.4** | 75.0 | 68.4 |

Table 28: For long-term forecasting tasks, comparison results with time series foundation models.

| Methods | LLM-based Models | | | | | | | | TS Foundation Models | | | | | |
|---|---|---|---|---|---|---|---|---|---|---|---|---|---|---|
| | FSCA | | S$^2$IP-LLM | | Time-LLM | | GPT4TS | | UniTS-ST | | MOMENT | | TSMixer | |
| Metric | MSE | MAE | MSE | MAE | MSE | MAE | MSE | MAE | MSE | MAE | MSE | MAE | MSE | MAE |
| Weather | 0.224 | 0.262 | 0.228 | 0.265 | 0.237 | 0.269 | 0.237 | 0.270 | **0.216** | **0.259** | 0.228 | 0.270 | 0.225 | 0.264 |
| ECL | 0.159 | **0.252** | 0.166 | 0.262 | 0.167 | 0.264 | 0.167 | 0.263 | **0.156** | 0.253 | 0.165 | 0.260 | 0.160 | 0.257 |
| Traffic | **0.386** | **0.263** | 0.405 | 0.286 | 0.407 | 0.289 | 0.414 | 0.294 | 0.409 | 0.278 | 0.415 | 0.293 | 0.408 | 0.284 |
| ETTh1 | **0.394** | **0.424** | 0.418 | 0.436 | 0.426 | 0.435 | 0.427 | 0.426 | 0.405 | 0.426 | 0.418 | 0.436 | 0.412 | 0.428 |
| ETTh2 | **0.316** | **0.375** | 0.355 | 0.399 | 0.361 | 0.398 | 0.354 | 0.394 | 0.331 | 0.387 | 0.352 | 0.395 | 0.355 | 0.401 |
| ETTm1 | 0.342 | 0.378 | 0.346 | 0.382 | 0.354 | 0.384 | 0.352 | 0.383 | **0.337** | **0.376** | 0.344 | 0.379 | 0.347 | 0.375 |
| ETTm2 | **0.250** | **0.314** | 0.262 | 0.326 | 0.275 | 0.334 | 0.266 | 0.326 | 0.254 | 0.315 | 0.382 | 0.376 | 0.267 | 0.322 |
| Avg. | **0.296** | **0.324** | 0.311 | 0.337 | 0.318 | 0.339 | 0.317 | 0.337 | 0.301 | 0.328 | 0.329 | 0.344 | 0.311 | 0.333 |

