# OpenReview forum: "Context-Alignment: Activating and Enhancing LLMs Capabilities in Time Series"
_ICLR.cc/2025/Conference — ICLR 2025 Poster_

### Official Review · Reviewer_N4CB · 2024-10-29

**Soundness:** 3
**Presentation:** 3
**Contribution:** 3
**Rating:** 6
**Confidence:** 4

**Summary:**

The authors propose a new method for adapting LLMs for time-series analysis tasks. They test on both time-series generation (forecasting) and time-series classification. Their paper proposes the use of Dual-Scale Context-Alignment GNNs which are used to perform context alignment of the time-series and text tokens/patches.
They combine this strategy with example prompts which they are calling (Demonstration Examples Prompt Technique) and show that using DECA and the dual scale context alignment improves the performance of LLM’s compared to existing LLM benchmarks for time-series analysis.

**Strengths:**

1. Their method of context alignment seems new and interesting
2. The course and fine-grained context alignments are logical and fit well in combination with patching which is a very popular embedding strategy for time-series data
3. They several different experiments using different tasks and experiment settings to show their model performs at a high level consistently

**Weaknesses:**

1. This paper focuses on enhancing LLM’s for time-series analysis and it takes a pretrained GPT model and compares it to other LLM-based approaches. This means this approach is ultimately a pretrained model approach (with the LLM being a pretrained model). The authors need to be comparing to other pretrained models and not just LLM based ones. Since its not being used for text generation-based tasks there’s no reason to limit this evaluation to only LLM based approaches and there are many pretrained models which follow the same experimental method.
2. The proposed method hinges heavily on the use of GNN’s and while they do run an ablation study, from what I understand they don’t compare other network architectures for performing the context alignment. Given that the GNN is a major part of the proposed implementation, an ablation showing other networks as course and fine-grained context aligners is necessary. For example, Linear layers, CNNs and self-attention could be a good start.
3. I would like more details on the training, their method does add a layer of complexity in adapting LLM’s for time-series analysis and the amount of added compute for finetuning these models (time and device) should be clearly stated.

**Questions:**

1. While your methodology considers the strengths of GNN’s as context aligners, I would like to see the choice of GNN’s to be validated experimentally as opposed to other options
2. In my opinion the terminology is unnecessarily complicated, and it takes away from the impact of the paper. One key example is “VCA w/o DSCA-GNNs” used in table 1. It seems to me that this is simply just prompting since there are no context aligners used. It may be more clear if you denote VCA as context alignment without DSCA-GNNs and mark in table 1 VCA + DSCA-GNNs.
 3. What is the difference between “(DECA) DEMONSTRATION EXAMPLES BASED CONTEXT-ALIGNMENT” and few-shot prompting as in NLP. It seems like this term was invented to enhance the complexity of the paper but at the cost of reader understanding, especially if that reader has an NLP background.
4. How does the prompt effect the context alignment? Are there some prompts that harm context alignment? Since prompting is a key component of this paper, it would be useful to know how their new method performs with different prompt types.
5. Since DECA involves using examples for context alignment a fair comparison against other baselines would be the VCA with the GNN as a context aligner. I think that should be included beside DECA in the results chart

---

> ### Author Response · Authors · 2024-11-22
>
> We are very grateful to **Reviewer N4CB** for the evaluation of our context alignment as new and interesting, and for affirming our implementation approach and experimental performance. Reviewer N4CB has provided us with invaluable feedback. We are committed to addressing your concerns and enhancing the quality of our paper.
>
> **NOTE**：We summarize the research background, challenges, and key distinctions of our method, which can be found at the beginning of the response page.

---

> ### Author Response · Authors · 2024-11-22
> **Response to Weakness 1**
>
> _________________
> **Weakness 1:** *This paper focuses on enhancing LLM’s for time-series analysis and it takes a pretrained GPT model and compares it to other LLM-based approaches. This means this approach is ultimately a pretrained model approach (with the LLM being a pretrained model). The authors need to be comparing to other pretrained models and not just LLM based ones. Since its not being used for text generation-based tasks there’s no reason to limit this evaluation to only LLM based approaches and there are many pretrained models which follow the same experimental method.*
> _________________
> **Response:** Thank you for your valuable suggestions, which significantly enhance the generality of our method across various large models. We have added experimental results on additional pre-trained LLMs (BERT & T5), and pre-trained vision model (BEiT). It is evident that our approach maintains stable performance across different LLMs. However, the results on BEiT notably underperform compared to LLMs. This discrepancy is due to BEiT’s inability to understand the logic and structure within language, rendering our method unsuitable for such pre-trained models.
>
> It is important to emphasize that our work, along with many related studies ( e.g., Time-LLM [1], $S^{2}$IP-LLM [2], TEST [3],   aLLM4TS [4], and so on), adopts the perspective of GPT4TS [5]—that pre-trained LLMs contain exploitable generic knowledge—and utilizes LLMs to complete TS tasks. We believe it is driven by the generalization capability of LLMs in various domain downstream tasks, inspiring researchers to use LLMs for TS tasks. The mainstream method involves aligning TS data with the language embeddings in the vocabulary to facilitate LLMs' understanding of TS data. This process also requires the base pre-trained model to have language knowledge. Our method introduces Context-Alignment, requiring pre-trained models to understand the logic and structure within language. Therefore, our work, including other related studies, is fundamentally based on pre-trained language models.
> | Methods | GPT2   |      | BERT  |      | T5    |      | BEiT  |      |
> |---------|--------|------|-------|------|-------|------|-------|------|
> | Metric  | MSE    | MAE  | MSE   | MAE  | MSE   | MAE  | MSE   | MAE  |
> | ETTh1   | 0.394  | 0.424| 0.416 | 0.432| 0.403 | 0.426| 0.443 | 0.457|
> | ETTh2   | 0.316  | 0.375| 0.336 | 0.384| 0.322 | 0.377| 0.394 | 0.420|
> | ETTm1   | 0.342  | 0.378| 0.348 | 0.381| 0.339 | 0.375| 0.371 | 0.388|
> | ETTm2   | 0.250  | 0.314| 0.261 | 0.319| 0.248 | 0.318| 0.280 | 0.335|

---

> ### Author Response · Authors · 2024-11-22
> **Response to Weekness 2 & Question 1**
>
> _________________
> **Weekness 2:** *The proposed method hinges heavily on the use of GNN’s and while they do run an ablation study, from what I understand they don’t compare other network architectures for performing the context alignment. Given that the GNN is a major part of the proposed implementation, an ablation showing other networks as course and fine-grained context aligners is necessary. For example, Linear layers, CNNs and self-attention could be a good start.*
>
> **Question 1:** *While your methodology considers the strengths of GNN’s as context aligners, I would like to see the choice of GNN’s to be validated experimentally as opposed to other options*
>
> _________________
> **Response:** Thank you for your constructive suggestion. We have added experiments to further demonstrate the reliability of our method. We replace the GNNs in the dual-scale GNNs framework with alternative networks (linear layers, CNNs and self-attention) to validate that graph structures are a superior choice within the Context-Alignm paradigm. Supplementary results show that GNN-based methods indeed outperform other implementations. This is because the unique node-edge structure of GNNs enables them to better represent structural-logical relationships compared to other networks; here, dual-scale nodes describe the hierarchical structure, while edges depict logical relationships. Therefore, our proposed dual-scale framework based on GNNs more effectively aligns TS within a context understandable by LLMs, thus activating the capabilities of pre-trained LLMs for TS tasks.
> Besides, to address your Question 1, we incorporate another popular variant of GNN, GraphSAGE, to validate the robustness of our framework across various graph networks.
>
> |     Variant      | ETTh1| ETTm1| ETTm1| ETTm2|
> |------------------|------|------|------|------|
> | GPT4TS           | 0.427                                 | 0.354| 0.352| 0.266|
> | DECA(GCN)        | **0.394**                             | **0.316**| 0.342| 0.250|
> | DECA(GraphSAGE)  | 0.397                                 | 0.321| **0.337**| **0.247**|
> | DECA(Atten)      | 0.435                                 | 0.347| 0.362| 0.271|
> | DECA(MLP)        | 0.407                                 | 0.334| 0.349| 0.269|
> | DECA(CNN)        | 0.411                                 | 0.340| 0.354| 0.262|

---

> ### Author Response · Authors · 2024-11-22
> **Response to Weakness 3**
>
> _________________
> **Weakness 3:** *I would like more details on the training, their method does add a layer of complexity in adapting LLM’s for time-series analysis and the amount of added compute for finetuning these models (time and device) should be clearly stated.*
>
> _________________
> **Response:** Thank you for your valuable suggestion. We have supplemented the analysis with metrics for parameter count and execution speed. The computational costs in our DECA method primarily arise from two parts. Firstly, dual-scale GNNs include two learnable matrixes (Eq. 3), as demonstrated by the comparison between “w/o Dual-Scale GNNs” and DECA in the table, where this component adds a minor increase in computational load. Secondly, in constructing coarse-grained inputs, two learnable linear layers are required to map fine-grained node embeddings to coarse-grained ones. The input dimension of this layer considers the number of input TS patches, thus it is the primary source of increased overhead.
>
> Overall, the additional computational costs of DECA are acceptable relative to its performance improvements. We have also added a comparison of experimental efficiency with other LLM-based methods; our approach is second only to GPT4TS [5], which merely adds linear layers at the input and output of the LLM. Other efforts to enable LLMs to understand TS data involve token-alignment, aligning TS data with embeddings from the vocabulary. Moreover, they introduce additional complex operations, such as Time-LLM [1] needing to regenerate prompts and obtain corresponding embeddings in each iteration, and  $S^{2}$IP-LLM [2] requiring decoupling of TS inputs and prompt retrieval. Thus, they require more training parameters and have slower training speeds. Our approach proposes a context-alignment paradigm utilizing inherent LLM advantages, achievable through GNN and linear layer without the need for costly token-alignment or additional complex operations, while still delivering state-of-the-art results.
>
> Experimental details: The results in the table were obtained using the ETTh1 dataset, with an input length of 512, a forecast length of 336, batch size set at 128, using the Adam optimizer, and conducted on NVIDIA H800 80GB GPU.
>
> | Method                     | Training Params | Training Params Percentages | Training Time for 1 iteration(s) | Inference Time for 1 iteration(s) |
> |----------------------------|-----------------|-----------------------------|----------------------------------|-----------------------------------|
> | GPT4TS                     | 17.33M          | 17.6                        | 0.457                            | 0.215                             |
> | Time-LLM                    | 70.85M          | 46.37                       | 2.894                            | 1.723                             |
> | $\text{S}^{2}$IP-LLM                       | 56.95M          | 41.25                       | 2.415                            | 1.316                             |
> | DECA w/o Coarse-grained Branch      | 12.43M          | 13.29                       | 0.348                            | 0.155                             |
> | DECA w/o Dual-Scale GNNs    | 35.83M          | 30.6                        | 0.556                            | 0.322                             |
> | DECA                       | 37.02M          | 31.3                        | 0.587                            | 0.331                             |

---

> ### Author Response · Authors · 2024-11-22
> **Response to Question 2&3**
>
> _________________
> **Question 2:** *In my opinion the terminology is unnecessarily complicated, and it takes away from the impact of the paper. One key example is “VCA w/o DSCA-GNNs” used in table 1. It seems to me that this is simply just prompting since there are no context aligners used. It may be more clear if you denote VCA as context alignment without DSCA-GNNs and mark in table 1 VCA + DSCA-GNNs.*
> **Question 3:** *What is the difference between “(DECA) DEMONSTRATION EXAMPLES BASED CONTEXT-ALIGNMENT” and few-shot prompting as in NLP. It seems like this term was invented to enhance the complexity of the paper but at the cost of reader understanding, especially if that reader has an NLP background.*
>
>
> _________________
> **Response:** We apologize for any confusion caused by the name DECA. To avoid confusion with the "Few-shot forecasting task" mentioned in Section 4.4, and to prevent the misconception that our method is applicable only to this task, we changed the name before submission. In the revised version, we have reverted the method name from Demonstration Examples based Context-Alignment (DECA) back to Few-Shot prompting based Context-Alignment (FSCA). Thank you for your valuable feedback.
>
> Additionally, your understanding of "VCA w/o DSCA-GNNs" as ``this is simply just prompting since there are no context aligners used’’ is correct. However, we are unclear about your suggestion to "denote VCA as context alignment without DSCA-GNNs," since VCA employs DSCA-GNNs to achieve context alignment.
>
> We apologize for any ambiguity or confusion. This might have arisen from our introduction of a new paradigm, Context-Alignment, and the various frameworks or variants defined around this paradigm. We will provide a clearer definition in the revised version. Below is a brief summary:
> * Context-Alignment: A new paradigm for activating the capabilities of LLMs in TS tasks, encompassing both logical and structural alignment.
>
> * Dual-Scale Context-Alignment GNNs (DSCA-GNNs): DSCA-GNNs is the proposed framework that implements Context-Alignment by utilizing dual-scale edges for logical alignment and dual-scale nodes for structural alignment.
>
> * Vanilla Context-Alignment (VCA): A straightforward method based on DSCA-GNNs, constructing the dual-scale graph structure for TS input data and task description prompt.
>
> * Few-Shot Prompting based Context-Alignment (FSCA): An advanced method based on DSCA-GNNs, enhancing VCA through the few-shot prompting techniques.

---

> ### Author Response · Authors · 2024-11-22
> **Response to Question 4**
>
> _________________
> **Question 4:** *How does the prompt effect the context alignment? Are there some prompts that harm context alignment? Since prompting is a key component of this paper, it would be useful to know how their new method performs with different prompt types*
>
> _________________
> **Response:** Thanks for your insightful feedback. We supplement experiments with two types of prompts for comparison: data domain and input statistics (following Time-LLM [1]). The following are examples of different prompt types:
> 1. The original prompt is concise: ``Predict future sequences using previous data."
> 2. For the data domain prompt, using the ETTh dataset as an example:``[Data domain:] The Electricity Transformer Temperature (ETT) plays a vital role in the long-term management of electric power systems. ETTh1 and ETTh2 are for 1-hour-level. Each data point comprises the target 'oil temperature' along with six power load characteristics. [Task:] Predict future sequences using previous data."
> 3. For input statistics prompt: ``[Input statistics:] The input features a minimum value of <min_val>, a maximum of <max_val>, and a median of <median_val>. The overall trend is <upward or downward>. [Task:] Predict future sequences using previous data."
>
> Results indicate that the data domain prompt performs almost identically to the original prompt. The input statistics prompt can slightly improve performance; however, it is important to note that each iteration requires the recalculation of statistics features and the regeneration of corresponding embeddings by the LLM tokenizer. This significantly slows down the running speed: training time for 1 iteration has increased from 0.587 to 1.431 seconds.
> |      Variant       | ETTh1   | ETTm1  | ETTm1  | ETTm2  |
> |--------------------|---------|--------|--------|--------|
> | GPT4TS             | 0.427   | 0.354  | 0.352  | 0.266  |
> | DECA(Original)     | 0.394   | 0.316  | **0.342**  | 0.250  |
> | DECA(Domain)       | 0.396   | 0.316  | 0.343  | 0.252  |
> | DECA(Statistics)   | **0.392** | **0.313** | 0.346 | **0.246** |

---

> ### Author Response · Authors · 2024-11-22
> **Response to Question 5**
>
> _________________
> **Question 5:** *Since DECA involves using examples for context alignment a fair comparison against other baselines would be the VCA with the GNN as a context aligner. I think that should be included beside DECA in the results chart*
>
> _________________
> **Response:** Thanks for your valuable suggestion. We present the results of Vanilla Context-Alignment (VCA) for the long-term prediction and classification tasks here, and the results of other tasks are in Appendix G.4. Compared to other LLM-based methods, VCA retains a significant advantage, confirming that the simplest method based on Context-Alignment paradigm surpasses previous methods.
>
> Since we establish the basic steps of utilizing LLMs for TS analysis tasks: activate first, and then enhance,  thus, Demonstration Examples based Context-Alignment (DECA, renamed as FSCA in the revision) is our final version (It builds upon the ability of VCA to activate LLMs, further enhancing performance with Few-shot prompting).
>
> Moreover, we wish to emphasize that although DECA uses few-shot prompting techniques in forecasting tasks, it still ensures fairness. DECA segments input TS data to construct examples without introducing additional data.
>
> **Classification tasks, bold is the best, italic is the second best.**
> | Methods             | GPT4TS | Time-LLM |  $\text{S}^{2}$IP-LLM  | VCA  | DECA  |
> |:---------------------:|:--------:|:----------:|:----------:|:------:|:-------:|
> | EthanolConcentration| 34.2   | 34.6     | *35.3*     | **39.2** | -     |
> | FaceDetection       | *69.2*  | 67.9     | 68.5     | 69.0 | **70.4**|
> | Handwriting         | 32.7   | 32.0     | *33.1*     | **38.4** | -     |
> | Heartbeat           | 77.2   | 78.0     | 77.5     | *78.5* | **79.5**|
> | JapaneseVowels      | *98.6*   | 98.1     | *98.6*     | **98.9** | -     |
> | PEMS-SF             | 87.9   | 87.2     | *88.4*     | **91.3** | -     |
> | SelfRegulationSCP1  | *93.2*   | 92.8     | 91.4     | 93.1 | **94.2**|
> | SelfRegulationSCP2  | 59.4   | 57.2     | 58.3     | *60.5* | **61.1**|
> | SpokenArabicDigits  | 99.2   | *99.5*     | 99.0     | **99.8** | -     |
> | UWaveGestureLibrary | 88.1   | *89.3*     | 88.7     | **91.3** | -     |
> | Average             | *74.0*   | 73.7     | 73.9     | **76.0** | -     |
> _________________
>
>
> **Long-term forecasting, bold is the best, italic is the second best.**
> | Methods   | DECA     |         | VCA      |         |  $\text{S}^{2}$IP-LLM  |         | Time-LLM |         | GPT4TS   |         |
> |:-----------:|:----------:|:---------:|:----------:|:---------:|:----------:|:---------:|:----------:|:---------:|:----------:|:---------|
> | Metric    | MSE      | MAE     | MSE      | MAE     | MSE      | MAE     | MSE      | MAE     | MSE      | MAE     |
> | ILI       | 1.380    | 0.783   | 1.428    | 0.799   | 1.552    | 0.826   | 1.713    | 0.858   | 1.925    | 0.903   |
> | Weather   | 0.224    | 0.262   | 0.230    | 0.268   | 0.228    | 0.265   | 0.237    | 0.269   | 0.237    | 0.270   |
> | ECL       | 0.159    | 0.252   | 0.163    | 0.257   | 0.166    | 0.262   | 0.167    | 0.264   | 0.167    | 0.263   |
> | Traffic   | 0.386    | 0.263   | 0.389    | 0.271   | 0.405    | 0.286   | 0.407    | 0.289   | 0.414    | 0.294   |
> | ETTh1     | 0.394    | 0.424   | 0.417    | 0.432   | 0.418    | 0.436   | 0.426    | 0.435   | 0.427    | 0.426   |
> | ETTh2     | 0.316    | 0.375   | 0.335    | 0.382   | 0.355    | 0.399   | 0.361    | 0.398   | 0.354    | 0.394   |
> | ETTm1     | 0.342    | 0.378   | 0.349    | 0.380   | 0.346    | 0.382   | 0.354    | 0.384   | 0.352    | 0.383   |
> | ETTm2     | 0.250    | 0.314   | 0.259    | 0.318   | 0.262    | 0.326   | 0.275    | 0.334   | 0.266    | 0.326   |
> | Avg       | **0.431**| **0.381**| *0.446*    | *0.388*   | 0.466    | 0.398   | 0.492    | 0.404   | 0.518    | 0.407   |

---

> ### Author Response · Authors · 2024-11-22
> **References**
>
> [1] Jin M, Wang S, Ma L, et al. Time-llm: Time series forecasting by reprogramming large language models[J]. arXiv preprint arXiv:2310.01728, 2023.
>
> [2] Pan Z, Jiang Y, Garg S, et al. $ S^ 2$ IP-LLM: Semantic Space Informed Prompt Learning with LLM for Time Series Forecasting[C]//Forty-first International Conference on Machine Learning. 2024.
>
> [3] Sun C, Li H, Li Y, et al. TEST: Text Prototype Aligned Embedding to Activate LLM's Ability for Time Series[C]//The Twelfth International Conference on Learning Representations.
>
> [4] Bian Y, Ju X, Li J, et al. Multi-Patch Prediction: Adapting Language Models for Time Series Representation Learning[C]//Forty-first International Conference on Machine Learning.
>
> [5] Zhou T, Niu P, Sun L, et al. One fits all: Power general time series analysis by pretrained lm[J]. Advances in neural information processing systems, 2023, 36: 43322-43355.

---

> ### Comment · Reviewer_N4CB · 2024-11-22
>
> Thank you for addressing many of my concerns with regarding to model ablations which justify the proposed architecture.
>
> As for **weakness 1**, I still have some reservations here. The argument "that pre-trained LLMs contain exploitable generic knowledge—and utilizes LLMs to complete TS tasks. We believe it is driven by the generalization capability of LLMs in various domain downstream tasks, inspiring researchers to use LLMs for TS tasks" is valid and motivates the use of LLM's for time-series analysis. I still believe however that because pretrained models for time-series have become more prominent and popular, you should still provide performance references to current pretrained time-series models and not exclusively LLM's considering they are easily benchmarked on the same datasets and tasks. This is especially prudent because the function of these two model subtypes are similar (time-series generation and classification). This would help to illustrate from a practical standpoint why augmenting time-series analysis with text is an important direction.

---

> > ### Author Response · Authors · 2024-11-23
> > **More Discussion**
> >
> > Thank you for your valuable feedback. We believe both technical routes are worth exploring. Here, referencing prior research and our own understanding, we explain why we focus on utilizing pre-trained LLMs rather than pre-trained TS models:
> >
> > * From a data perspective:
> >
> >     1. Training pre-trained TS models require extensive datasets, which are more challenging to gather in the TS field compared to NLP; LLM-based models can achieve desired outcomes with smaller datasets specific to downstream tasks[4].
> >
> >     2. TS datasets vary significantly in frequency and cyclical patterns, leading to large differences in data distribution and posing challenges in knowledge transferability across different domains[2,4,5]. However, LLMs contain exploitable generic knowledge.
> >
> > * From a model pespective: Training pre-trained TS models is generally time-consuming; for instance, training MOMENT Small takes up to 300 GPU hours[2]. In contrast, LLM-based methods require little or even no training, making them more general and convenient.
> >
> > * From a multimodal perspective: LLM-based models typically freeze LLM pre-trained weights, maintaining text processirng capabilities. Thus, we can use NLP techniques like prompts to guide the model (eg. our method). Additionally, in tasks like weather and financial forecasting, incorporating real-time data from social media can enhance accuracy[5,6]. Further research into leveraging complementary insights across different modalities in multimodal large models could not only improve time series forecasting performance but also enhance interpretability.
> >
> > _________________
> >
> > **Reference**
> >
> > [1] Gao S, Koker T, Queen O, et al. Units: Building a unified time series model[J]. arXiv preprint arXiv:2403.00131, 2024. (Accepted by NeurIPS2024)
> >
> > [2] Goswami M, Szafer K, Choudhry A, et al. MOMENT: A Family of Open Time-series Foundation Models[C]//Forty-first International Conference on Machine Learning.
> >
> > [3] Chen S A, Li C L, Arik S O, et al. TSMixer: An All-MLP Architecture for Time Series Forecast-ing[J]. Transactions on Machine Learning Research.
> >
> > [4] Sun C, Li H, Li Y, et al. TEST: Text Prototype Aligned Embedding to Activate LLM's Ability for Time Series[C]//The Twelfth International Conference on Learning Representations
> >
> > [5] Ye J, Zhang W, Yi K, et al. A Survey of Time Series Foundation Models: Generalizing Time Series Representation with Large Language Mode[J]. arXiv preprint arXiv:2405.02358, 2024.
> >
> > [6] Wang X, Feng M, Qiu J, et al. From News to Forecast: Integrating Event Analysis in LLM-Based Time Series Forecasting with Reflection[J]. arXiv preprint arXiv:2409.17515, 2024. (Accepted by NeurIPS2024)

---

> > > ### Comment · Reviewer_N4CB · 2024-11-25
> > >
> > > Thank you for addressing my concerns. Based on this I am happy to increase my score.

---

> ### Author Response · Authors · 2024-11-23
> **Re.Response to Weakness 1**
>
> We sincerely apologize for the misunderstanding in our previous response. We have now included comparative results with pre-trained TS models (UniTS-ST[1], MOMENT[2], TSMixer[3]). As shown in the following tables, while UniTS-ST (NeurIPS2024), the best performing among the pre-trained TS models, surpasses some previous LLM-based methods, our method still demonstrates superior performance. We attribute this to our method's effective use of the deep logical and structural understanding of LLMs, which better harnesses LLM capabilities for TS tasks. This further confirms the potential of LLMs in TS applications.
> _________________
>
> **Baseline results of pre-trained TS models UniTS-ST，MOMENT，TSMixer are obtained from their original papers. UniTS-ST is the latest paper accepted by NeurIPS2024. TSMixer doesn’t offer the results of classification tasks.**
>
> **The results of long-term forecasting tasks, bold is the best, italic is the second best.**
>
> _________________
>
>
> | Methods | FSCA      |        | $\text{S}^{2}$IP-LLM   |       | Time-LLM     |     | GPT4TS    |        | UniTS-ST     |     | MOMENT       |     | TSMixer     |      |
> |---------|---------|---------|---------|---------|---------|---------|---------|---------|---------|---------|---------|---------|---------|---------|
> | Metric  | MSE     | MAE     | MSE     | MAE     | MSE     | MAE     | MSE     | MAE     | MSE     | MAE     | MSE     | MAE     | MSE     | MAE     |
> | Weather | 0.224   | *0.262*   | 0.228   | 0.265   | 0.237   | 0.269   | 0.237   | 0.270   | **0.216** | **0.259** | 0.228   | 0.270   | *0.225*   | 0.264   |
> | ECL     | *0.159*   | **0.252** | 0.166   | 0.262   | 0.167   | 0.264   | 0.167   | 0.263   | **0.156** | *0.253* | 0.165   | 0.260   | 0.160   | 0.257   |
> | Traffic | **0.386** | **0.263**   | *0.405* | 0.286   | 0.407 | 0.289   | 0.414 | 0.294   | 0.409 | *0.278*   | 0.415 | 0.293   | 0.408 | 0.284   |
> | ETTh1   | **0.394**   | **0.424** | 0.418   | 0.436   | 0.426   | 0.435   | 0.427   | *0.426*   | *0.405*   | *0.426*   | 0.418   | 0.436   | 0.412   | 0.428   |
> | ETTh2   | **0.316**   | **0.375**   | 0.355   | 0.399   | 0.361   | 0.398   | 0.354   | 0.394   | *0.331*   | *0.387*   | 0.352   | 0.395   | 0.355   | 0.401   |
> | ETTm1   | *0.342*   | *0.378*   | 0.346   | 0.382   | 0.354   | 0.384   | 0.352   | 0.383   | **0.337** | **0.376** | 0.344   | 0.379   | 0.347   | 0.375   |
> | ETTm2   | **0.250**   | **0.314**   | 0.262   | 0.326   | 0.275   | 0.334   | 0.266   | 0.326   | *0.254*   | *0.315*   | 0.382 | 0.376 | 0.267   | 0.322   |
> | Avg.    | **0.296**   | **0.324**   | 0.311   | 0.337   | 0.318   | 0.339   | 0.317   | 0.337   | *0.301*   | *0.328*   | 0.329   | 0.344   | 0.311   | 0.333   |
>
>
> _________________
>
> **The results of classification tasks, bold is the best, italic is the second best.**
>
> | Methods             | GPT4TS | Time-LLM | $\text{S}^{2}$IP-LLM | **FSCA** | **UniTS-ST** | MOMENT |
> |---------------------|--------|----------|----------|----------|--------------|--------|
> | Ethanol Concentration | 34.2   | 34.6     | 35.3     | **39.2** | *37.6*       | 35.7   |
> | FaceDetection       | 69.2   | 67.9     | 68.5     | *70.4*   | **70.5**     | 63.3   |
> | Handwriting         | 32.7   | 32       | *33.1*   | **38.4** | 29.7         | 30.8   |
> | Heartbeat           | 77.2   | 78       | 77.5     | *79.5*   | **80.0**     | 72.2   |
> | JapaneseVowels      | *98.6* | 98.1     | *98.6*   | **98.9** | 97.8         | 71.6   |
> | PEMS-SF             | 87.9   | 87.2     | 88.4     | *91.3*   | **93.1**     | 89.6   |
> | SelfRegulationSCP1  | 93.2   | 92.8     | 91.4     | **94.2** | *93.9*       | 84.0   |
> | SelfRegulationSCP2  | *59.4* | 57.2     | 58.3     | **61.1** | **61.1**     | 47.8   |
> | SpokenArabicDigits  | 99.2   | *99.5*   | 99.0     | **99.8** | 98.9         | 98.1   |
> | UWaveGestureLibrary | 88.1   | 89.3     | 88.7     | **91.3** | 87.7         | *90.9* |
> | Average             | 74     | 73.7     | 73.9     | **76.4** | *75.0*       | 68.4   |

---

> ### Author Response · Authors · 2024-11-25
>
> Thank you for your positive feedback and for acknowledging the updates we made to the paper. We appreciate your decision to increase the score and am glad that the changes met your expectations. Once again, thank you for your time, effort, and support.

---

### Official Review · Reviewer_oFgL · 2024-11-04

**Soundness:** 2
**Presentation:** 2
**Contribution:** 3
**Rating:** 6
**Confidence:** 2

**Summary:**

This paper introduces Context-Alignment, a novel paradigm for enhancing the capabilities of LLMs in time series tasks. The authors argue that leveraging LLMs' strengths in natural language processing, particularly their understanding of linguistic logic and structure, is key to improving their performance on time series data. They propose a Dual-Scale Context-Alignment Graph Neural Networks (DSCA-GNNs) framework that aligns time series data with linguistic components, enabling structural and logical alignment. This framework is used to develop Demonstration Examples based Context-Alignment (DECA), which integrates seamlessly into pre-trained LLMs to enhance their awareness of logic and structure. Extensive experiments across various time series tasks, including forecasting and classification, demonstrate DECA's effectiveness, especially in few-shot and zero-shot scenarios, highlighting the importance of context alignment in activating and enhancing LLMs' potential in time series applications.

**Strengths:**

The paper strengthens the representation of time series data through GNNs, addressing a gap in prior research on applying LLMs to time series tasks. The method shows promising results and could inspire future work in this area.

**Weaknesses:**

- The paper's incorporation of GNNs is a functional approach, yet it overlooks a comparison of modeling time consumption, which is a critical aspect of efficiency that should be measured against baselines that do not employ this method.
- Additionally, while the paper presents a generalized embedding technique, further validation across a broader range of time series scenarios is needed to establish its robustness. Testing the method in other contexts, such as time series anomaly detection and imputation, would strengthen the claims of its effectiveness.

**Questions:**

I have raised several concerns within the weaknesses. Please address the issues I've mentioned there.

---

> ### Author Response · Authors · 2024-11-22
>
> We sincerely thank **Reviewer oFgL** for recognizing our efforts in addressing a gap in processing TS tasks with LLMs, for affirming our experimental results and value on future work. Reviewer oFgL's suggestions on our experimental analysis, particularly regarding cost comparisons and additional tasks, have helped us present our research more comprehensively and enhance its quality. We hope your concerns will be addressed.
>
> **NOTE**：We summarize the research background, challenges, and key distinctions of our method, which can be found at the beginning of the response page.

---

> ### Author Response · Authors · 2024-11-22
> **Response to Weakness 1**
>
> _________________
> **Weakness 1:** *The paper's incorporation of GNNs is a functional approach, yet it overlooks a comparison of modeling time consumption, which is a critical aspect of efficiency that should be measured against baselines that do not employ this method.*
>
> _________________
> **Response:** Thanks for your constructive suggestion.
> We have added comparisons of experimental efficiency with other LLM-based methods, including parameters amount and execution speed.  As shown in the table, our method is second only to GPT4TS [1], which merely adds linear layers at the input and output of LLMs. Other mainstream approaches require token-alignment to make LLMs comprehend TS data (aligning TS data with word embeddings in the vocabulary). Moreover, they often incorporate additional operations. For instance, Time-LLM [2] regenerates prompts and obtains corresponding embeddings in each iteration, while $S^{2}$IP-LLM [3] involves decoupling TS inputs and conducting prompt retrieval.
>
> In contrast, our method utilizes the intrinsic advantages of LLMs to propose the Context-Alignment paradigm, eliminating the need for token-alignment and additional operations to achieve SOTA results. In our DECA method (renamed as FSCA in the revision), computational costs mainly arise from two aspects: first, the trainable weight matrix in the GCN that involves straightforward matrix multiplication (Eq. 3); second, two learnable linear layers that map fine-grained node embeddings to coarser ones when constructing coarse-grained inputs. Thus, despite introducing a dual-scale GNNs framework, our approach still consumes less time compared to other baseline methods.
>
> Experimental details: The results in the table were obtained using the ETTh1 dataset, with an input length of 512, a forecast length of 336, and batch size set at 128, using the Adam optimizer, and conducted on NVIDIA H800 80GB GPU.
>
> | Method                     | Training Params | Training Params Percentages | Training Time for 1 iteration(s) | Inference Time for 1 iteration(s) |
> |----------------------------|-----------------|-----------------------------|----------------------------------|-----------------------------------|
> | GPT4TS                     | 17.33M          | 17.6                        | 0.457                            | 0.215                             |
> | Time-LLM                    | 70.85M          | 46.37                       | 2.894                            | 1.723                             |
> | $\text{S}^{2}$IP-LLM                       | 56.95M          | 41.25                       | 2.415                            | 1.316                             |
> | DECA w/o Coarse-grained Branch      | 12.43M          | 13.29                       | 0.348                            | 0.155                             |
> | DECA w/o Dual-Scale GNNs    | 35.83M          | 30.6                        | 0.556                            | 0.322                             |
> | DECA                       | 37.02M          | 31.3                        | 0.587                            | 0.331                             |

---

> ### Author Response · Authors · 2024-11-22
> **Response to Weakness 2**
>
> _________________
> **Weakness 2:** Additionally, while the paper presents a generalized embedding technique, further validation across a broader range of time series scenarios is needed to establish its robustness. Testing the method in other contexts, such as time series anomaly detection and imputation, would strengthen the claims of its effectiveness.
>
> _________________
> **Response:** Thanks for your constructive suggestion.
>
> We have supplemented our experiments on anomaly detection and imputation. It is important to note that for the anomaly detection task, only the test dataset is labeled, while the training dataset does not provide labels. Therefore, to ensure a fair comparison, we implement this using our method VCA (without few-shot prompting). The imputation is performed using our advanced method DECA (with few-shot prompting, renamed as FSCA in the revision). Our method demonstrates superior performance, indicating its effective applicability across multiple TS tasks.
>
> **The results of anomaly detection of F1-score for each dataset, bold is the best, italic is the second best.**
>
> | Methods | DECA   | $\text{S}^{2}$IP-LLM | Time-LLM | GPT4TS  | iTransformer | DLinear | PatchTST | TimesNet | FEDformer | Stationary | ETSformer |
> |---------|--------|----------|----------|---------|--------------|---------|----------|----------|-----------|------------|-----------|
> | SMD     | **87.04** | 86.74    | 85.93    | *86.89*   | 86.52        | 77.10   | 84.62    | 84.61    | 85.08     | 84.72      | 83.13     |
> | MSL     | 84.61  | 83.09     | 84.24    | 82.45   | 83.30        | *84.88*   | 78.70    | 81.84    | 78.57     | 77.50      | **85.03** |
> | SMAP    | *73.46*  | 73.11     | **73.81** | 72.88   | 69.67        | 69.26   | 68.82    | 69.39    | 70.76     | 71.09      | 69.50     |
> | SWaT    | 93.78  | *93.85*     | 93.41    | **94.23** | 87.43        | 87.52   | 85.72    | 93.02    | 93.19     | 79.88      | 84.91     |
> | PSM     | **97.65** | 97.31   | 97.02  | 97.13 | 96.69        | 93.55   | 96.08    | 97.34  | 97.23   | 97.29    | 91.76     |
> | Average | **87.31** | 86.82    | *86.88*    | 86.72   | 84.72        | 82.46   | 82.79    | 85.24    | 84.97     | 82.10      | 82.87     |
>
>
> _________________
>
> **The results of imputation, bold is the best, italic is the second best.**
>
> | Methods | DECA   |      |  $\text{S}^{2}$IP-LLM  |      | Time-LLM |      | GPT4TS |      | iTransformer |      | DLinear |      | PatchTST |      | TimesNet |      | FEDformer |      | Stationary |      | ETSformer |      |
> |---------|--------|------|----------|------|----------|------|--------|------|--------------|------|---------|------|----------|------|----------|------|----------|------|------------|------|-----------|------|
> | Metric  | MSE    | MAE  | MSE      | MAE  | MSE      | MAE  | MSE    | MAE  | MSE          | MAE  | MSE     | MAE  | MSE      | MAE  | MSE      | MAE  | MSE      | MAE  | MSE        | MAE  | MSE       | MAE  |
> | ETTh1   | **0.063**  | **0.171**| 0.074    | 0.177| 0.080    | 0.185| *0.069*  | *0.173*| 0.102        | 0.207| 0.201   | 0.306| 0.115    | 0.224| 0.078    | 0.187| 0.117    | 0.246| 0.094      | 0.201| 0.202     | 0.329|
> | ETTh2   | **0.039**  | **0.132**| *0.044*    | *0.139* | 0.053    | 0.150| 0.048  | 0.141| 0.057        | 0.154| 0.142   | 0.259| 0.065    | 0.163| 0.049    | 0.146| 0.163    | 0.279| 0.053      | 0.152| 0.367     | 0.436|
> | ETTm1   | *0.024*  | *0.095* | **0.022**| 0.096| 0.025    | *0.095*| 0.028  | 0.105| 0.035        | **0.013**| 0.093   | 0.206| 0.047    | 0.140| 0.027    | 0.107| 0.062    | 0.177| 0.036      | 0.126| 0.120     | 0.253|
> | ETTm2   | **0.021**  | *0.087*| 0.026    | 0.092| 0.034    | 0.101| **0.021**| **0.084**| 0.034        | 0.110| 0.096   | 0.208| 0.029    | 0.102| *0.022*    | 0.088| 0.101    | 0.215| 0.026      | 0.099| 0.208     | 0.327|

---

> > ### Comment · Reviewer_oFgL · 2024-11-24
> > **Thanks for the response!**
> >
> > I appreciate the additional experiments and explanations, and I'd like to raise my overall assessments.
> > I believe that strengthening the representation of time-series data through GNNs holds significant value for future research.

---

> > > ### Author Response · Authors · 2024-11-24
> > > **Thank you for your insightful feedback and improved rating!**
> > >
> > > We would like to express sincere gratitude for your constructive feedback and raising your rating. The additional experiments and explanations you suggested have significantly improved my work. I appreciate your positive assessment and look forward to exploring the potential of Context-Alignment in time-series tasks. Your guidance has been invaluable.

---

> ### Author Response · Authors · 2024-11-22
> **References**
>
> [1] Zhou T, Niu P, Sun L, et al. One fits all: Power general time series analysis by pretrained lm[J]. Advances in neural information processing systems, 2023, 36: 43322-43355.
>
> [2] Jin M, Wang S, Ma L, et al. Time-llm: Time series forecasting by reprogramming large language models[J]. arXiv preprint arXiv:2310.01728, 2023.
>
> [3] Pan Z, Jiang Y, Garg S, et al. $ S^ 2$ IP-LLM: Semantic Space Informed Prompt Learning with LLM for Time Series Forecasting[C]//Forty-first International Conference on Machine Learning. 2024.

---

### Official Review · Reviewer_2N38 · 2024-11-04

**Soundness:** 3
**Presentation:** 2
**Contribution:** 3
**Rating:** 6
**Confidence:** 2

**Summary:**

This paper aims at addressing the alignment of Time series not just on the token level as done by previous works but at a level that enable LLMs to contextualize and comprehend TS in the same manner as it does for natural language. To this aim, the authors proposed Dual-Scale Context-Alignment GNNs that achieves context level alignment comprising structural alignment and logical alignment thereby activating LLMs’ potential capabilities in time series tasks.

**Strengths:**

-  The paper addresses an important question of context alignment which promises better use of LLMs in the time series domain.
- The idea of using GNNs to introduce context alignment brings novelty to this work.

**Weaknesses:**

- Since the authors use GNNs for context alignment, a befitting diagram showing the nodes and edges would have made it easier to follow the text.
- Some typos here and there in the paper.
- Datasets not described clearly in the experimental setup.

**Questions:**

- I am not completely sure if averaging the error metrics across different prediction lengths is the best way to report and compare the results.
- Is there a justification as to why this averaging approach is adopted?

---

> ### Author Response · Authors · 2024-11-22
>
> We greatly appreciate **Reviewer 2N38**'s recognition of our better use of LLMs and the novelty of our idea. Additionally, Reviewer 2N38 has provided valuable feedback on the presentation of our paper, and we are committed to making it clearer and avoiding ambiguities.
>
> **NOTE**：We summarize the research background, challenges, and key distinctions of our method, which can be found at the beginning of the response page.

---

> ### Author Response · Authors · 2024-11-22
> **Response to Weakness 1**
>
> _________________
> **Weakness 1:** *Since the authors use GNNs for context alignment, a befitting diagram showing the nodes and edges would have made it easier to follow the text.*
>
> _________________
> **Response:** Thank you very much for your valuable advice, which is crucial for improving the clarity of our work. To illustrate our methods more clearly, we have included a schematic diagram in Appendix E including the presentation of the graph structure. Here is a brief description:
>
> 1.For VCA, we first tokenize the input TS sequence and task prompt to obtain feature embeddings, depicted as light blue and light green blocks respectively, which is the fine-grained sequence. We then establish a fine-grained graph: directed edges from all input TS tokens to the first prompt token. Through learnable linear layers, the fine-grained nodes are mapped to coarse-grained nodes. The coarse-grained branch is represented by dark blue and dark green blocks, with similar edge constructed. In the training phase, information from the coarse-grained branch is transmitted to the fine-grained branch through the learnable mapping.
>
> 2.For FSCA (we have renamed DECA to FSCA in the revision) in the forecasting tasks, the overall process is similar to VCA, but we divide the input TS sequence into subsequences. In the diagram, 2 subsequences are used as a few-shot example: the first subsequence serves as a sequence used for prediction, and the second subsequence acts as the ground truth for the first. Based on this, we construct directed edges as shown in the diagram. It's important to note that both the first and second subsequences must be connected to the final prompt, as the entire sequence is required for the final prediction
>
> 3.For FSCA in the classification tasks, we extract one sample for each category in the training set as the fixed example, with the remaining process similar to VCA or FSCA.

---

> ### Author Response · Authors · 2024-11-22
> **Response to Weakness 2**
>
> _________________
> **Weakness 2:** *Some typos here and there in the paper.*
>
> _________________
> **Response:** We apologize for the typos. We have carefully reviewed and corrected several typos in the revised version. Thank you for your attention to detail, it is crucial for improving our work.

---

> ### Author Response · Authors · 2024-11-22
> **Response to Weakness 3**
>
> _________________
> **Weakness 3:** *Datasets not described clearly in the experimental setup.*
>
> _________________
> **Response:** Thank you for your valuable suggestions.
>
> In the experimental section of the main text, we briefly mention the datasets used for each task due to space limitations. A detailed description of the datasets, including statistics and more, is provided in Appendix A.2. We have refered to this detail in the Section 4 Experiment of revision (Line 301-302). If you have any further questions about the dataset, we are happy to assist you as best as we can.

---

> ### Author Response · Authors · 2024-11-22
> **Response to Question 1&2**
>
> _________________
> **Question 1:** *I am not completely sure if averaging the error metrics across different prediction lengths is the best way to report and compare the results.*
>
> **Question 2:** *Is there a justification as to why this averaging approach is adopted?*
>
> _________________
> **Response:** Thank you for your valuable question. We have presented the complete results in Appendix C “Full results” of the original manuscript. We hope this could help you further understand the performance of our work. Due to space limitations, like other studies in this field, we only show average metrics in the main text, as averages facilitate quicker comparison of different methods' performance.

---

> ### Author Response · Authors · 2024-11-25
>
> Thank you for your valuable suggestions! We have incorporated the necessary corrections and detailed explanations in the revised version, which are highlighted in red. The specific locations of these revisions have been provided in our previous itemized response.
>
> Should you have any further questions, we are eager to address them and hope to receive your positive feedback on our updates. We deeply appreciate the time and effort you have invested in reviewing our work. Thank you once again!

---

> ### Author Response · Authors · 2024-12-01
> **Reminder from Authors**
>
> Dear Reviewer 2N38,
>
> Thank you for your feedback during the review process! We believe that our detailed response has addressed your concerns. If you have any concerns or questions, please do not hesitate to let us know before the author discussion period ends (less than two days). We will be happy to answer them during the discussion.
>
> Thank you!

---

### Official Review · Reviewer_Dkrj · 2024-11-10

**Soundness:** 3
**Presentation:** 2
**Contribution:** 3
**Rating:** 6
**Confidence:** 2

**Summary:**

This work aims at leveraging and improving LLMs for time series tasks. Specifically, the authors propose context alignment, a technique that utilizes dual-scale GNNs in addition to the basic LLM architectures that helps LLM comprehend time series data. Few-shot prompting techniques in regular LLMs are also used in the time series design. Through various experiments on different time series benchmarks, the authors show the performance advantage of the proposed method over baseline models.

**Strengths:**

This paper is tackling an interesting problem of leveraging and improving pretrained LLMs to do time series tasks. The authors explored a diverse set of benchmarks and baselines and showed a superior performance of the proposed method.

**Weaknesses:**

This work motivates and explains the advantage of the proposed method by LLMs' deep understanding of linguistic logic and structure rather than superficial embedding processing. However, such explanations lack support from experiments. More analysis can be included on the LLM side. For example, does an inferior LLM basic architecture (either old design or small model sizes) or a badly trained LLM (undertrained or untrained) lead to a bad time series performance?


This work freezes most of the LLM structure and tunes dual-scale context-alignment GNNs between Transformer layers. There seems to be lacking analysis on the effect of fully tuning the attention and feed-forward layers as well, and the subsequent necessity of the dual-scale context-alignment GNNs in that case.


One of the interesting attributes of LLMs is the scaling effect (e.g., on model sizes, training data sizes, few-shot prompting amounts). The scaling aspect of LLMs and the proposed DECA seems to be unknown in the context of time series.


The writing can be improved for clarity, e.g., by providing qualitative examples besides key equations, especially in the few-shot prompting part, and adding an algorithm table.

**Questions:**

N/A

---

> ### Author Response · Authors · 2024-11-22
>
> We wish to express our sincere gratitude to **Reviewer Dkrj** for the interesting evaluation of the problem we addressed, and for acknowledging our extensive experimentation and effectiveness. We also greatly appreciate suggestions for further experimental analysis. Rest assured, we are committed to addressing these concerns and improving our work.
>
> **NOTE**：We summarize the research background, challenges, and key distinctions of our method, which can be found at the beginning of the response page.

---

> ### Author Response · Authors · 2024-11-22
> **Response to Weakness 1**
>
> _________________
> **Weakness 1:** *This work motivates and explains the advantage of the proposed method by LLMs' deep understanding of linguistic logic and structure rather than superficial embedding processing. However, such explanations lack support from experiments. More analysis can be included on the LLM side. For example, does an inferior LLM basic architecture (either old design or small model sizes) or a badly trained LLM (undertrained or untrained) lead to a bad time series performance？*
>
> _________________
> **Response:**
> Thank you for your constructive review suggestions. We agree that this experiment is interesting and essential, as it provides more favorable validation for our method. We have added the relevant content in Appendix G.1 of the revised version.
>
> As the table illustrates, we add experiments **on the LLM side**, using randomly initialized GPT-2 pre-trained weights in various proportions for under-trained and untrained scenarios. As the initialization proportion rises, the LLM's contextual understanding weakens, resulting in reduced performance of our method. When the model's capability is weak, our results are as poor as those of GPT4TS [1] (the most direct method to utilize LLM for TS tasks) and $S^{2}$IP-LLM [2] (a token-alignment based method). However, as the LLM's capability improves, our method significantly outperforms GPT4TS and $S^{2}$IP-LLM, achieving a lower MSE. In Appendix G.1 of the revision includes trend line graphs of the experimental results.This result demonstrates that our proposed Context-Alignment paradigm, which emphasizes LLMs' deep understanding of linguistic context, more effectively activates the potential of pre-trained LLMs in TS tasks.
>
> Besides, the original text includes experimental support for this explanation **on the method side**. Variant A.1, shown in Table 6, removes the Dual-Scale Context Alignment GNN framework, and Variant A.2 involves random initialization of GNN connectivity (i.e., flawed logic guidance), which significantly decreases performance. The more severe impact observed in A.2 suggests that disrupted logic and structure have more severe negative effects, highlighting LLMs' deep understanding of linguistic context.
>
> | Random initialized ratio | DECA on ETh1 | GTP4TS on ETh1 | $\text{S}^{2}$IP-LLM on ETh1 | DECA on ETh2 | GTP4TS on ETh2 | $\text{S}^{2}$IP-LLM on ETh2 |
> |---------------------------|--------------|----------------|--------------|--------------|----------------|--------------|
> | 0%                        | 0.394        | 0.427          | 0.418        | 0.316        | 0.354          | 0.355        |
> | 20%                       | 0.401        | 0.435          | 0.425        | 0.329        | 0.358          | 0.363        |
> | 40%                       | 0.407        | 0.442          | 0.438        | 0.335        | 0.367          | 0.365        |
> | 60%                       | 0.421        | 0.445          | 0.453        | 0.338        | 0.369          | 0.381        |
> | 80%                       | 0.465        | 0.477          | 0.481        | 0.404        | 0.416          | 0.411        |
> | 100%                      | 0.534        | 0.531          | 0.538        | 0.441        | 0.445          | 0.437        |

---

> ### Author Response · Authors · 2024-11-22
> **Response to Weakness 2**
>
> _________________
> **Weakness 2:** *This work freezes most of the LLM structure and tunes dual-scale context-alignment GNNs between Transformer layers. There seems to be lacking analysis on the effect of fully tuning the attention and feed-forward layers as well, and the subsequent necessity of the dual-scale context-alignment GNNs in that case.*
> _________________
> **Response:** Thank you for your valuable suggestions regarding our work. We have added the relevant content in Appendix G.3 of the revised version.
>
> In fact, GPT4TS [1] has already demonstrated experiments using fully tuning LLMs for TS tasks, which is the most straightforward approach (adding only two linear layers to the input and output ends of the LLMs) to leverage LLMs in this domain. However, fully tuning requires higher computational overhead, and yields suboptimal results. GPT4TS demonstrated through experiments and theoretical analysis that LLMs possess inherent generic knowledge that can be utilized directly, and that fully tuning LLMs would compromise this generic knowledge.
> Consequently, our approach, along with other TS analysis methods based on pre-trained LLMs (e.g. $S^{2}$IP-LLM[2], Time-LLM [3], TEST [4], aLLM4TS [5], and so on), focuses on freezing most of the LLM structure to explore efficient ways of activating their potential for TS tasks.
> However, we believe that the fully tuning experiment is necessary, given our significant differences from GPT4TS. Thank you for your reminder. We supplemented the results of our method with fully tuning, which yielded similarly suboptimal performance to GPT4TS.
>
> | Method               | ETTh1 | ETTm1 |
> |----------------------|-------|-------|
> | GPT4TS               | 0.427 | 0.352 |
> | GPT4TS (Fully tuning)| 0.469 | 0.406 |
> | DECA                 | 0.394 | 0.342 |
> | DECA (Fully tuning)  | 0.457 | 0.383 |

---

> ### Author Response · Authors · 2024-11-22
> **Response to Weakness 3**
>
> _________________
> **Weakness 3:** *One of the interesting attributes of LLMs is the scaling effect (e.g., on model sizes, training data sizes, few-shot prompting amounts). The scaling aspect of LLMs and the proposed DECA seems to be unknown in the context of time series.*
>
> _________________
> **Response:** Thank you very much for your advice. We conducted additional experiments to verify the effectiveness of our method under various conditions. We have added the relevant content in Appendix G.2 of the revised version.
>
> 1. On model sizes, we have conducted ablation experiments on the number of GPT-2 layers in variants Tab.6. As the number of layers increases, performance declines, consistent with observations made with GPT4TS[1] and aLLM4TS [5].
>
> 2. On training data sizes, we have trained models with 5% and 10% of the data in few-shot forecasting tasks, and further extended our experiments to settings using 25%, 50%, and 75% of the training data. Increasing data volumes continuously improve outcomes, especially notable at the 50% data point.
>
> 3. Regarding the amounts of few-shot prompting examples, experiments with various amounts reveal that as the number of examples increases, there is a modest improvement in short prediction length (96), but effectiveness diminishes with more examples. In contrast, for long prediction lengths like 336 or 720, more examples lead to worse outcomes. This could be due to our examples being derived from divisions of the TS input (where the total length of TS is 512). More divisions mean shorter lengths per example, yet the required prediction length is much longer, leading to adverse effects due to this mismatch.
>
> **The results of  training data sizes**
>
> | Training data ratio  | ETTh1 | ETTm1 |
> |-------|-------|-------|
> | 5%    | 0.575 | 0.435 |
> | 10%   | 0.538 | 0.435 |
> | 25%   | 0.486 | 0.411 |
> | 50%   | 0.409 | 0.366 |
> | 75%   | 0.398 | 0.350 |
> | 100%  | 0.394 | 0.342 |
>
> _________________
>
> **The results of  the amounts of few-shot prompting examples**
>
> | Examples amount |    1   |       |    2   |       |   3    |      |   4    |        |        |
> |-----------------|--------|--------|--------|--------|--------|--------|--------|--------|--------|
> | **Metric**      | **MSE**| **MAE**| **MSE**| **MAE**| **MSE**| **MAE**| **MSE**| **MAE**|
> | **ETTh1**       |        |        |        |        |        |        |        |        |
> | 96              | 0.349  | 0.389  | 0.343  | 0.385  | 0.341  | 0.382  | 0.356  | 0.394  |
> | 192             | 0.390  | 0.415  | 0.387  | 0.416  | 0.393  | 0.419  | 0.402  | 0.428  |
> | 336             | 0.402  | 0.432  | 0.407  | 0.439  | 0.414  | 0.445  | 0.440  | 0.456  |
> | 720             | 0.433  | 0.460  | 0.446  | 0.471  | 0.462  | 0.488  | 0.485  | 0.495  |
> | **Avg**         | 0.394  | 0.424  | 0.396  | 0.428  | 0.403  | 0.434  | 0.421  | 0.443  |
> | **ETTm1**       |        |        |        |        |        |        |        |        |
> | 96              | 0.282  | 0.343  | 0.277  | 0.340  | 0.275  | 0.341  | 0.296  | 0.352  |
> | 192             | 0.324  | 0.369  | 0.326  | 0.374  | 0.331  | 0.385  | 0.341  | 0.377  |
> | 336             | 0.356  | 0.386  | 0.366  | 0.391  | 0.370  | 0.395  | 0.391  | 0.408  |
> | 720             | 0.405  | 0.417  | 0.412  | 0.425  | 0.428  | 0.432  | 0.451  | 0.438  |
> | **Avg**         | 0.342  | 0.378  | 0.345  | 0.383  | 0.351  | 0.388  | 0.370  | 0.394  |

---

> ### Author Response · Authors · 2024-11-22
> **Response to Weakness 4**
>
> _________________
> **Weakness 4:** *The writing can be improved for clarity, e.g., by providing qualitative examples besides key equations, especially in the few-shot prompting part, and adding an algorithm table.*
>
> _________________
> **Response:** Thank you for your valuable suggestions, which are crucial for enhancing the reading experience of our paper.
>
> In the revised Appendix E&F, we have supplemented the qualitative examples and algorithm tables. We have also added a diagram illustrating data processing and graph construction (including the few-shot prompting and VCA part).
>
> Here, we further explain the few-shot prompting part with an example for your clarity, which can also be found in the revision.
>
> Assuming we have inputs processed through patching and token embedding, these include TS embeddings of length 8 and task description prompt embeddings of length 2 (the prompt is “Predict future sequences using previous data:" in our method, here the length is an example) :
>
> Firstly, for the fine-grained branch, let's take the example where the input TS embeddings are divided into 2 subsequences, each containing 4 embeddings. Thus, $TS_{sub}^{1}$ is $[\mathbf e_{1,1},\mathbf e_{1,2},\mathbf e_{1,3},\mathbf e_{1,4}]$, and $TS_{sub}^{2}$ is $[\mathbf e_{2,1},\mathbf e_{2,2},\mathbf e_{2,3},\mathbf e_{2,4}]$, where $\mathbf e_{i,j}$ indicates $j$-th embedding in subsequence $i$. Similarly, $\mathbf z_{i,j}$ refers to $j$-th embedding in the task prompt of subsequence $i$. Here, $[\mathbf z_{1,1}, \mathbf z_{1,2}]=[\mathbf z_{2,1}, \mathbf z_{2,2}]$. Ultimately, Eq.5 is instantiated as:
> $[\mathbf e_{1,1},\mathbf e_{1,2},\mathbf e_{1,3},\mathbf e_{1,4}, \mathbf z_{1,1}, \mathbf z_{1,2}, \mathbf e_{2,1},\mathbf e_{2,2},\mathbf e_{2,3},\mathbf e_{2,4}, \mathbf z_{2,1}, \mathbf z_{2,2}]$.
>
> Secondly, we need to construct a graph structure for this input before it enters LLM. The basic logic for constructing the graph is that $TS_{sub}^{2}$ serves as the ground truth for $TS_{sub}^{1}$ (The latter subsequence serves as the correct label for the former subsequence). Specifically, starting with all elements in $TS_{sub}^{1}$, construct directed edges to the first item of the corresponding task description, $\mathbf z_{1,1}$. Subsequently, from the last item of the task description, $\mathbf z_{1,2}$, construct directed edges to all elements in $TS_{sub}^{2}$. Since all TS subsequences are used to predict future sequences, the first token of the last prompt, $\mathbf{z}_{2,1}$, needs to establish edge connections with both TS subsequences.
>
> Thirdly, for the coarse-grained branch, it is essential to inform the LLM that a time series should be treated as a whole. Thus, $TS_{sub}^{i}$ must be mapped to individual node embedding by a linear layer. To align the scales, the prompt embeddings are also mapped to a node embeding. The coarse-grained sequences can be denoted as $[\tilde{\mathbf{e}}_1, \tilde{\mathbf{z}}^{(1)}, \tilde{\mathbf{e}}_2, \tilde{\mathbf{z}}^{(2)}]$ (instantiation of Eq.6). Additionally, the graph construction logic is consistent with that of the fine-grained branch.

---

> ### Author Response · Authors · 2024-11-22
> **References**
>
> [1] Zhou T, Niu P, Sun L, et al. One fits all: Power general time series analysis by pretrained lm[J]. Advances in neural information processing systems, 2023, 36: 43322-43355.
>
> [2] Pan Z, Jiang Y, Garg S, et al. $ S^ 2$ IP-LLM: Semantic Space Informed Prompt Learning with LLM for Time Series Forecasting[C]//Forty-first International Conference on Machine Learning. 2024.
>
> [3] Jin M, Wang S, Ma L, et al. Time-llm: Time series forecasting by reprogramming large language models[J]. arXiv preprint arXiv:2310.01728, 2023.
>
> [4] Sun C, Li H, Li Y, et al. TEST: Text Prototype Aligned Embedding to Activate LLM's Ability for Time Series[C]//The Twelfth International Conference on Learning Representations.
>
> [5] Bian Y, Ju X, Li J, et al. Multi-Patch Prediction: Adapting Language Models for Time Series Representation Learning[C]//Forty-first International Conference on Machine Learning.

---

> ### Author Response · Authors · 2024-11-25
>
> We sincerely thank you for your valuable questions, and we are more than happy to address them. We have supplemented the experiments and included the relevant corrections and explanations in the revised version (marked in red). The specific locations of the revision have been added in our previous itemized response. Based on your comments, we believe that you have well understood the background, motivation, and key contributions of our work. We hope you might slightly consider increasing your confidence or potentially raising your evaluation score.
>
> Once again, we truly appreciate your time and effort for our paper.

---

> ### Author Response · Authors · 2024-12-01
> **Reminder from Authors**
>
> Dear Reviewer Dkrj,
>
> Thank you for your feedback during the review process! We believe that our detailed response has addressed your concerns. If you have any concerns or questions, please do not hesitate to let us know before the author discussion period ends (less than two days). We will be happy to answer them during the discussion.
>
> Thank you!

---

### Author Response · Authors · 2024-11-22
**Background & Challenges & Distinctions & Summary of revisions**

We appreciate all reviewers for recognizing the contribution of our paper. In particular, reviewers 2N38, oFgL, and N4CB for recognizing the novelty of our paper, with N4CB further affirming logical model design. Following our supplementary experiments and responses, reviewers oFgL and N4CB have improved their overall assessments. Our thanks go to all reviewers and the AC for their support.

We summarize the research background, challenges, and key distinctions of our method below, then discuss each comment specifically in the reviewers' responses.

**Research Background:** Time series (TS) datasets span multiple domains such as medical, industrial, transportation, power, etc. Thus, there is a critical need for models like LLMs that can generalize across diverse domains. Recent research, including our method, aims to efficiently use pre-trained LLMs for time series tasks. Particularly, GPT4TS [1] has proven, both theoretically and empirically, that LLMs possess generic knowledge useful for these tasks.

**Challenges:** The significant differences between the training text data for LLMs and TS data hinder the effective activation of LLM capabilities in TS tasks.

**Distinction from Mainstream Token-Level Alignment Methods:** Token-alignment methods utilize a vocabulary and employ various techniques to align TS inputs with vocabulary words that describe temporal features, such as rise, fall, periodic, steady, short, long, and so on, facilitating comprehension by LLMs. Differing from these methods, we leverage the deep understanding of linguistic context by LLMs, and inspired by insights from the field of linguistics, to propose the Context-Alignment (including logical alignment and structural alignment). Context-Alignment aligns TS with a linguistic component to enable LLMs to contextualize and comprehend TS data, thereby activating their capabilities.

**Distinction from Directly Enhancing LLMs' Capabilities on TS:** We argue that without LLMs understanding TS data, enhancements are less interpretable and less effective. Thus, we first suggest a two-step route: activate, then enhance. Our Vanilla Context-Alignment (VCA) activates LLM capabilities through Context-Alignment, while our final method, DECA, extends VCA with Few-shot Prompting to boost performance, following our proposed steps scheme.

**Summary of revisions:** Following reviewer feedback, we primarily made the following modifications:

1. Renamed DECA to FSCA (Few-Shot Prompting based Context-Alignment) as suggested by Reviewer N4CB.

2. Added qualitative examples, algorithm tables, and a schematic diagram of the graph structure in Appendices E,F, as recommended by Reviewers Dkrj and 2N38.

3. Incorporated additional experiments and analyses in Appendix G, focusing on computational efficiency, comparisons with TS foundation models, and ablation studies such as alternative network structures to GNNs, based on broader reviewer suggestions.

[1] Zhou T, Niu P, Sun L, et al. One fits all: Power general time series analysis by pretrained lm[J]. Advances in neural information processing systems, 2023, 36: 43322-43355.

---

### Meta-Review · Area_Chair_1hBz · 2024-12-22

**Metareview:**

This paper proposes a novel approach to leveraging LLMs for time series tasks by aligning time series data to linguistic contexts through dual-scale context-alignment GNNs (DSCA-GNNs). The authors validate their method with extensive experiments across forecasting, classification, and anomaly detection, demonstrating improved performance over several baselines, particularly in few-shot and zero-shot scenarios.

Strengths:
The paper tackles an interesting problem of adapting LLMs for time series tasks. The proposed context alignment induces "structural and logical alignment". The paper includes a wide range of experiments with meaningful results.

Weaknesses:
Despite its strengths, the paper has several limitations. It lacks sufficient comparisons to non-LLM-based pretrained models in its original submission, which was mostly addressed during the rebuttal.

Decision:
Based on strong empirical performance thorough revisions during the rebuttal, I recommend acceptance. The contributions set a strong foundation for future work in integrating LLMs with time series data.

**Additional Comments On Reviewer Discussion:**

The review process highlighted several important points:

1. Comparison with non-LLM-based pretrained models:
Reviewers suggested comparisons to pre-trained time series models (e.g., UniTS-ST, MOMENT), which the authors incorporated during the rebuttal. The added experiments demonstrated the superiority of the proposed method in most tasks.

2. Evaluation of GNN choice:
Concerns about GNNs' effectiveness compared to other architectures (e.g., CNNs, self-attention) were addressed with additional ablation studies. These confirmed that GNNs performed best for context alignment.

3. Complexity and computational overhead:
The authors clarified computational costs, showing that while their method only introduces modest overhead, it remains competitive with baseline methods.

---

### Decision · Program_Chairs · 2025-01-22

Accept (Poster)